

SciPost Phys. Lect. Notes 72 (2023)

# Quantum computation with cat qubits

**Jérémie Guillaud[1], Joachim Cohen[2] and Mazyar Mirrahimi[3]⋆**

**1** Alice and Bob, 53 Bd du Général Martial Valin, 75015, Paris, France
**2** Université de Sherbrooke, Sherbrooke, Quebec J1K2R1, Canada
**3** Laboratoire de Physique de l'Ecole Normale Supérieure, Inria, ENS-PSL, Mines-Paristech, CNRS, Sorbonne Université, PSL Research University, Paris, France

⋆ mazyar.mirrahimi@inria.fr

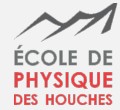

*Part of the Quantum Information Machines*
*Session 113 of the Les Houches School, July 2019*
*published in the Les Houches Lecture Notes Series*

## Abstract

These are the lecture notes from the 2019 Les Houches Summer School on "Quantum Information Machines". After a brief introduction to quantum error correction and bosonic codes, we focus on the case of cat qubits stabilized by a nonlinear multi-photon driven dissipation process. We argue that such a system can be seen as a self-correcting qubit where bit-flip errors are robustly and exponentially suppressed. Next, we provide some experimental directions to engineer such a multi-photon driven dissipation process with superconducting circuits. Finally, we analyze various logical gates that can be implemented without re-introducing bit-flip errors. This set of bias-preserving gates pave the way towards a hardware-efficient and fault-tolerant quantum processor.

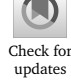

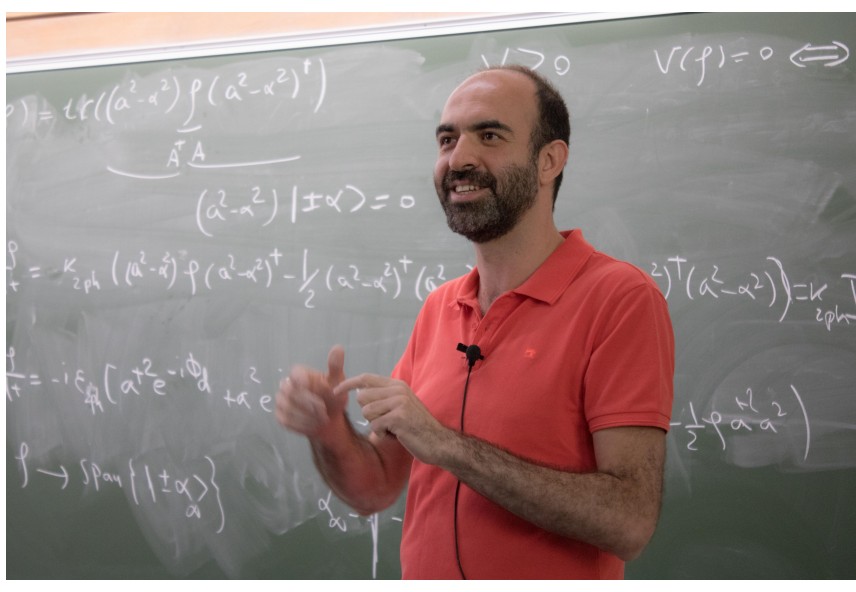

# 1   Introduction

Decoherence is regarded as the major obstacle towards scalable and robust processing of quantum information. It is caused by the interaction of a quantum system with its noisy environment, through which the system gets entangled with an infinite number of degrees of freedom. Despite one's effort to isolate the quantum system of interest, some amount of undesired interaction persists limiting the lifetime of the information. In the case of superconducting qubits, this effort over the past two decades has led to an increase of the lifetime from a few nanoseconds [1] to $100\mu$s-1ms in best cases [2–4]. Quantum Error Correction (QEC) has emerged as an inevitable tool to go beyond this limitation and significantly enhance the lifetime of quantum information [5, 6]. By designing an encoded logical qubit, possibly using many physical qubits, one protects the quantum information against major decoherence channels and hence

ensures a longer coherence time than a physical qubit [7,8].[1]

## 1.1 Quantum Error Correction

The contents of this section are strongly inspired by [5] and [11], and set a general framework for QEC. In Subsection 1.1.1, we briefly introduce the formalism of open quantum systems. More precisely, we give a general description of quantum operations acting on a system and in particular modeling the occurring errors. Such a quantum error channel is described in Subsection 1.1.2 through the analysis of a quantum code, the three-qubit bit-flip code. Next, the Subsection 1.1.3 recalls some general results on QEC. In particular, given a quantum code, it provides a necessary and sufficient condition for a set of errors to be correctable.

### 1.1.1 Quantum maps and decoherence

The state of an open quantum system $S$ is described by a density matrix $\boldsymbol{\rho}_S$, a semi-definite positive hermitian operator of unit trace, defined on the Hilbert space $\mathcal{H}_S$ of the system. After a time interval $\tau$, the state of this open system is updated by a trace-preserving *quantum operation*

$$\forall \boldsymbol{\rho}_S, \quad \mathbb{E}(\boldsymbol{\rho}_S) = \sum_\mu \boldsymbol{M}_\mu \boldsymbol{\rho}_S \boldsymbol{M}_\mu^\dagger. \tag{1}$$

Here, the trace-preserving property of the above operation is ensured via the relation $\sum_\mu \boldsymbol{M}_\mu^\dagger \boldsymbol{M}_\mu = \boldsymbol{I}_S$. The *superoperator* $\mathbb{E}$ is also called a quantum map or a Kraus map, and $\{\boldsymbol{M}_\mu\}$ is a set of associated Kraus operators. This choice of Kraus operators is not unique as the operators $\tilde{\boldsymbol{M}}_\mu = \sum_\nu r_{\mu,\nu} \boldsymbol{M}_\nu$, with $(r_{\mu,\nu})$ an arbitrary unitary matrix, satisfy $\sum_\mu \tilde{\boldsymbol{M}}_\mu \boldsymbol{\rho}_S \tilde{\boldsymbol{M}}_\mu^\dagger = \sum_\mu \boldsymbol{M}_\mu \boldsymbol{\rho}_S \boldsymbol{M}_\mu^\dagger$ for all $\boldsymbol{\rho}_S$. Note that, a pure state of the quantum system S, corresponding to a density matrix of the form $\boldsymbol{\rho}_S = |\psi\rangle\langle\psi|$, is generally mapped to a mixed state, therefore leading to its decoherence.

So far, we have modeled the harmful decoherence phenomena as a quantum operation. We will see throughout this chapter, that it is also possible to engineer a particular quantum operation that rather purifies a state by evacuating the entropy of the quantum system. More precisely, such a quantum operation $\mathbb{R}$ can correct the decoherence (given by $\mathbb{E}$) acting on a manifold $\mathcal{C} \subset \mathcal{H}_S$, where the information is encoded:

$$(\mathbb{R} \circ \mathbb{E})(\boldsymbol{\rho}) = \boldsymbol{\rho}, \qquad \forall \boldsymbol{\rho} \in \mathcal{C}.$$

### 1.1.2 An example: three-qubit bit-flip code

In classical information theory, one can protect a logical bit of information against bit-flip errors, by encoding it, redundantly, in three bits: $0 \rightarrow 0_L = 000$ and $1 \rightarrow 111$. Provided that the probability $p$ for an error to occur on a bit is small, and that the errors are not correlated, this code prevents these errors from damaging the information. Indeed, through a majority voting, the erroneous state 100, is associated to 000. This reduces the error probability from $p$ to $3p^2$ (case where two bits have flipped).

There exists a direct quantum analog, called the three-qubit bit-flip code, which protects the information against single bit-flip errors mapping an arbitrary superposition state $c_0|0\rangle + c_1|1\rangle$ of a qubit to $c_0|1\rangle + c_1|0\rangle$. Three qubits are used to encode a single logical qubit with $|0_L\rangle = |000\rangle$ and $|1_L\rangle = |111\rangle$. Starting from a superposition in the *code space* $\mathcal{E}_0 = \mathrm{span}\{|000\rangle, |111\rangle\}$, a single bit-flip error maps the states to one of the *error subspaces* $\mathcal{E}_1 = \mathrm{span}\{|100\rangle, |011\rangle\}$, $\mathcal{E}_2 = \mathrm{span}\{|010\rangle, |101\rangle\}$ or $\mathcal{E}_3 = \mathrm{span}\{|001\rangle, |110\rangle\}$. We can associate

[1]Some parts of Ref. [9] and Ref. [10] have been reused with permission of APS and IOP under CC BY (Creative Commons Attribution 4.0 International License).

to these error processes, the Kraus operators $M_0 = \sqrt{1-3p}\mathbf{I}$, $M_1 = \sqrt{p}\sigma_x^1$, $M_2 = \sqrt{p}\sigma_x^2$ and $M_3 = \sqrt{p}\sigma_x^3$, where $p \ll 1$ is the probability of a single bit-flip, $\mathbf{I}$ is the identity on the qubits Hilbert space, and $\sigma_x^k$ is the Pauli matrix along the $X$ axis of the $k$'th qubit. The associated quantum operation $\mathbb{E}$ reads

$$\forall \boldsymbol{\rho} \, , \ \mathbb{E}(\boldsymbol{\rho}) = (1-3p)\boldsymbol{\rho} + p\sigma_x^1 \boldsymbol{\rho}\sigma_x^1 + p\sigma_x^2 \boldsymbol{\rho}\sigma_x^2 + p\sigma_x^3 \boldsymbol{\rho}\sigma_x^3 \, .$$

The measurement of the two-qubit parities $\sigma_z^1\sigma_z^2$ and $\sigma_z^2\sigma_z^3$ reveals the subspace on which the three-qubit system lies, without leaking information about the quantum superposition. The subspace $\mathcal{E}_0$ corresponds to error syndrome $\{\sigma_z^1\sigma_z^2, \sigma_z^2\sigma_z^3\} = \{1,1\}$, $\mathcal{E}_1$ to $\{-1,1\}$, $\mathcal{E}_2$ to $\{-1,-1\}$, and $\mathcal{E}_3$ to $\{1,-1\}$. One can recover the initial state by applying the inverse operation. Here, it corresponds to flipping back the qubit on which the error occurred. This recovery operation $\mathbb{R}$ is defined by

$$\forall \boldsymbol{\rho}, \ \mathbb{R}(\boldsymbol{\rho}) = \Pi_{\mathcal{E}_0}\boldsymbol{\rho}\Pi_{\mathcal{E}_0} + \sigma_x^1\Pi_{\mathcal{E}_1}\boldsymbol{\rho}\Pi_{\mathcal{E}_1}\sigma_x^1 + \sigma_x^2\Pi_{\mathcal{E}_2}\boldsymbol{\rho}\Pi_{\mathcal{E}_2}\sigma_x^2 + \sigma_x^3\Pi_{\mathcal{E}_3}\boldsymbol{\rho}\Pi_{\mathcal{E}_3}\sigma_x^3 \, ,$$

where $\Pi_{\mathcal{E}_i}$ is the projector on the subspace $\mathcal{E}_i$. One can easily check that $\mathbb{R}$ is a recovery operation for the error map $\mathbb{E}$, with

$$\forall \boldsymbol{\rho} \in \mathcal{E}_0, \ (\mathbb{R} \circ \mathbb{E})(\boldsymbol{\rho}) = \boldsymbol{\rho} \, .$$

Similarly, the three-qubit phase-flip code protects a logical qubit against single phase flips, mapping $c_0|0\rangle + c_1|1\rangle$ to $c_0|0\rangle - c_1|1\rangle$. Encoding the information in the basis $|0_L\rangle = |+++\rangle$ and $|1_L\rangle = |---\rangle$ with $|\pm\rangle = (|0\rangle \pm |1\rangle)/\sqrt{2}$, the error syndromes are provided by the measurement of the two-qubit operators $\sigma_x^1\sigma_x^2$ and $\sigma_x^2\sigma_x^3$.

### 1.1.3 Basics of quantum error correction

In quantum error correction, one encodes the information in a subspace $\mathcal{C}$, the *code space*, of a larger Hilbert space. The decoherence channels are described by quantum maps (described by a set of Kraus operators) acting on the code space. These Kraus operators are referred to as the *errors*. The protection by QEC is characterized by the code space $\mathcal{C}$ and the images of this code space through various errors. We start by giving a necessary and sufficient condition for the Kraus operators, to ensure the existence of a recovery operation. Next, through an error discretization theorem, we explain that linear combinations of correctable errors remain correctable by the same code. From this theorem, one infers that an arbitrary single-qubit error can be corrected with a code correcting for bit flips, phase flips, and simultaneous bit flip and phase flip. Finally, we present a well-known example of such a quantum code, the so-called Shor code [7].

**Quantum error correction condition**

Let us consider that a quantum system of interest is subject to a noise map $\mathbb{E}$, represented by a set of Kraus operators (or errors) $\{M_\mu\}$. The logical information is encoded in a subspace $\mathcal{C}$. Can we find a quantum operation $\mathbb{R}$ that recovers the initial state, i.e $\forall \boldsymbol{\rho} \in \mathcal{C}, \ (\mathbb{R} \circ \mathbb{E})(\boldsymbol{\rho}) = \boldsymbol{\rho}$?

A central theorem in QEC theory addresses this problem by giving a necessary and sufficient condition on the errors $M_\mu$ (Theorem 10.1 of [5]). There exists a recovery operation $\mathbb{R}$ for the noise map $\mathbb{E}$, if and only if there exists a Hermitian matrix $(c_{\mu\nu})$ satisfying

$$\Pi_{\mathcal{C}} M_\mu^\dagger M_\nu \Pi_{\mathcal{C}} = c_{\mu\nu}\Pi_{\mathcal{C}} \, . \tag{2}$$

Here $\Pi_{\mathcal{C}}$ is the projection operator over $\mathcal{C}$. Under this condition, $\{M_\mu\}$ is a set of *correctable errors* for the code defined by $\mathcal{C}$.

Let us provide an intuitive explanation for this theorem. First, we can choose a more suitable set of Kraus operators $\{E_\nu\}$ for the map $\mathbb{E}$, such that condition (2) becomes

$\Pi_{\mathcal{C}} E_\mu^\dagger E_\nu \Pi_{\mathcal{C}} = d_\mu \delta_{\mu,\nu} \Pi_{\mathcal{C}}$, with $d_\mu > 0$ and $\sum_\mu d_\mu = 1$. Furthermore, $\delta_{\mu,\nu} = 1$ if and only if $\mu = \nu$ and $\delta_{\mu,\nu} = 0$ otherwise. This change of Kraus operators is justified in the next paragraph. As two distinct errors $E_\mu$ and $E_\nu$ satisfy $\Pi_{\mathcal{C}} E_\mu^\dagger E_\nu \Pi_{\mathcal{C}} = 0$, they map the code space $\mathcal{C}$ to mutually orthogonal error subspaces $\mathcal{E}_\mu$ and $\mathcal{E}_\nu$. These errors can be unambiguously diagnosed by measuring a set of commuting observables which admit the subspaces $\mathcal{E}_\nu$ as common eigenspaces. In addition, considering an orthonormal basis, $\{|i_L\rangle\}$, of the code space, through the relation $\langle i_L | \frac{E_\mu^\dagger}{\sqrt{d_\mu}} \frac{E_\mu}{\sqrt{d_\mu}} | j_L \rangle = \delta_{i,j}$, an error $E_\mu$ rotates this logical basis to an orthonormal basis of the error subspace $\mathcal{E}_\mu$. This ensures that once an error is diagnosed, one can reverse the operation by applying the inverse unitary. Equivalently, it means that no information about the logical superposition is leaked to the environment through the error channels $E_\mu$. More precisely, we have $E_\mu \Pi_{\mathcal{C}} = \sqrt{d_\mu} U_\mu \Pi_{\mathcal{C}}$, with $U_\mu$ a unitary operation, and therefore $\Pi_{\mathcal{E}_\mu} = U_\mu \Pi_{\mathcal{C}} U_\mu^\dagger$ is the projector on the error subspace $\mathcal{E}_\mu$. The mutual orthogonality of the error subspaces is expressed through the relation $\Pi_{\mathcal{E}_\mu} \Pi_{\mathcal{E}_\nu} = \delta_{\mu,\nu} \Pi_{\mathcal{E}_\mu}$. The quantum operation $\mathbb{R}$ described by the Kraus operators $\{R_\nu = U_\nu^\dagger \Pi_{\mathcal{E}_\nu}\}$ is a recovery map for the quantum operation $\mathbb{E}$. Indeed, for an initial state $\rho \in \mathcal{C}$, we have

$$
\begin{aligned}
(\mathbb{R} \circ \mathbb{E})(\rho) &= \sum_{\nu,\mu} R_\nu E_\mu \rho E_\mu^\dagger R_\nu^\dagger \\
&= \sum_{\nu,\mu} d_\mu U_\nu^\dagger \Pi_{\mathcal{E}_\nu} \Pi_{\mathcal{E}_\mu} U_\mu \rho U_\mu^\dagger \Pi_{\mathcal{E}_\mu} \Pi_{E_\nu} U_\nu \\
&= \sum_\nu d_\nu U_\nu^\dagger U_\nu \rho U_\nu^\dagger U_\nu = \rho \,.
\end{aligned}
$$

In this paragraph, we justify the existence of a set of operators $\{E_\nu\}$ for the map $\mathbb{E}$, such that condition (2) becomes $\Pi_{\mathcal{C}} E_\mu^\dagger E_\nu \Pi_{\mathcal{C}} = d_\mu \delta_{\mu,\nu} \Pi_{\mathcal{C}}$. The hermitian matrix $c$ of eq. (2) can be written as $c = p d p^\dagger$, with $d$ a diagonal matrix and $p$ a unitary matrix. We define the operators $E_\nu = \sum_\mu p_{\mu\nu} M_\mu$. They satisfy

$$
\Pi_{\mathcal{C}} E_{\nu_1}^\dagger E_{\nu_2} \Pi_{\mathcal{C}} = \left( \sum_{\mu_1,\mu_2} p_{\mu_1 \nu_1}^* p_{\mu_2 \nu_2} c_{\mu_1 \mu_2} \right) \Pi_{\mathcal{C}} = \delta_{\nu_1,\nu_2} d_{\nu_1,\nu_1} \Pi_{\mathcal{C}} \,,
$$

as $d_{\nu_1,\nu_2} = \sum_{\mu_1,\mu_2} p_{\mu_1 \nu_1}^* p_{\mu_2 \nu_2} c_{\mu_1 \mu_2}$ stems from the relations $d = p^\dagger c p$. Following Subsection 1.1.1, the matrix $p$ being unitary, the map $\mathbb{E}$ is equivalently described by the set of Kraus operators $\{E_\nu\}$.

### Error discretization

Provided that the logical information is encoded in a code space $\mathcal{C}$, the condition (2) states the existence of a recovery operation for a given set of errors. Here we see that, a quantum code can be subject to an infinite number of noise maps, and still remain correctable. It would greatly simplify the design of QEC protocols, if a same correction operation $\mathbb{R}$ could work for various correctable sets of errors. Fortunately, this is the case through the following sufficient condition [5]:

Consider $\{E_\nu\}$ a set of correctable errors associated to a noise map $\mathbb{E}$, and an associated recovery operation $\mathbb{R}$. Let $\mathbb{F}$ be the noise map represented by the set of errors $\{F_\mu\}$, where the operators $F_\mu$ are linear combinations of the operators $E_\nu$. Then the set $\{F_\mu\}$ is a correctable set of errors with the same recovery operation $\mathbb{R}$.

While this statement can be easily proven by inserting into the equation $\mathbb{R} \circ \mathbb{F}(\rho) = \rho$, the relation $F_\mu = \sum_\nu \lambda_{\mu\nu} E_\nu$, here we provide a more physical insight into this result. The noise map $\mathbb{F}$ can be represented by a unitary operation $U_{SE}$ acting on the system and an

environment E, with $U_{SE}(|\psi\rangle_S \otimes |g_{\mu_0}\rangle_E) = \sum_\mu (F_\mu |\psi\rangle_S) \otimes |g_\mu\rangle_E$, where $\{|g_\mu\rangle_E\}$ form an orthonormal basis for the Hilbert space of $E$. Here, we have assumed the initial state of system $S$ to be a pure state for simplicity sakes. By inserting $F_\mu = \sum_\nu \lambda_{\mu\nu} E_\nu$, we obtain $U_{SE}(|\psi\rangle_S \otimes |g_{\nu_0}\rangle_E) = \sum_\nu (E_\nu |\psi\rangle_S) \otimes |e_\nu\rangle_E$, with the states $|e_\nu\rangle_E = \sum_\mu \lambda_{\mu\nu} |g_\mu\rangle_E$. Note that, the states $|e_\nu\rangle_E$ are not necessarily orthogonal as the matrix $(\lambda_{\mu\nu})$ is not necessarily unitary. The recovery operation $\mathbb{R}$ associated to the error set $\{E_\nu\}$, can be described by the set of Kraus operators $R_\nu = U_\nu^\dagger \Pi_{\mathcal{E}_\nu}$ (see previous paragraph). Similarly to the noise map $\mathbb{E}$, the quantum operation $\mathbb{R}$ can be equivalently represented by a unitary operation $U_{SA}$ acting on the system S and an ancillary system A, such that $U_{SA}(|\psi\rangle_S \otimes |a_{\mu_0}\rangle_A) = \sum_\mu (R_\mu |\psi\rangle_S) \otimes |a_\mu\rangle_A$ for all states $|\psi\rangle_S$. Therefore, for a state $|\psi\rangle_S \in \mathcal{C}$, the state after the correction reads

$$
\begin{aligned}
U_{SA}[U_{SE}(|\psi\rangle_S \otimes |g_{\nu_0}\rangle_E) \otimes |a_{\mu_0}\rangle_A] &= \sum_\nu U_{SA}[(E_\nu |\psi\rangle_S \otimes |a_{\mu_0}\rangle_A)] \otimes |e_\nu\rangle_E \\
&= \sum_{\mu,\nu} (U_\mu^\dagger \Pi_{\mathcal{E}_\mu} E_\nu |\psi\rangle_S) \otimes |e_\nu\rangle_E \otimes |a_\mu\rangle_A \\
&= |\psi_S\rangle \otimes [\sum_\nu \sqrt{d_\nu} |e_\nu\rangle_E \otimes |a_\nu\rangle_A].
\end{aligned}
$$

Here, to obtain the third line from the second one, we have used the fact that $U_\mu^\dagger \Pi_{\mathcal{E}_\mu} E_\nu = d_\nu \delta_{\mu,\nu} \Pi_{\mathcal{C}}$. Note that the state of the environment and the ancilla $\sum_\nu \sqrt{d_\nu} |e_\nu\rangle_E \otimes |a_\nu\rangle_A$ does not depend on $|\psi_S\rangle$, which means that no information on this state has leaked out to the environment nor the ancilla.

**Example: a multi-qubit code**

The theory provided in previous two subsections applies to general QEC schemes. In particular, they apply to encodings on a single quantum harmonic oscillator, where the redundancy is insured through the infinite dimensional Hilbert space. Such codes will be discussed in Section 1.3. In this subsection, though, we focus on multi-qubit codes similar to three-qubit bit-flip code. As we shall see, the single-qubit errors can be cast into three types of errors.

Here, we study the effect of a noise map defined by the Kraus operators $\{E_\mu\}$ on a system S composed of $n$ qubits. Let us denote $X$, $Y$ and $Z$ the standard Pauli matrices, and $I$ the identity on a qubit's Hilbert space. As $\{I, X, Y, Z\}$ forms a basis for the space of linear operators on $\mathbb{C}^2$, the Kraus operators $E_\mu$ are linear combinations of operators of the Pauli group $G_n = \{I, X, Y, Z\}^{\otimes n}$. In particular, a single-qubit error is a linear combination of the operators $I$, $X_i$, $Y_j$ and $Z_k$ acting on a single qubit, where $X_i$ is the operator that acts as $X$ on the qubit $i$ and as the identity on the other qubits (idem for $Y_j$ and $Z_k$). An error $X_k$ is called a bit-flip error as it maps $|0\rangle \longleftrightarrow |1\rangle$, and an error $Z_k$ is a phase-flip error, since it maps $|0\rangle \rightarrow |0\rangle$ and $|1\rangle \rightarrow -|1\rangle$. The error $Y = iZX$ can be seen as a simultaneous bit flip and phase flip.

From the error discretization theorem follows a remarkable corollary. Consider the set of single qubit errors $\{I, X_j, Y_j, Z_j, \ j = 1...n\}$. Let us assume that this set (unnormalized here) is a correctable set of errors and $\mathbb{R}$ a recovery operation for this set. Then any arbitrary single qubit noise map is correctable by the map $\mathbb{R}$. In other words, to protect the information against any kind of noise occurring on a single qubit, it is enough to correct for single phase flips, bit flips and simultaneous bit flips and phase flips. The Shor code [7], presented below, provides such a protection.

One can encode a single logical qubit using nine "physical" qubits. The idea consists in a concatenation of a three-qubit bit-flip code with a three-qubit phase-flip code. First, we group the qubits three by three, and each group encodes a single intermediate logical qubit via the three-qubit bit-flip code. Next, the three intermediate qubits, protected against bit flips, are used to encode a single logical qubit through the three-qubit phase-flip code. The logical states,

$|0_L\rangle$ and $|1_L\rangle$, are given by

$$|0_L\rangle = |+\rangle_{1,L} \otimes |+\rangle_{2,L} \otimes |+\rangle_{3,L} = \frac{(|000\rangle + |111\rangle) \otimes (|000\rangle + |111\rangle) \otimes (|000\rangle + |111\rangle)}{2\sqrt{2}},$$

$$|1_L\rangle = |-\rangle_{1,L} \otimes |-\rangle_{2,L} \otimes |-\rangle_{3,L} = \frac{(|000\rangle - |111\rangle) \otimes (|000\rangle - |111\rangle) \otimes (|000\rangle - |111\rangle)}{2\sqrt{2}}.$$

Let us study the effect of single bit flips and single phase flips on the code space. The logical qubit is protected against bit flips as it is encoded by intermediate ones which are themselves protected. More precisely, a single bit-flip error occurring on the first three qubits is revealed by measuring the two joint-parties $Z_1 Z_2$ and $Z_2 Z_3$, and so on for the two other groups of three qubits. Hence, single bit flips are revealed by measuring the six two-qubit parities $\{Z_1 Z_2, Z_2 Z_3, Z_4 Z_5, Z_5 Z_6, Z_7 Z_8, Z_8 Z_9\}$. Next, by construction, the logical qubit is protected against the single phase flips of the intermediate logical qubits. Note, however, that a single phase flip occurring on any of the three "physical" qubits, results in a phase flip of the intermediate qubit. These phase flips are identified by the measurement of the operators $X_{1,L} X_{2,L} = X_1 X_2 X_3 X_4 X_5 X_6$ and $X_{2,L} X_{3,L} = X_4 X_5 X_6 X_7 X_8 X_9$, where $X_{1,L}$, $X_{2,L}$ and $X_{3,L}$ are the logical $X$ operators of the intermediate qubits. If a phase flip on the first intermediate qubit is diagnosed, the initial state is recovered by applying any of the operators $Z_1$, $Z_2$ and $Z_3$. Finally, note that an error $Y_i$ is diagnosed as a bit flip $X_i$, and a phase flip $Z_{j,L}$ of the corresponding intermediate qubit. Consequently, such an error is also correctable.

Since the Shor code corrects single qubit errors $X_i$, $Y_i$ and $Z_i$, it corrects any arbitrary single qubit errors.

## 1.2 Autonomous quantum error correction

In the previous subsection, we have discussed some general results on QEC. The system, redundantly encoding the logical information, is subject to a noise map $\mathbb{E}$. The initial state is then restored through a recovery procedure represented by the map $\mathbb{R}$. However, we haven't discussed yet how these recovery operations are physically implemented. The subsection 1.2.1 introduces the concept of continuous QEC versus discrete QEC. The Subsection 1.2.2 presents reservoir engineering as a mean to achieve continuous autonomous QEC, and provides a few examples of existing QEC schemes based on this method.

### 1.2.1 Continuous QEC versus discrete QEC

So far, we have implicitly adopted a discrete vision of QEC. The system undergoes a noise map, followed by a correction step. The noise map $\mathbb{E}_T$ corresponds to the evolution super-operator over a time duration $T$ of the system: $\rho(t + T) = \mathbb{E}_T(\rho)$. Consider that after each time interval $T_{\text{error}}$, one applies a recovery operation $\mathbb{R}$. The state of the system at time $t = nT_{\text{error}}$, is $\rho(nT_{\text{error}}) = [(\mathbb{R} \circ \mathbb{E}_{T_{\text{error}}}) \circ \cdots \circ (\mathbb{R} \circ \mathbb{E}_{T_{\text{error}}})](\rho(0))$. Here, the time $T_{\text{error}}$ between two successive recovery operations, is assumed to be small enough, so that the error model remains simple enough to be correctable. This recovery operations often involves a projective measurement of some error syndromes followed by an appropriate unitary operation (see Subsection 1.1.3). While the above description neglects the finite time needed for the recovery operation, the finite bandwidth of the measurement protocol usually limits the performance. This aspect was carefully analyzed in the experimental work of Kelly et al. [12], where a repetition bit-flip code was realized.

Continuous QEC [13], as opposed to discrete QEC, considers a situation where the recovery operation is applied continuously in time. Continuous QEC was first explored by Ahn et al. in a measurement-based feedback strategy [13] (see also [14] for a more recent contribution in this regard). In this article, several continuous correction schemes based on existing QEC

codes are presented. The error syndromes are continuously monitored through weak measurements, and the corresponding correction is achieved by implementing a time-dependent feedback Hamiltonian. This Hamiltonian, based on the measurement records, continuously steers the system back to the coding subspace. Let us define $\mathbb{R}_t$ the evolution operation on time duration $t$, resulting from the dynamics of the correction procedure only, while excluding the decoherence channels resulting in $\mathbb{E}_t$. Note that, $\mathbb{R}_t$ represents a recovery operation only for large enough times $t > T_{\text{corr}}$ (larger e.g. than the error syndrome measurement time). The evolution operator $\mathbb{F}_T$ of the continuous QEC scheme, can intuitively be thought as the limit $\mathbb{F}_T = \lim_n (\mathbb{R}_{T/n} \circ \mathbb{E}_{T/n})^n$.

### 1.2.2 Continuous autonomous QEC via reservoir engineering

Reservoir engineering consists of carefully coupling the system we wish to control/manipulate, with a dissipative reservoir. The idea is to transfer the entropy introduced by errors in the system of interest, onto an ancillary system (reservoir). This entropy is next evacuated via the strong dissipation of the ancilla. Several experiments based on this method have led to the continuous stabilization of specific quantum states, in circuit quantum electrodynamics [15–18].

In [16], Geerlings et al. demonstrated the continuous stabilization of the ground state of a transmon qubit. In this experiment, a transmon qubit is dispersively coupled with a lossy driven resonator, via a Hamiltonian of the form $-\hbar\chi\boldsymbol{\sigma}_Z \boldsymbol{a}^\dagger \boldsymbol{a}/2$. Here, $\boldsymbol{\sigma}_Z$ and $\boldsymbol{a}$ denote the $Z-$Pauli matrix of the qubit and the annihilation operator of the resonator mode. The transmon spontaneously jumps to the excited state $|e\rangle$ at a rate $\gamma_\uparrow$, while the cavity decay rate $\kappa_c$ is taken to be much larger than $\gamma_\uparrow$. Through the dispersive coupling, the frequency of the resonator depends on the qubit state, and the qubit frequency depends on the number of photons in the resonator. In this protocol, one applies a drive at frequency $\omega_c^g$, where $\omega_c^g$ is the frequency of the cavity when the qubit is in the ground state $|g\rangle$. If the transmon is in $|g\rangle$, the resonator evolves towards a coherent state $|\alpha\rangle$ in a time of order $1/\kappa_c$, where $\alpha$ is given by the ratio between the drive amplitude and the cavity rate $\kappa_c$. In this case, the state of the global system is $|g\rangle \otimes |\alpha\rangle$. If the transmon is in the excited state $|e\rangle$, the drive is off-resonant, and the cavity evolves to the vacuum state $|0\rangle$ in a mean time $1/\kappa_c$, leading to the global system state $|e\rangle \otimes |0\rangle$. The state of the transmon is thus imprinted on the state of the resonator. In other words, the cavity realizes a measurement of the qubit state, with the pointer states $|0\rangle$ and $|\alpha\rangle$. Indeed, one could access to the measurement output by looking at the amplitude of the transmitted cavity field, although it is not required by this scheme. Instead, one can regularly apply a fast $\pi$-pulse at frequency $\omega_{ge}^0$, where $\omega_{ge}^0$ is the qubit frequency when the cavity is in the vacuum state. More precisely, after a time larger than $1/\kappa_c$, and before the application of the $\pi$-pulse, the state of the total system is given by $\boldsymbol{\rho}_{SA} = (1-p)|g\rangle\langle g| \otimes |\alpha\rangle\langle\alpha| + p|e\rangle\langle e| \otimes |0\rangle\langle 0|$, with $0 \leq p \leq 1$. The conditional $\pi$-pulse maps $\boldsymbol{\rho}_{SA}$ to the state $|g\rangle \otimes ((1-p)|\alpha\rangle\langle\alpha| + p|0\rangle\langle 0|)$ and next the continuous drive resets the resonator to the state $|\alpha\rangle$ (entropy evacuation). In [16], this reservoir engineering scheme is implemented in a continuous manner, by using a continuous Rabi drive at frequency $\omega_{ge}^0$ instead of $\pi$-pulses. As the cavity drive pumps the population on $|g\rangle \otimes |0\rangle$ out to the state $|g\rangle \otimes |\alpha\rangle$ at a rate $\kappa_c$, the system is rapidly projected to the steady state $|g\rangle \otimes |\alpha\rangle$. Hence, the entropy introduced by the spontaneous excitations of the transmon at a rate $\gamma_\uparrow$ is evacuated via the resonator at a rate of order $\kappa_c \gg \gamma_\uparrow$.

Inspired by this protocol, Shankar et al. demonstrated the autonomous stabilization of an entangled Bell state by dispersively coupling two transmon qubits to a lossy cavity [17].

Similarly, reservoir engineering QEC schemes use the coupling to an ancillary quantum system to mediate the evacuation of the information entropy created by errors. From Subsection 1.1.3, one recalls that the recovery operation involves the use of an ancillary system. More precisely, the effect of a recovery operation is expressed through a unitary operator $U_{SA}$

on the system S and an ancillary system A, such that

$$U_{SA}\left[\sum_\nu (\boldsymbol{E}_\nu |\psi\rangle_S) \otimes |a_{\mu_0}\rangle_A \otimes |e_\nu\rangle_E\right] = |\psi_S\rangle \otimes \left(\sum_\mu \sqrt{d_\mu}|a_\mu\rangle_A \otimes |e_\mu\rangle_E\right).$$

Here, the increase of entropy on the system S is expressed through the entangled state $\sum_\nu(\boldsymbol{E}_\nu|\psi\rangle_S) \otimes |e_\nu\rangle_E$ between S and the environment E. By applying the unitary operation $\boldsymbol{U}_{SA}$, we have transferred the entropy onto the ancilla, resulting in the creation of the entangled state $\sum_\mu \sqrt{d_\mu}|a_\mu\rangle_A \otimes |e_\mu\rangle_E$. The strong dissipation of the ancilla naturally evacuates the entropy by resetting its state to $|a_{\mu_0}\rangle_A$.

So far, the recovery procedure is not continuous. The discrete operation $\boldsymbol{U}_{SA}$ corresponds to the conditional $\pi$-pulse applied in the above example. The resonator plays the role of the ancilla, and the ancilla state $|a_{\mu_0}\rangle_A$ is the coherent state $|\alpha\rangle$. The operation $\boldsymbol{U}_{SA}$ and the decay of the ancilla, can be realized in a simultaneous manner as illustrated through the example of [16].

We would like to stress the fact that in a reservoir engineering QEC scheme, error detection and correction are not two distinct steps. A few proposals of such protocols can be found in the literature. In [19], Kerckhoff et al. proposed to implement the three-qubit bit-flip (and phase-flip) code in a photonic circuit through an autonomous feedback loop embedded in the system. More precisely, error syndromes are collected through optical beams interacting with the qubits, and then conveyed to two quantum controllers (ancillas) via directional couplings. The beams, combined with the dissipation of the ancillas, drive the controllers to steady states which depend on the error syndromes. Two additional "feedback" beams interact with the controllers, and are injected into the qubit system to drive it back to the code space. Kerckhoff et al. have also presented an extension of this work to the implementation of the 9-qubit Bacon-Shor code [20]. An autonomous QEC scheme based on three-qubit phase-flip code was also proposed in the field of circuit QED by Kapit et al. [21]. In [21], such a scheme is realized by coupling three transmon qubits to three dissipative ancillary qubits. The transmon qubits are two-by-two coupled through well-chosen magnetic fluxes. When the system stepped out of the code space through a single phase flip, it is irreversibly brought back through the dissipation of the ancillas. In [22], we presented another autonomous QEC protocol for three-qubit bit-flip (or phase-flip) code with transmon qubits. As illustrated in the PhD dissertation of Joachim Cohen [23], such an autonomous QEC scheme can be adapted to the case of a repetition cat code (such codes are the main topic of the present notes).

## 1.3  Bosonic codes

In the previous sections we re-called some general results on quantum error correcting codes, illustrated by a few multi-qubit codes as examples. Instead of using many qubits to provide the redundancy required to protect the encoded information, one can also encode the information in bosonic modes, such as a single harmonic oscillator, and benefit from the vastness of the associated Hilbert space. Such an idea has been pursued along two different directions. One direction is to mainly focus on the infinite dimensional Hilbert space of such systems and encode information in such a way that some major error channels, such as the photon loss due to energy relaxation, become tractable [24–26]. The question of benefiting from this infinite dimensions in an *optimal* manner then becomes a relevant question [27]. The second direction is to focus on the phase space of the Harmonic oscillator and encode information in a non-local manner in this phase space [10, 28]. Then a protection can be ensured against all types of errors with a local action on the phase space. Such a protection appears to be strong as it involves most physical error mechanisms such as photon-loss, photon dephasing, thermal excitations, or spurious non-linear Hamiltonians with bounded potentials (e.g. those induced by Josephson junctions).

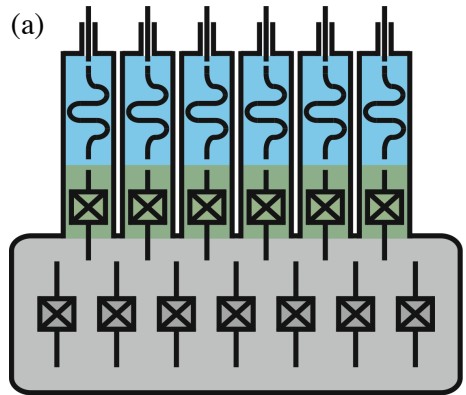
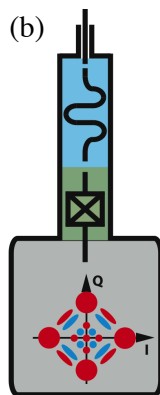

Figure 1: (a) A protected logical qubit consisting of a register of many qubits: here, we see a possible architecture for the Steane code [8] consisting of 7 qubits requiring the measurement of 6 error syndromes. In this sketch, 7 transmon qubits in a high-Q resonator and the measurement of the 6 error syndromes is ensured through 6 additional ancillary qubits with the possibility of individual readout of the ancillary qubits via independent low-Q resonators. (b) Minimal architecture for a protected logical qubit, adapted to circuit quantum electrodynamics experiments. Quantum information is encoded in a Schrödinger cat state of a single high-Q resonator mode and a single error syndrome is measured, using a single ancillary transmon qubit and the associated readout low-Q resonator.

In the literature, the cat codes have been exploited in both above directions. While these notes have for goal to study the cat-codes uniquely from the second perspective, here we provide a brief overview of both approaches.

### 1.3.1 Exploiting infinite dimensional Hilbert space

The infinite dimensional Hilbert space of a quantum harmonic oscillator (e.g. a high-Q mode of a 3D superconducting cavity) can be used to redundantly encode quantum information. The power of this idea lies in the fact that the dominant decoherence channel in a cavity is photon damping, and no extra decay channels are added if we increase the number of photons we insert in the cavity. Hence, only a single error syndrome needs to be measured to identify if an error has occurred or not.

One such scheme was proposed in [25], where the logical qubit is encoded in a four-component Schrödinger cat state. Repeated quantum non-demolition (QND) monitoring of a single physical observable, consisting of photon number parity, ensures then the tractability of single photon jumps. We obtain therefore a first-order quantum error correcting code using only a single high-Q cavity mode (for the storage of quantum information), a single qubit (providing the non-linearity needed for controllability) and a single low-Q cavity mode (for reading out the error syndrome). As sketched in Figure 1, this leads to a significant hardware economy for realization of a protected logical qubit.

The idea consists in mapping the qubit state $c_0|0\rangle + c_1|1\rangle$ into a superposition of four coherent states of a quantum harmonic oscillator $|\psi_\alpha^{(0)}\rangle = c_0|0\rangle_L + c_1|1\rangle_L = c_0|\mathcal{C}_\alpha^{(0\mathrm{mod}4)}\rangle + c_1|\mathcal{C}_\alpha^{(2\mathrm{mod}4)}\rangle$, where

$$|\mathcal{C}_\alpha^{(0\mathrm{mod}4)}\rangle = \mathcal{N}_0(|\alpha\rangle + |-\alpha\rangle + |i\alpha\rangle + |-i\alpha\rangle),$$
$$|\mathcal{C}_\alpha^{(1\mathrm{mod}4)}\rangle = \mathcal{N}_2(|\alpha\rangle - |-\alpha\rangle - i|i\alpha\rangle + i|-i\alpha\rangle),$$

$$|\mathcal{C}_\alpha^{(2\mathrm{mod}4)}\rangle = \mathcal{N}_1(|\alpha\rangle + |-\alpha\rangle - |i\alpha\rangle - |-i\alpha\rangle),$$
$$|\mathcal{C}_\alpha^{(3\mathrm{mod}4)}\rangle = \mathcal{N}_3(|\alpha\rangle - |-\alpha\rangle + i|i\alpha\rangle - i|-i\alpha\rangle).$$

Here, $\mathcal{N}_0 \approx \mathcal{N}_1 \approx \mathcal{N}_2 \approx \mathcal{N}_3 \approx 1/2$ are normalization factors, and $|\alpha\rangle$ denotes a coherent state of complex amplitude $\alpha$. For $\alpha$ large enough, $|\alpha\rangle$, $|-\alpha\rangle$, $|i\alpha\rangle$ and $|-i\alpha\rangle$ are quasi-orthogonal (note that for $\alpha = 2$, $|\langle\alpha|i\alpha\rangle|^2 < 10^{-3}$) and therefore the normalization constants $\mathcal{N}_k$ are well-approximated by $1/2$. The four-component cat state $|\mathcal{C}_\alpha^{(j\mathrm{mod}4)}\rangle$ is a linear superposition of Fock states $|4n + j\rangle$, i.e Fock states with photon numbers $j$ mod 4. As a consequence, these states form an orthonormal basis of the 4D-manifold $\mathcal{M}_{4,\alpha} = \mathrm{span}\{|\alpha\rangle, |-\alpha\rangle, |i\alpha\rangle, |-i\alpha\rangle\}$. This manifold is the direct sum of the even-parity subspace $\mathcal{E}_+ = \mathrm{span}\{|\mathcal{C}_\alpha^{(0\mathrm{mod}4)}\rangle, |\mathcal{C}_\alpha^{(2\mathrm{mod}4)}\rangle\}$ and the odd-parity subspace $\mathcal{E}_- = \mathrm{span}\{|\mathcal{C}_\alpha^{(1\mathrm{mod}4)}\rangle, |\mathcal{C}_\alpha^{(3\mathrm{mod}4)}\rangle\}$. In [25], a logical qubit is encoded on the even-parity subspace, such that

$$|0_L\rangle = |\mathcal{C}_\alpha^{(0\mathrm{mod}4)}\rangle, \quad |1_L\rangle = |\mathcal{C}_\alpha^{(2\mathrm{mod}4)}\rangle.$$

We also define the logical operators

$$\sigma_Z^{\mathrm{even}} = |\mathcal{C}_\alpha^{(0\mathrm{mod}4)}\rangle\langle\mathcal{C}_\alpha^{(0\mathrm{mod}4)}| - |\mathcal{C}_\alpha^{(2\mathrm{mod}4)}\rangle\langle\mathcal{C}_\alpha^{(2\mathrm{mod}4)}|,$$
$$\sigma_X^{\mathrm{even}} = |\mathcal{C}_\alpha^{(0\mathrm{mod}4)}\rangle\langle\mathcal{C}_\alpha^{(2\mathrm{mod}4)}| + |\mathcal{C}_\alpha^{(2\mathrm{mod}4)}\rangle\langle\mathcal{C}_\alpha^{(0\mathrm{mod}4)}|,$$
$$\sigma_Z^{\mathrm{odd}} = |\mathcal{C}_\alpha^{(3\mathrm{mod}4)}\rangle\langle\mathcal{C}_\alpha^{(3\mathrm{mod}4)}| - |\mathcal{C}_\alpha^{(1\mathrm{mod}4)}\rangle\langle\mathcal{C}_\alpha^{(1\mathrm{mod}4)}|,$$
$$\sigma_X^{\mathrm{odd}} = |\mathcal{C}_\alpha^{(3\mathrm{mod}4)}\rangle\langle\mathcal{C}_\alpha^{(0\mathrm{mod}4)}| + |\mathcal{C}_\alpha^{(1\mathrm{mod}4)}\rangle\langle\mathcal{C}_\alpha^{(2\mathrm{mod}4)}|.$$

This encoding enables us to protect the quantum information against photon loss events [25]. In order to see this, let us also define $|\psi_\alpha^{(1)}\rangle = c_0|\mathcal{C}_\alpha^{(3\mathrm{mod}4)}\rangle + c_1|\mathcal{C}_\alpha^{(1\mathrm{mod}4)}\rangle$, $|\psi_\alpha^{(2)}\rangle = c_0|\mathcal{C}_\alpha^{(2\mathrm{mod}4)}\rangle + c_1|\mathcal{C}_\alpha^{(0\mathrm{mod}4)}\rangle$ and $|\psi_\alpha^{(3)}\rangle = c_0|\mathcal{C}_\alpha^{(1\mathrm{mod}4)}\rangle + c_1|\mathcal{C}_\alpha^{(3\mathrm{mod}4)}\rangle$. The state $|\psi_\alpha^{(n)}\rangle$ evolves after a photon loss event to $a|\psi_\alpha^{(n)}\rangle/\|a|\psi_\alpha^{(n)}\rangle\| = |\psi_\alpha^{[(n-1)\mathrm{mod}4]}\rangle$, where $a$ is the harmonic oscillator's annihilation operator. Furthermore, in the absence of jumps during a time interval $t$, $|\psi_\alpha^{(n)}\rangle$ deterministically evolves to $|\psi_{\alpha e^{-\kappa t/2}}^{(n)}\rangle$, where $\kappa$ is the decay rate of the harmonic oscillator. Now, the photon number parity operator $\Pi = \exp(i\pi a^\dagger a)$ can act as a photon jump indicator. Indeed, we have $\langle\psi_\alpha^{(n)} | \Pi | \psi_\alpha^{(n)}\rangle = (-1)^n$ and therefore measuring a change in the photon number parity indicates the occurrence of a single photon loss event.

While the parity measurements keep track of the photon loss events, the deterministic relaxation of the energy, replacing $\alpha$ by $\alpha e^{-\kappa t/2}$, remains inevitable. To overcome this relaxation of energy, we need to intervene before the coherent states start to overlap in a significant manner to re-pump energy into the codeword. This energy repumping in the cat state requires a non-linear interaction with the cavity mode. In [25], it was proposed that such an energy re-pumping can be performed through the application of a sequence of well-chosen pulses on a qubit-cavity system. Indeed, as proven in [29], the strong dispersive coupling of a quantum harmonic oscillator to a qubit, together with frequency-resolved microwave drives, provide the means towards the universal controllability of the state of the harmonic oscillator Furthermore, as shown recently in [30] this universal controllability can be extended to the case of weak dispersive couplings as well.

Furthermore, the quantum non-demolition (QND) measurements of the parity observable can also be performed by coupling the harmonic oscillator to a qubit in the dispersive regime and exploiting the same controllability. Such parity measurements using a superconducting qubit was experimentally performed in [31]. This experiment, later, led to the first implementation of a QEC experiment at the break-even point [32]. Here, the break-even means that the encoded quantum information is protected over a longer time than the best physical component of the system.

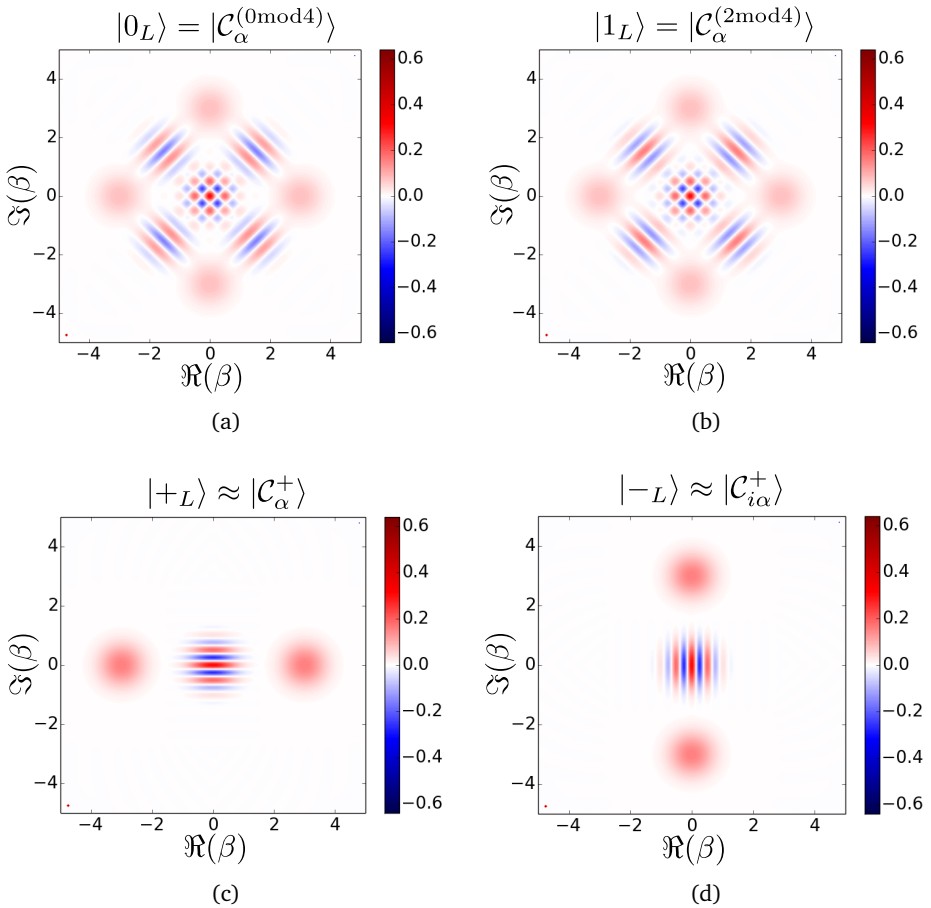

Figure 2: (a) and (b): Wigner representation of the logical states $|0_L\rangle = |\mathcal{C}_\alpha^{(0\bmod4)}\rangle$ and $|1_L\rangle = |\mathcal{C}_\alpha^{(2\bmod4)}\rangle$ of the four-component cat code, with $\alpha = 3$. (c) and (d): Wigner representation of the logical states $|+_L\rangle = (|0_L\rangle + |1_L\rangle)/\sqrt{2} \approx |\mathcal{C}_\alpha^+\rangle$ and $|-_L\rangle = (|0_L\rangle - |1_L\rangle)/\sqrt{2} \approx |\mathcal{C}_{i\alpha}^+\rangle$.

Such a bosonic code based on Schrödinger cat states can be extended to higher order errors. Indeed, in order to implement an $N$'th order correcting scheme, one can consider superposing $2(N+1)$ quasi-orthogonal coherent states for logical states. As an example a 2nd order coding can be obtained in the following manner

$$|0_L\rangle = |\mathcal{C}_\alpha^{(0\bmod6)}\rangle = \sum_{r=0}^{5} |\alpha e^{ir\pi/3}\rangle, \qquad |1_L\rangle = |\mathcal{C}_\alpha^{(3\bmod6)}\rangle = \sum_{r=0}^{5} e^{ir\pi} |\alpha e^{ir\pi/3}\rangle.$$

By continuously monitoring the photon number modulo 3, we can track photon jumps up to two in a measurement time. Similarly to the case of the four-component cat, we need to re-pump energy in the cat state before the coherent states start to significantly overlap because of the energy damping. Indeed, in order to ensure a small overlap between the two neighbouring coherent states, we need to start with cat states with larger amplitude $|\alpha|$ than the case of the four-component cat.

Later, a more economic use of the Hilbert space of the harmonic oscillator was considered in the proposal for binomial codes [26]. In order to develop a generalized code which protects the information against the set of errors $\{I, a, a^2, \ldots, a^L, a^\dagger, \ldots, (a^\dagger)^G, n, \ldots, n^D\}$, the information

is encoded in two states

$$|W_{\uparrow/\downarrow}\rangle = \frac{1}{2^N} \sum_{\substack{p \text{ even/odd}}}^{[0 \ N+1]} \sqrt{\binom{N+1}{p}} |p(S+1)\rangle,$$

where $S = L + G$ is the spacing, $N = \max(L, G, 2D)$ is the the maximum order, and $p$ ranges from 0 to $N+1$. In the case of $N \to \infty$ the binomal code states, asymptotically approach to the $2(S+1)$ component cat code. Here, the error syndromes are more complex than simply photon number parity (or photon number modulo $N$ in general). However, using the universal control provided by the dispersive coupling to a qubit, it is possible to track and correct the associated errors. Such a code at first order and against photon loss has been recently experimentally implemented in [33] approaching the break-even point. In this case, the associated logical states are

$$|0_L\rangle = \frac{1}{\sqrt{2}}(|0\rangle + |4\rangle), \qquad |1_L\rangle = |2\rangle,$$

and the error syndrome corresponds to the photon number parity.

In the next subsection, we will see how the infinite dimensional Hilbert space of a harmonic oscillator can be exploited in a different manner, paving the way towards hardware-efficient implementations of a fault-tolerant quantum processor.

### 1.3.2 Exploiting non-locality in phase space

Quite early in the process of the theoretical proposals on quantum error correction, Gottesman, Kitaev and Preskill came up with an ingenious idea to encode a qubit in a harmonic oscillator [28]. The idea consisted in encoding the information in the so-called grid states of the harmonic oscillator:

$$|0_L\rangle = \sum_{r=-\infty}^{\infty} |q = 2r\sqrt{\pi}\rangle, \qquad |1_L\rangle = \sum_{r=-\infty}^{\infty} |q = (2r+1)\sqrt{\pi}\rangle. \tag{3}$$

Here, $q = (a + a^\dagger)/\sqrt{2}$ is the position operator and the state $|q = q\rangle$ corresponds to the position state which is infinitely squeezed along the $q$-axis. Very importantly, such states can also be written in the following form

$$|0_L\rangle = \sum_{r=-\infty}^{\infty} \left(|p = 2r\sqrt{\pi}\rangle + |p = (2r+1)\sqrt{\pi}\rangle\right), \quad |1_L\rangle = \sum_{r=-\infty}^{\infty} \left(|p = 2r\sqrt{\pi}\rangle - |p = (2r+1)\sqrt{\pi}\rangle\right),$$

where $p = (a - a^\dagger)/i\sqrt{2}$ is the momentum operator. Therefore, the states along the logical $X$-axis ($|\pm_L\rangle = (|0_L\rangle \pm |1_L\rangle)$) are given by

$$|+_L\rangle = \sum_{r=-\infty}^{\infty} |p = 2r\sqrt{\pi}\rangle, \qquad |-_L\rangle = \sum_{r=-\infty}^{\infty} |p = (2r+1)\sqrt{\pi}\rangle.$$

One notes that the two states $|0_L\rangle$ and $|1_L\rangle$ have a disjoint support in the phase space. The state $|1_L\rangle$ is achieved from $|0_L\rangle$ by shifting the position by value $\sqrt{\pi}$. In the same manner the two states $|+_L\rangle$ and $|-_L\rangle$ have also a disjoint support in the phase space and $|+_L\rangle$ is achieved from $|-_L\rangle$ by shifting the momentum operator by the same value $\sqrt{\pi}$. It is therefore possible to protect such a qubit against bit-flips and phase-flips if the shifts in the phase space occur slowly enough. Indeed, by measuring both quadratures $q$ and $p$ modulo $\sqrt{\pi}$, it is possible to correct for shifts which are not larger than $\sqrt{\pi}/2$.

One important detail is that, in the above definitions, we have intentionally avoided to talk about any state normalization. Indeed, the above logical states are not physical as they correspond to states with infinite energy. In practice, one can approach such states by considering a

superposition of finitely squeezed states of the $q$ quadrature. More precisely, the logical states are defined as

$$|0_L\rangle = \mathcal{N}_0 \sum_{r=-\infty}^{\infty} e^{-\frac{\delta^2(2r\sqrt{\pi})^2}{2}} \int_{-\infty}^{+\infty} dq\, e^{-\frac{(q-2r\sqrt{\pi})^2}{2\delta^2}} |q = q\rangle,$$

$$|1_L\rangle = \mathcal{N}_1 \sum_{r=-\infty}^{\infty} e^{-\frac{\delta^2((2r+1)\sqrt{\pi})^2}{2}} \int_{-\infty}^{+\infty} dq\, e^{-\frac{(q-(2r+1)\sqrt{\pi})^2}{2\delta^2}} |q = q\rangle.$$

Note that these states are not eigenstates of the observables associated to position (or momentum) modulo $\sqrt{\pi}$ and their protection comes with extra complications. It is however possible to protect them as far as the squeezing level (characterized by $\delta$) is high enough.

The above qualitative description of the protection against local errors can be made more precise. Let us assume that a particular error mechanism leads to a diffusion of the state of harmonic oscillator in its phase space. Let us also assume the measurement time to be given by $\tau_M$. In the case of ideal grid states and perfect measurement, the effective error probability after correction is given by the probability that during the time $\tau_M$, the state has diffused along the $q$ or $p$ quadratures on a distance that is longer than $\sqrt{\pi}/2$. Assuming $\tau_M$ to be short, one can hope that such a probability is very small. In order to achieve even lower error probabilities however, one needs to further concatenate this bosonic code with another multi-qubit code. The references [34–36] consider such a concatenation with e.g. toric/surface codes or 3D color codes. Note furthermore that in practice, the physical errors lead to diffusions in the phase space where the diffusion rate depends on the energy (faster diffusion for high photon numbers). Therefore, in order to obtain an optimal effective error probability after error correction with the grid states, one should not infinitely increase the level of squeezing towards ideal GKP states. Despite these complications, the first level of protection provided by the grid states encoding can lead to a significant reduction of hardware overhead for quantum error correction.

One might ask if is possible to achieve a better protection by making the encoding states even more non-local? What happens if the computational states $|0_L\rangle$ and $|1_L\rangle$ are even further apart in the phase space? While this is indeed possible and make the bit-flip errors less probable, it unfortunately also comes at the expense of closer states $|\pm_L\rangle$ in the dual basis and therefore higher phase-flip probability. Through these notes, we will however see that such an asymmetric situation can still be very useful and can lead to significant hardware shortcuts for quantum computation. Indeed, through these notes, we will consider an encoding where the non-locality in the phase space is only employed to efficiently suppress the bit-flip errors. The phase-flip errors are then handled differently.

Indeed, while it is possible to design a grid state with such asymmetric property, a better option is to consider the cat encoding once again. However, instead of a four-component cat, we will simply focus on a simpler two-component one. Our choice of encoding is as follows (see also Figs. 3 and 4). We define the cat states

$$|\mathcal{C}_\alpha^\pm\rangle = \mathcal{N}_\pm(|\alpha\rangle \pm |\pm\alpha\rangle), \qquad \mathcal{N}_\pm = \frac{1}{\sqrt{2(1 \pm e^{-2|\alpha|^2})}}.$$

The cat state $|\mathcal{C}_\alpha^+\rangle$ is a superposition of only even Fock states, while $|\mathcal{C}_\alpha^-\rangle$ is a superposition of the odd ones. We define the cat qubit states to be

$$|0\rangle_c = \frac{1}{\sqrt{2}}(|\mathcal{C}_\alpha^+\rangle + |\mathcal{C}_\alpha^-\rangle) = |\alpha\rangle + \mathcal{O}(\exp(-2|\alpha|^2)),$$

$$|1\rangle_c = \frac{1}{\sqrt{2}}(|\mathcal{C}_\alpha^+\rangle - |\mathcal{C}_\alpha^-\rangle) = |-\alpha\rangle + \mathcal{O}(\exp(-2|\alpha|^2)).$$

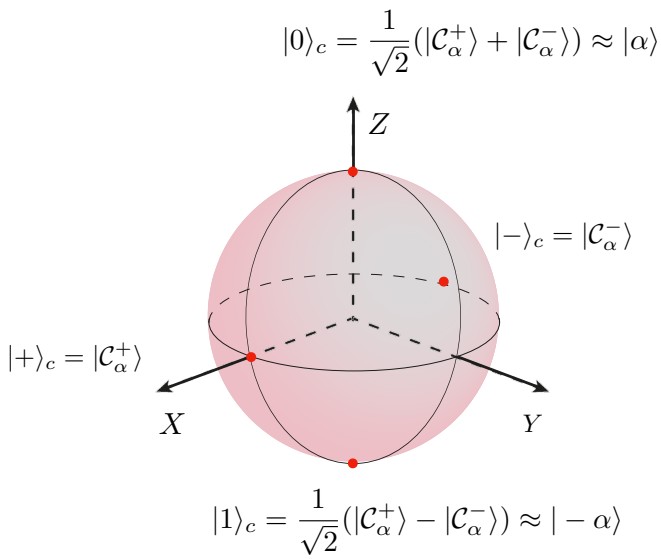

Figure 3: Bloch sphere representation of a cat qubit. Reproduced with permission from Ref. [9].

The non-locality in the phase space can be tuned by the amplitude $|\alpha|$ of the cat states. As it will be seen in these notes, through an appropriate protection mechanism the bit-flip rate can be exponentially suppressed with $|\alpha|^2$. This happens while the phase-flip rate only increases linearly in $|\alpha|^2$. This favorable scaling leads to strong reduction of hardware overhead requirements for fault-tolerant quantum computation. One very important feature of such cat qubits is that their protection against bit-flips can be performed through an autonomous error correction mechanism which is within the reach of experiments in superconducting circuits. This autonomous QEC scheme is based on a two-photon driven dissipative process. We will explain through the next chapter how such a process can be engineered and how it leads to protection against bit-flips.

## 2 Two-photon driven dissipation and bit-flip suppression

A damped classical harmonic oscillator which is driven periodically converges asymptotically to a steady periodic solution with the same frequency as the drive. Indeed, considering the equations of a driven damped harmonic oscillator

$$\frac{d}{dt}a = -i\omega_a a - \frac{\kappa}{2}a - i\epsilon_d e^{-i(\omega_d t + \phi_d)},$$

where $\omega_a$ is the frequency of the harmonic oscillator, $\kappa$ its damping rate, $\omega_d$, $\epsilon_d$ and $\phi_d$, respectively, the frequency, amplitude and the phase of the drive, the system converges to the steady state

$$a_\infty(t) = \bar{a}e^{-i(\omega_d t + \bar{\phi})}, \quad \text{with} \quad \bar{a}e^{-i\bar{\phi}} = \frac{-i\epsilon_d e^{-i\phi_d}}{i(\omega_a - \omega_d) + \kappa/2}.$$

Interestingly, a driven damped quantum harmonic oscillator behaves in a similar manner. Under a Markovian approximation (which is valid in an under-damped regime), the Lindblad master equation of a driven damped harmonic oscillator is given by

$$\frac{d}{dt}\rho = -i\omega_a[a^\dagger a, \rho] - i\epsilon_d[e^{-i(\omega_d t + \phi_d)}a^\dagger + e^{i(\omega_d t + \phi_d)}a, \rho] + \kappa\mathcal{D}[a]\rho,$$

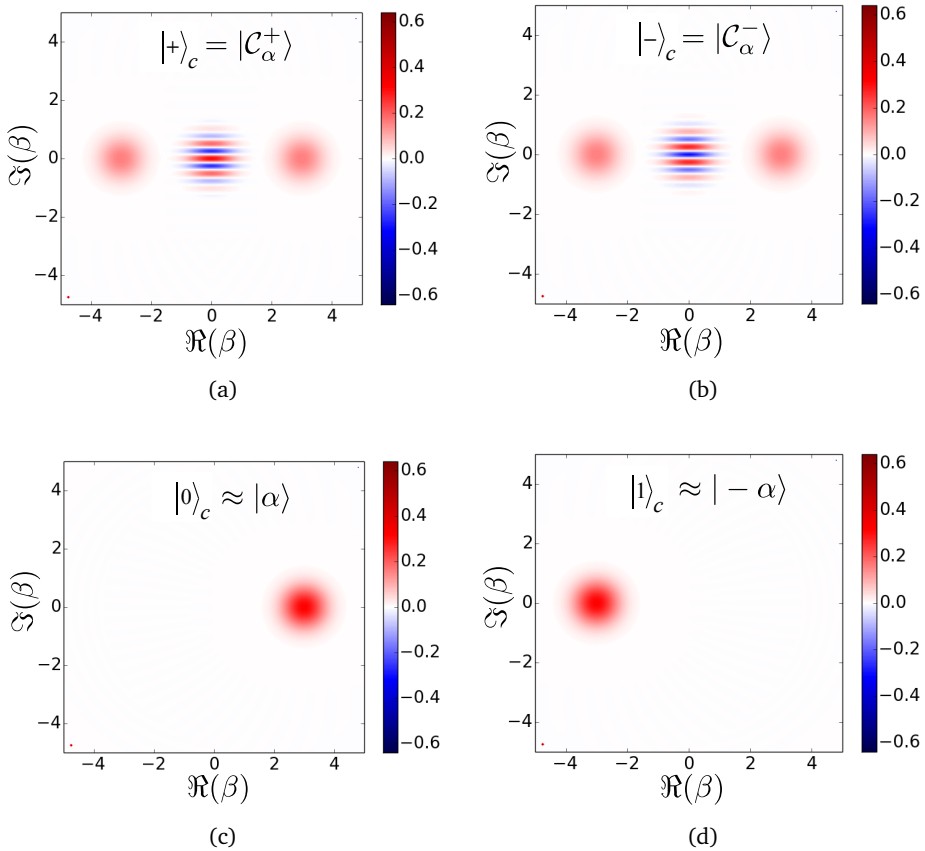

Figure 4: (a) and (b): Wigner representation of the logical states $|+\rangle_c = |\mathcal{C}_\alpha^+\rangle$ and $|-\rangle_c = |\mathcal{C}_\alpha^-\rangle$ of the two-component cat code, with $\alpha = 3$. The color of the fringe at the center (red or blue), indicates the parity of the cat state (resp. even and odd). (c) and (d): Wigner representation of the computational states $|0\rangle_c = (|+\rangle_c + |-\rangle_c)/\sqrt{2} \approx |\alpha\rangle$ and $|1\rangle_c = (|+\rangle_c - |-\rangle_c)/\sqrt{2} \approx |-\alpha\rangle$.

where

$$\mathcal{D}[L]\rho = L\rho L^\dagger - \frac{1}{2}L^\dagger L\rho - \frac{1}{2}\rho L^\dagger L.$$

This master equation admits as the steady state a periodic solution given by a pure coherent state

$$\rho_\infty(t) = |\alpha_\infty(t)\rangle\langle\alpha_\infty(t)|, \quad \text{with} \quad \alpha_\infty(t) = \bar{\alpha}e^{-i(\omega_d t + \bar{\phi})}, \quad \bar{\alpha}e^{-i\bar{\phi}} = \frac{-i\epsilon_d e^{-i\phi_d}}{i(\omega_a - \omega_d) + \kappa/2}.$$

One way of seeing this is to perform the following unitary change of variables (rotating frame of the drive and an appropriate mode displacement)

$$\tilde{\rho} = U(t)\rho U(t)^\dagger, \quad \text{with} \quad U(t) = D(-\bar{\alpha}e^{-i\bar{\phi}})e^{i\omega_d t a^\dagger a} \quad \text{and} \quad D(\alpha) = e^{\alpha a^\dagger - \alpha^* a}.$$

The master equation satisfied by $\tilde{\rho}$

$$\frac{d}{dt}\tilde{\rho} = -i(\omega_a - \omega_d)[a^\dagger a, \tilde{\rho}] + \kappa\mathcal{D}[a]\tilde{\rho},$$

is that of an undriven damped harmonic oscillator. The asymptotic convergence of the undriven damped harmonic oscillator to the vacuum state implies the convergence of the driven

one to the above periodic solution consisting of a pure coherent state. This convergence to a pure state is a remarkable fact as usually the steady state of a dissipative quantum system is a mixed one.

It is reasonable to ask if it is possible to find other such driven damped systems with pure steady states, possibly of more interesting character. The answer is yes! At least at a theoretical level, one such system can be built of a harmonic oscillator with a multi-photon drive and dissipation. Let us consider a Lindblad master equation of the form

$$\frac{d}{dt}\boldsymbol{\rho} = -i\Delta_d[\boldsymbol{a}^\dagger\boldsymbol{a},\boldsymbol{\rho}] - i\epsilon_{r-}[e^{-i\phi_d}\boldsymbol{a}^{\dagger r} + e^{i\phi_d}\boldsymbol{a}^r, \boldsymbol{\rho}] + \kappa_{r-}\mathcal{D}[\boldsymbol{a}^r]\boldsymbol{\rho}\,.$$

written in the rotating frame of the $r$-photon drive. This master equation represents the dynamics of a harmonic oscillator which exchanges photons in multiples of $r$ with its environment. Indeed, while the first term in the dynamics represents a simple detuned Hamiltonian evolution of a harmonic oscillator (the dynamics is in the rotating frame of the drive), the second term indicates a nonlinear driving Hamiltonian where the exchange of photons with the harmonic oscillator occurs at multiples of $r$. Also, the final Lindbladian term models a damping where the harmonic oscillator loses photons in multiples of $r$.

We will postpone the question of how such a non-standard dynamics can be achieved in practice to the end of this section, and we will start by discussing the interest of such a non-linear driven dissipative system. We will start by analysing the asymptotic behaviour of the above master equation. We will see how this leads to a natural choice of physical qubits with "simple" error mechanisms.

## 2.1 Multi-photon driven dissipative processes: asymptotic behaviour

Let us consider the above $r$-photon driven dissipative process and assume, for simplicity sakes, that the detuning term vanishes $\Delta_d = 0$. We are therefore interested in the following dynamics

$$\frac{d}{dt}\boldsymbol{\rho} = -i\epsilon_{r-}[e^{-i\phi_d}\boldsymbol{a}^{\dagger r} + e^{i\phi_d}\boldsymbol{a}^r, \boldsymbol{\rho}] + \kappa_{r-}\mathcal{D}[\boldsymbol{a}^r]\boldsymbol{\rho}\,.$$

It is easy to see that the righthand side can be regrouped in the following form

$$\frac{d}{dt}\boldsymbol{\rho} = \kappa_{r-}\mathcal{D}[\boldsymbol{a}^r - \alpha^r]\boldsymbol{\rho}\,, \qquad \alpha = e^{-i\frac{\phi_d}{r}}\sqrt{-\frac{2i\epsilon_{r-}}{\kappa_{r-}}}\,. \tag{4}$$

Any state in the kernel of the dissipation operator $\boldsymbol{a}^r - \alpha^r$ is necessarily a fixed state of the above dynamics. It is easy to see that all the coherent states

$$\bar{\boldsymbol{\rho}}_k = |\alpha_k\rangle\langle\alpha_k|\,, \qquad \alpha_k = e^{i\frac{2k\pi}{r}}\alpha = e^{i\frac{2k\pi}{r}}e^{-i\frac{\phi_d}{r}}\sqrt{-\frac{2i\epsilon_{r-}}{\kappa_{r-}}}\,, \quad k = 0,\cdots,r-1\,,$$

satisfy this property

$$(\boldsymbol{a}^r - \alpha^r)|\alpha_k\rangle = 0\,,$$

and are the fixed points of the dynamics. More importantly any superposition $\sum_k c_k|\alpha_k\rangle$ of these coherent states is also in the kernel of the dissipation operator and therefore a fixed point of the dynamics. Indeed, one can show that the manifold of the fixed points is given by the density operators in the $r$-dimensional Hilbert space

$$\mathcal{H}_r = \text{Span}\{|\alpha_k\rangle\}_{k=0}^{r-1}\,.$$

Initializing the system in any state out of this manifold, it will end up converging to a state defined in this manifold. One important question is how to characterize the asymptotic solution

for a given initial state. The answer is easy at least for one particular initial state. If we initilize the system in vacuum and let it evolve according to the above dynamics, it will asymptotically converge to the $r$-component Schrödinger cat state

$$|\mathcal{C}_\alpha^{(0\mathrm{mod}r)}\rangle = \mathcal{N}_\alpha^{(0\mathrm{mod}r)} \sum_{k=0}^{r-1} |\alpha_k\rangle = \mathcal{N}_\alpha^{(0\mathrm{mod}r)} \sum_{k=0}^{r-1} |\alpha e^{i\frac{2k\pi}{r}}\rangle,$$

where

$$\mathcal{N}_\alpha^{(0\mathrm{mod}r)} = \left\| \sum_{k=0}^{r-1} |\alpha e^{i\frac{2k\pi}{r}}\rangle \right\|^{-1}$$

is a normalizing constant close to $1/\sqrt{r}$ for large enough $|\alpha|$. In order to see this, one needs to note that in the above dynamics the number of photons are conserved modulo $r$. Indeed, as any exchange of photons with the drive or with the bath occurs in multiples of $r$, initializing the system in the vacuum state $|0\rangle$, the asymptotic state can only be a superposition of Fock states associated to photon numbers which are multiples of $r$. There is a single state in the Hilbert space $\mathcal{H}_r$ with this property and it is given by

$$|\mathcal{C}_\alpha^{(0\mathrm{mod}r)}\rangle = \mathcal{N}_\alpha^{(0\mathrm{mod}r)} \sum_{k=0}^{r-1} |\alpha e^{i\frac{2k\pi}{r}}\rangle = r\mathcal{N}_\alpha^{(0\mathrm{mod}r)} e^{-|\alpha|^2/2} \sum_{m=0}^{\infty} \frac{\alpha^{mr}}{\sqrt{(mr)!}} |mr\rangle.$$

Finding the steady state for other initializations is a delicate question. In what follows, we will concentrate on the simple case of $r = 2$ which is central for our the definition of cat qubits. In this case, the system converges towards a density matrix defined on the 2-dimensional Hilbert space $\mathrm{Span}\{|\pm\alpha\rangle\}$, with

$$\alpha = e^{-i\phi_d/2} \sqrt{-\frac{2i\epsilon_2}{\kappa_2}}.$$

The two coherent states $|\pm\alpha\rangle$ are not exactly orthogonal, but the cat states

$$|\mathcal{C}_\alpha^\pm\rangle = \mathcal{N}_\pm(|\alpha\rangle \pm |-\alpha\rangle), \qquad \mathcal{N}_\pm = \frac{1}{\sqrt{2(1 \pm e^{-2|\alpha|^2})}},$$

are exactly orthogonal. Indeed, the cat state $|\mathcal{C}_\alpha^+\rangle$ is a superposition of only even Fock states, while $|\mathcal{C}_\alpha^-\rangle$ is a superposition of the odd ones.

All initial states evolving under the two-photon driven dissipative process

$$\frac{d}{dt}\boldsymbol{\rho} = \kappa_{2\mathrm{ph}} \mathcal{D}[\boldsymbol{a}^2 - \alpha^2]\boldsymbol{\rho}, \tag{5}$$

will exponentially converge to a specific (possibly mixed) asymptotic density matrix defined on the Hilbert space spanned by the two-component Schrödinger cat states $\{|\mathcal{C}_\alpha^+\rangle, |\mathcal{C}_\alpha^-\rangle\}$. In order to characterize the Bloch vector of this asymptotic density matrix $\boldsymbol{\rho}_\infty$, it is sufficient to determine three degrees of freedom: the population of one of the cats ($c_{++} = \langle \mathcal{C}_\alpha^+|\boldsymbol{\rho}_\infty|\mathcal{C}_\alpha^+\rangle$) and the complex coherence between the two ($c_{+-} = \langle \mathcal{C}_\alpha^-|\boldsymbol{\rho}_\infty|\mathcal{C}_\alpha^+\rangle$). There exist conserved quantities $\boldsymbol{J}_{++}, \boldsymbol{J}_{+-}$ corresponding to these degrees of freedom [37] such that $c_{++} = \mathrm{Tr}(\boldsymbol{J}_{++}^\dagger \boldsymbol{\rho}(0))$ and $c_{+-} = \mathrm{Tr}(\boldsymbol{J}_{+-}^\dagger \boldsymbol{\rho}(0))$ for any initial state $\boldsymbol{\rho}(0)$. These conserved quantities are given by

$$\boldsymbol{J}_{++} = \sum_{n=0}^{\infty} |2n\rangle\langle 2n|, \tag{6}$$

$$\boldsymbol{J}_{+-} = \sqrt{\frac{2|\alpha|^2}{\sinh(2|\alpha|^2)}} \sum_{q=-\infty}^{\infty} \frac{(-1)^q}{2q+1} I_q(|\alpha|^2) \boldsymbol{J}_{+-}^{(q)} e^{-i\theta_\alpha(2q+1)}, \tag{7}$$

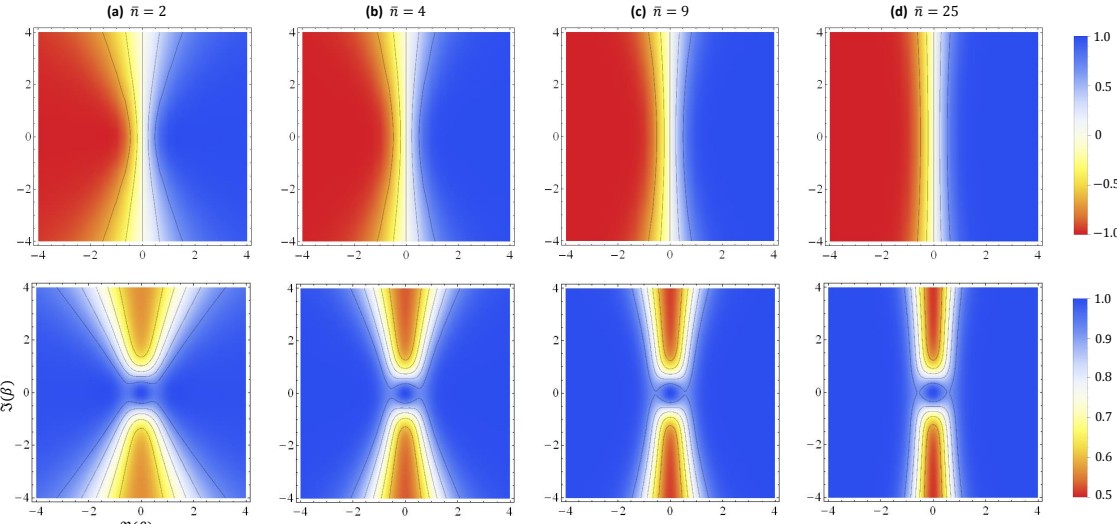

Figure 5: Asymptotic (infinite-time) behavior of the two-photon driven dissipative process given by Eq. (5) where the density matrix is initialized in a coherent state. Here a point $\beta$ in the phase space corresponds to the coherent state $|\beta\rangle$ at which the process is initialized. The upper row illustrates the difference between the population of the two steady coherent states $\{|\pm\alpha\rangle\}$ ($\langle\alpha|\boldsymbol{\rho}|\alpha\rangle-\langle-\alpha|\boldsymbol{\rho}|-\alpha\rangle$ varying between $-1$ and $1$) with $\bar{n}=|\alpha|^2=2,4,9$ and $25$. We observe that for most coherent states except for a narrow vertical region in the center of the phase space, the system converges to one of the steady coherent states $|\pm\alpha\rangle$. The lower row illustrates the purity of the steady state to which we converge $(\mathrm{Tr}(\boldsymbol{\rho}_\infty^2))$ for various initial coherent states. Besides the asymptotic state being the pure $|\pm\alpha\rangle$ away from the vertical axis, one can observe that the asymptotic state is also pure for initial states near the center of phase space. Indeed, starting in the vacuum state, the two-photon process drives the system to the pure Schrödinger cat state $|\mathcal{C}_\alpha^+\rangle$. Reproduced with permission from Ref. [10].

where $I_q(.)$ is the modified Bessel function of the first kind and

$$
\boldsymbol{J}_{+-}^{(q)} = \begin{cases} \dfrac{(\boldsymbol{a}^\dagger\boldsymbol{a}-1)!!}{(\boldsymbol{a}^\dagger\boldsymbol{a}+2q)!!}\boldsymbol{J}_{++}\boldsymbol{a}^{2q+1} & q \geq 0, \\[2ex] \boldsymbol{J}_{++}\boldsymbol{a}^{\dagger 2|q|-1}\dfrac{(\boldsymbol{a}^\dagger\boldsymbol{a})!!}{(\boldsymbol{a}^\dagger\boldsymbol{a}+2|q|-1)!!} & q < 0. \end{cases}
$$

In the above, $n!! = n \times (n-2)!!$ is the double factorial.

**Initializing in a coherent state.** The conserved quantities $\{\boldsymbol{J}_{++}, \boldsymbol{J}_{+-}\}$ are sufficient to calculate the population $c_{++} = \langle\mathcal{C}_\alpha^+|\boldsymbol{\rho}_\infty|\mathcal{C}_\alpha^+\rangle$ and coherence $c_{+-} = \langle\mathcal{C}_\alpha^-|\boldsymbol{\rho}_\infty|\mathcal{C}_\alpha^+\rangle$ of the asymptotic state for any initial state $\boldsymbol{\rho}(0)$. Letting $\boldsymbol{\rho}(0) = |\beta\rangle\langle\beta|$ with $\beta = |\beta|e^{i\theta_\beta}$, the respective terms are

$$
c_{++} = \mathrm{Tr}\left(\boldsymbol{J}_{++}^\dagger\boldsymbol{\rho}(0)\right) = \frac{1}{2}\left(1 + e^{-2|\beta|^2}\right), \tag{8}
$$

$$
c_{+-} = \mathrm{Tr}\left(\boldsymbol{J}_{+-}^\dagger\boldsymbol{\rho}(0)\right) = \frac{i\alpha\beta^\star e^{-|\beta|^2}}{\sqrt{2\sinh(2|\alpha|^2)}}\int_{\phi=0}^\pi d\phi\, e^{-i\phi}I_0\left(|\alpha^2 - \beta^2 e^{2i\phi}|\right). \tag{9}
$$

Assuming real $\alpha$ and using Eq. (5.8.1.15) from [38], one can calculate limits for large $|\beta|^2$

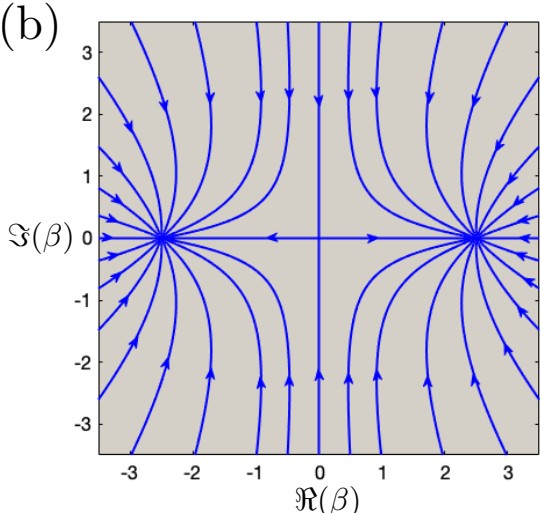

Figure 6: Vector field associated to the semi-classical dynamics behind the master equation (5) represented in the phase space of the harmonic oscillator. This vector field governs the dynamics of coherent states. It admits two stable equilibria $|\pm\alpha\rangle$ and one saddle point at zero. Reproduced with permission from Ref. [9].

along the real and imaginary axes in phase space:

$$\lim_{\beta\to\infty} c_{+-} = \frac{1}{2}\frac{\mathrm{erf}(\sqrt{2}|\alpha|)}{\sqrt{1-e^{-4|\alpha|^2}}} \overset{|\alpha|\to\infty}{\longrightarrow} \frac{1}{2} \qquad \text{and} \qquad \lim_{\beta\to i\infty} c_{+-} = -i\frac{1}{2}\frac{\mathrm{erfi}(\sqrt{2}|\alpha|)}{\sqrt{e^{4|\alpha|^2}-1}} \overset{|\alpha|\to\infty}{\longrightarrow} 0\,,$$

where erf(.) and erfi(.) are the error function and imaginary error function, respectively. Both limits analytically corroborate Fig. 5 and show that the two-photon system is similar to a classical double-well system in the combined large $\alpha, \beta$ regime. This behaviour is also well presented by Figure 6 where the vector field associated to the semi-classical dynamics of a coherent state governed by (5) is plotted in the phase space of the oscillator.

## 2.2 A qubit with biased noise

As stated in the previous subsection, a two-photon driven dissipative harmonic oscillator modelled by (5) admits as steady state the 2 dimensional manifold spanned by the two cat states $\{\mathcal{C}_\alpha^\pm\}$. This leads to a natural definition of an effective quantum bit, the so-called cat qubit. The cat qubit states are defined as $|\pm\rangle_c = |\mathcal{C}_\alpha^\pm\rangle$, or equivalently as

$$|0\rangle_c = \frac{1}{\sqrt{2}}\left(|\mathcal{C}_\alpha^+\rangle + |\mathcal{C}_\alpha^-\rangle\right) = |\alpha\rangle + \mathcal{O}\left(\exp(-2|\alpha|^2)\right),$$
$$|1\rangle_c = \frac{1}{\sqrt{2}}\left(|\mathcal{C}_\alpha^+\rangle - |\mathcal{C}_\alpha^-\rangle\right) = |-\alpha\rangle + \mathcal{O}\left(\exp(-2|\alpha|^2)\right).$$

Note that, with respect to most of our early publications, we have changed the computational basis to the dual basis along the $X$-axis. This choice is motivated by the simplifications in the presentation of the implemented logical gates proposed recently [9].

In terms of quantum information processing, the interest of this cat qubit lies in the fact that its physical implementation endows it with a natural protection. As soon as the action of a noise process is local in the phase space of the harmonic oscillator, the effective bit-flip errors (jumps between $|0\rangle_c$ and $|1\rangle_c$) are exponentially suppressed with $|\alpha|^2$ (the effective bit-flip rate is proportional to $\exp(-c|\alpha|^2)$ with an appropriate constant $c$). The idea behind this protection can be deduced from Fig. 6, where the vector field associated to the semi-classical

dynamics of a coherent state governed by (5) is plotted in the phase space of the oscillator. Any noise process that perturbs the coherent state $|\pm\alpha\rangle$ locally in the phase space, keeps it in the attraction domain of the departing point $|\pm\alpha\rangle$. Such a protection is similar to the one achieved by topological qubits such as Majorana fermions, but the non-locality of information is here engineered through the particular driven dissipative process of the harmonic oscillator. In particular, the non-locality can be tuned by modifying the cat "size", given by the mean number of photons $|\alpha|^2$. This mean number is itself easily modulated by controlling the strength, $\epsilon_{2ph}$, of the two-photon drive. The local character of the noise processes is an omnipresent concept in information protection, and in the case of superconducing oscillators, it includes various mechanisms such as photon loss, thermal excitations, photon dephasing and non-linear interaction Hamiltonians induced by Josephson circuits (this fact will be seen in the next section).

Note however that phase-flips, or equivalently jumps between even-parity cat state $|\mathcal{C}_\alpha^+\rangle$ and the odd-parity one $|\mathcal{C}_\alpha^-\rangle$, can be induced by noise mechanisms such as photon loss or thermal excitations. As a result, an increase of the mean photon number (in order to suppress the bit-flip errors) comes at the expense of higher phase-flip rates. This rate increase is however expected to be only linear with respect to $|\alpha|^2$. The noise bias $\exp(-c|\alpha|^2)/|\alpha|^2$ of cat qubits is therefore tunable with the cat size. An experimental proof of such an exponential and tunable bias have been recently observed [39].

Throughout the rest of this chapter, we provide a more detailed discussion of this bit-flip suppression.

## 2.3 A detailed discussion of bit-flip suppression

The two-photon driven dissipative process (5) can be seen as an autonomous error recovery operation. Throughout this section and for simplicity sakes, we assume $\alpha$ real and we define $\mathcal{M}_{2,\alpha}$ to be the manifold of density matrices on the Hilbert space spanned by $\{|\mathcal{C}_\alpha^\pm\rangle\}$. Given an initial state $\boldsymbol{\rho}(0)$, the system converges, via the two-photon process, to an asymptotic state $\boldsymbol{\rho}_f \in \mathcal{M}_{2,\alpha}$. This defines a quantum map $\mathbb{R}_2$ such that $\boldsymbol{\rho}_f = \mathbb{R}_2(\boldsymbol{\rho}(0))$. Applying the conserved quantity $\boldsymbol{J}_{+-}$, the super-operator $\mathbb{R}_2$ satisfies the following statements: for all complex number $\beta$ such that $|\Re(\beta)| < \alpha$,

$$\mathbb{R}_2(|\alpha+\beta\rangle\langle\alpha+\beta|) = |0\rangle_c\langle 0| + \mathcal{O}\left(e^{-\alpha^2-|\alpha+\beta|^2+|\beta(2\alpha+\beta)|}\right). \tag{10}$$

Indeed, the population of $|0\rangle_c$ and $|1\rangle_c$ in $\mathbb{R}_2(\boldsymbol{\rho}(0))$ is given by $\frac{1\pm\mathrm{Tr}(J_Z\boldsymbol{\rho}(0))}{2}$, where $J_Z = J_{+-} + J_{+-}^\dagger$. The statement (10) is therefore equivalent to

$$\langle\alpha+\beta|J_Z|\alpha+\beta\rangle = 1 + \mathcal{O}\left(e^{-\alpha^2-|\alpha+\beta|^2+|\beta(2\alpha+\beta)|}\right). \tag{11}$$

Following (9), we have

$$\langle\alpha+\beta|J_Z|\alpha+\beta\rangle = \frac{i\alpha|\alpha+\beta|e^{-|\alpha+\beta|^2-\alpha^2}}{\sqrt{1-e^{-4\alpha^2}}}$$

$$\times \left[\int_0^\pi d\Phi \left[e^{-i(\Phi_b+\Phi)}I_0(|\alpha^2-|\alpha+\beta|^2e^{i2(\Phi_b+\Phi)}|)-c.c\right]\right],$$

where $\alpha+\beta = |\alpha+\beta|e^{i\Phi_b}$. Noting that

$$|\alpha^2-|\alpha+\beta|^2e^{i2(\Phi_b+\Phi)}| = \sqrt{\alpha^4+|\alpha+\beta|^4-2\alpha^2|\alpha+\beta|^2\cos(2(\Phi+\Phi_b))},$$

We expand $\cos(2(\Phi + \Phi_b)) = 2\cos^2(\Phi + \Phi_b) - 1$, and make the change of variable $\Phi \to u := \cos(\Phi + \Phi_b)$. Note that this is allowed since the function $f(\Phi) = \cos(\Phi + \Phi_b)$ is bijective from the domain $]0, \pi[$ to the domain $]-\cos(\Phi_b), \cos(\Phi_b)[$. Indeed, from the condition $|\Re(\beta)| < \alpha$, one satisfies $\Phi_b \in ]-\frac{\pi}{2}, \frac{\pi}{2}[$. This change of variable gives

$$\langle \alpha + \beta | J_Z | \alpha + \beta \rangle = \frac{4\alpha |\alpha + \beta| e^{-|\alpha+\beta|^2 - \alpha^2}}{\sqrt{1 - e^{-4\alpha^2}}} \int\limits_0^{\cos(\Phi_b)} du \, I_0\Big(\sqrt{(\alpha^2 + |\alpha + \beta|^2)^2 - 4\alpha^2 |\alpha + \beta|^2 u^2}\Big).$$

We do another change of variable $u \to v := \sqrt{(\alpha^2 + |\alpha + \beta|^2)^2 - 4\alpha^2 |\alpha + \beta|^2 u^2}$ leading to

$$\langle \alpha + \beta | J_Z | \alpha + \beta \rangle = \text{sign}(\cos(\Phi_b)) \frac{2e^{-|\alpha+\beta|^2 - \alpha^2}}{\sqrt{1 - e^{-4\alpha^2}}} \int\limits_{m(\alpha, |\alpha+\beta|, \Phi_b)}^{m(\alpha, |\alpha+\beta|, \frac{\pi}{2})} \frac{I_0(v) v \, dv}{\sqrt{(\alpha^2 + |\alpha + \beta|^2)^2 - v^2}}$$

$$= \text{sign}(\cos(\Phi_b)) \frac{2e^{-|\alpha+\beta|^2 - \alpha^2}}{\sqrt{1 - e^{-4\alpha^2}}} \left[ \int\limits_0^{m(\alpha, |\alpha+\beta|, \frac{\pi}{2})} \frac{I_0(v) v \, dv}{\sqrt{(\alpha^2 + |\alpha + \beta|^2)^2 - v^2}} \right.$$

$$\left. - \int\limits_0^{m(\alpha, |\alpha+\beta|, \Phi_b)} \frac{I_0(v) v \, dv}{\sqrt{(\alpha^2 + |\alpha + \beta|^2)^2 - v^2}} \right],$$

where $m(\alpha, |\alpha+\beta|, \theta) = \sqrt{(\alpha^2 + |\alpha + \beta|^2)^2 - 4\alpha^2 |\alpha + \beta|^2 \cos^2(\theta)}$, and $\text{sign}(x) = +1$ if $x > 0$, and $\text{sign}(x) = -1$ if $x < 0$. The condition $|\Re(\beta)| < \alpha$ implies $\cos(\Phi_b) > 0$. Using (2.15.2.6) from [38], the first term gives

$$\frac{2e^{-|\alpha+\beta|^2 - \alpha^2}}{\sqrt{1 - e^{-4\alpha^2}}} \int\limits_0^{m(\alpha, |\alpha+\beta|, \frac{\pi}{2})} \frac{I_0(v) v \, dv}{\sqrt{(\alpha^2 + |\alpha + \beta|^2)^2 - v^2}} = \frac{1 - e^{-2\alpha^2 - 2|\alpha+\beta|^2}}{\sqrt{1 - e^{-4\alpha^2}}}.$$

The same formula implies that

$$\frac{2e^{-|\alpha+\beta|^2 - \alpha^2}}{\sqrt{1 - e^{-4\alpha^2}}} \int\limits_0^{m(\alpha, |\alpha+\beta|, \Phi_b)} \frac{I_0(v) v \, dv}{\sqrt{(\alpha^2 + |\alpha + \beta|^2)^2 - v^2}} < \mathcal{O}\left( \frac{e^{-\alpha^2 - |\alpha+\beta|^2 + |\beta(2\alpha+\beta)|}}{\sqrt{1 - e^{-4\alpha^2}}} \right).$$

Hence, we have

$$\langle \alpha + \beta | J_Z | \alpha + \beta \rangle = \frac{1}{\sqrt{1 - e^{-4\alpha^2}}} \left( 1 - \mathcal{O}\left( e^{-\alpha^2 - |\alpha+\beta|^2 + |\beta(2\alpha+\beta)|} \right) \right)$$

$$> 1 - \mathcal{O}\left( e^{-\alpha^2 - |\alpha+\beta|^2 + |\beta(2\alpha+\beta)|} \right).$$

From $|\text{Tr}(\rho J_{+-})| = |c_{+-}| \leq 1$, we infer that the operator $J_Z$ satisfies, $\forall \rho, |\text{Tr}(\rho J_Z)| \leq 1$. This leads to

$$1 - \mathcal{O}\left( e^{-\alpha^2 - |\alpha+\beta|^2 + |\beta(2\alpha+\beta)|} \right) < \langle \alpha + \beta | J_Z | \alpha + \beta \rangle \leq 1.$$

This clearly shows that

$$\langle \alpha + \beta | J_Z | \alpha + \beta \rangle = 1 - \mathcal{O}\left( e^{-\alpha^2 - |\alpha+\beta|^2 + |\beta(2\alpha+\beta)|} \right).$$

Note that $e^{-\alpha^2-|\alpha+\beta|^2+|\beta(2\alpha+\beta)|} = e^{-\alpha^2-|\alpha+\beta|^2+|(\alpha+\beta)^2-\alpha^2|}$. In particular, for a real negative number $\beta$, we have $e^{-\alpha^2-|\alpha+\beta|^2+|\beta(2\alpha+\beta)|} = e^{-2(\alpha-\beta)^2}$. In the sequel, we provide a discussion of the conditions for the noise processes on a harmonic oscillator such that they are corrected through the recovery operation $\mathbb{R}_2$.

*General analysis.* Let us consider a general operator $E(a, a^\dagger)$ defined on the Hilbert space of a harmonic oscillator. Here, we assume $E$ to be analytical function of its arguments $a$ and $a^\dagger$. We would like to study the effect of such an operator on the code space $\mathcal{M}_{2,\alpha}$. This operator can be written in the form

$$E(a, a^\dagger) = F^I\left(a^2, a^{\dagger 2}, a^\dagger a\right) I + F^{Z,-}\left(a^2, a^{\dagger 2}, a^\dagger a\right) a + F^{Z,+}\left(a^2, a^{\dagger 2}, a^\dagger a\right) a^\dagger,$$

where $F^I$, $F^{Z,\pm}$, are analytical functions of $a^2$, $a^{\dagger 2}$ and $a^\dagger a$. In particular, the photon number parity is conserved through the action of such operators. From the relation $a|\mathcal{C}_\alpha^\pm\rangle = \alpha|\mathcal{C}_\alpha^\mp\rangle$, we infer that $a\Pi_{\mathcal{M}_{2,\alpha}} = \alpha Z_c$ and $a^2\Pi_{\mathcal{M}_{2,\alpha}} = \alpha^2\Pi_{\mathcal{M}_{2,\alpha}}$, where $\Pi_{\mathcal{M}_{2,\alpha}} = |\mathcal{C}_\alpha^+\rangle\langle\mathcal{C}_\alpha^+| + |\mathcal{C}_\alpha^-\rangle\langle\mathcal{C}_\alpha^-|$ and $Z_c = |\mathcal{C}_\alpha^+\rangle\langle\mathcal{C}_\alpha^-| + |\mathcal{C}_\alpha^-\rangle\langle\mathcal{C}_\alpha^+|$. Thus, a single photon jump maps the logical subspace onto itself, and acts as a phase-flip $Z_c$ of the cat qubit. While we will deal with such phase-flips later, based on the same argument, note that the action of the parity-preserving operators $F^I$, $F^{X,\pm}$ cannot result in phase-flip errors. As we shall see below, these errors induce at most bit-flip errors in the cat qubit. Since we are interested in the action of $E$ on the manifold $\mathcal{M}_{2,\alpha}$, we focus on the operator $E\Pi_{\mathcal{M}_{2,\alpha}}$. By writing $a^\dagger\Pi_{\mathcal{M}_{2,\alpha}} = a^\dagger a^2 \Pi_{\mathcal{M}_{2,\alpha}}/\alpha^2 = a^\dagger a \sigma_X^L/\alpha$, the operator $E(a, a^\dagger)\Pi_{\mathcal{M}_{2,\alpha}}$ admits the decomposition

$$E(a, a^\dagger)\Pi_{\mathcal{M}_{2,\alpha}} = F^I(a^2, a^{\dagger 2}, a^\dagger a)\Pi_{\mathcal{M}_{2,\alpha}} + F^{Z,\alpha}(a^2, a^{\dagger 2}, a^\dagger a)Z_c,$$

with $F^{Z,\alpha}(a^2, a^{\dagger 2}, a^\dagger a) = \alpha F^{Z,-}(a^2, a^{\dagger 2}, a^\dagger a) + F^{Z,+}(a^2, a^{\dagger 2}, a^\dagger a)a^\dagger a/\alpha$. A general error $E$ acts therefore as a linear combination of a parity-preserving operator (unable to induce cat qubit phase-flips) and the product of a parity-preserving operator with a cat qubit phase-flip. Although the cat qubit phase-flip component is needed to be taken care of otherwise, we show here that this two-photon process is capable of protecting the information against a large class of parity-preserving errors of the type $F(a^2, a^{\dagger 2}, a^\dagger a)$.

Let us consider a noise map $\mathbb{F}$, described by parity-preserving errors $\{F_k(a^2, a^{\dagger 2}, a^\dagger a)\}$. This set of errors is correctable if and only if the operators $F_k$ satisfy the criteria, i.e

$$\Pi_{\mathcal{M}_{2,\alpha}} F_j^\dagger F_k \Pi_{\mathcal{M}_{2,\alpha}} = c_{jk}\Pi_{\mathcal{M}_{2,\alpha}}. \tag{12}$$

As the operators $F_k$ are invariant under the transformation $a \to -a$, we have the equality $\langle\alpha|F_j^\dagger F_k|\alpha\rangle = \langle-\alpha|F_j^\dagger F_k|-\alpha\rangle = c_{jk}$. This leads to

$$\Pi_{\mathcal{M}_{2,\alpha}} F_j^\dagger F_k \Pi_{\mathcal{M}_{2,\alpha}} = c_{jk}\Pi_{\mathcal{M}_{2,\alpha}} + m_{jk}X_c,$$

where $X_c = |\mathcal{C}_\alpha^+\rangle\langle\mathcal{C}_\alpha^+| - |\mathcal{C}_\alpha^-\rangle\langle\mathcal{C}_\alpha^-|$ is the cat qubit bit-flip operation. If one satisfies $m_{jk} := \langle-\alpha|F_j^\dagger F_k|\alpha\rangle = \langle\alpha|F_j^\dagger F_k|-\alpha\rangle = \mathcal{O}(\epsilon)$ with $\epsilon$ a small parameter, we can find a recovery map $\mathbb{R}$ such that $\forall\rho \in \mathcal{M}_{2,\alpha}$, $(\mathbb{R} \circ \mathbb{F})(\rho) = \rho + \mathcal{O}(\epsilon)$. In this case, we say that the noise map $\mathbb{F}$ is (approximately) correctable *up to* $\mathcal{O}(\epsilon)$ [40]. Roughly, a sufficient condition for this, is that for all $j, k$, the states $F_k|\alpha\rangle$ and $F_j|-\alpha\rangle$ remain far enough, perhaps in the right half and left half planes of the phase space. This can be seen by writing $\langle-\alpha|F_j^\dagger F_k|\alpha\rangle = (1/\pi)\int_\mathbb{C} d^2\beta\,\langle-\alpha|F_j^\dagger|\beta\rangle\langle\beta|F_k|\alpha\rangle$ and by noting that such a condition ensures the smallness of the quantity $\langle-\alpha|F_j^\dagger|\beta\rangle\langle\beta|F_k|\alpha\rangle$ for all $\beta \in \mathbb{C}$.

Let us outline the strategy of our analysis. We show that if the action of the error operators $F(a^2, a^{\dagger 2}, a^\dagger a)$, send the code states $|\pm\alpha\rangle$ to states that are spanned by near enough coherent

states span$\{| \pm \alpha + \beta \rangle \ | \ |\beta| < \varepsilon\}$, they are corrected by $\mathbb{R}_2$ up to an $\mathcal{O}(\exp(-2|\alpha|^2))$. Next, we show that the major physical error channels lead to such operators.

*Protection agains parity-preserving errors-* Consider a set of parity-preserving errors $\{F_k(a^2, a^{\dagger 2}, a^\dagger a)\}$. Let us assume that there exists a function $c(\alpha): \mathbb{C} \mapsto \mathbb{R}^+$ satisfying

$$\lim_{|\alpha| \to \infty} \frac{c(\alpha)}{|\alpha|} = 0,$$

such that

$$F_k | \pm \alpha \rangle \in \text{span}\{| \pm \alpha + \beta \rangle \ | \ |\beta| < c(\alpha)\}. \tag{13}$$

This set of errors is correctable up to $\mathcal{O}(e^{-2|\alpha|^2})$. More precisely, one can find a recovery operation such that any state $\rho \in \mathcal{M}_{2,\alpha}$ subject to error channels of the form $\{F_k(a^2, a^{\dagger 2}, a^\dagger a)\}$, can be recovered with a unit fidelity up to $\mathcal{O}(e^{-2|\alpha|^2})$. Furthermore, this recovery operation is given by the two-photon process $\mathbb{R}_2$. This claim is easily proven through the formula (10).

Here, we illustrate the previous analysis with various examples of decoherence channels. In what follows, we choose $\alpha$ real and positive.

*Example: photon loss channel -* The dynamics of an oscillator subject only to single-photon dissipation at rate $\kappa$ is well described by the Lindblad master equation

$$\frac{d}{dt}\rho = \kappa \mathcal{D}[a]\rho.$$

For such a master equation the evolution of the system's density matrix $\rho$ over a time interval $\delta t$, can be represented by the Kraus map [24]

$$\rho(t + \delta t) = \sum_{k=0}^{\infty} E_k \rho(t) E_k^\dagger, \quad E_k = \sqrt{\frac{(1 - e^{-\kappa \delta t})^k}{k!}} e^{-\frac{\kappa \delta t}{2}n} a^k, \tag{14}$$

where $n = a^\dagger a$ represents the photon number operator. The term $E_k \rho(t) E_k^\dagger$ is the state of the system at time $t + \delta t$ if $k$ photon jumps (losses) have occurred within the time interval $\delta t$, weighted by the probability of this event. The state of the system $\rho(t + \delta t)$ is then obtained by summing over the number of jumps. The set of errors $E_k$ can be decomposed into the operators $E_{2k}$ and $E_{2k+1}$ involving an even and odd number of photon jumps respectively. Moreover, $E_{2k+1}$ expands as $E_{2k+1} = F_{2k} a$, where $F_{2k} = \sqrt{\frac{(1-e^{-\kappa \delta t})^{2k+1}}{(2k+1)!}} e^{-\frac{\kappa \delta t}{2}n} a^{2k}$. Since $a^2 \Pi_{\mathcal{M}_{2,\alpha}} = \alpha^2 \Pi_{\mathcal{M}_{2,\alpha}}$, the operators $E_{2k} \Pi_{\mathcal{M}_{2,\alpha}}$ read

$$E_{2k} \Pi_{\mathcal{M}_{2,\alpha}} = \alpha^{2k} \sqrt{\frac{(1 - e^{-\kappa \delta t})^{2k}}{(2k)!}} e^{-\frac{\kappa \delta t}{2}n} \Pi_{\mathcal{M}_{2,\alpha}}.$$

Similarly, we have $F_{2k} \Pi_{\mathcal{M}_{2,\alpha}} = \alpha^{2k} \sqrt{(1 - e^{-\kappa \delta t})^{2k+1}/(2k+1)!} e^{-\frac{\kappa \delta t}{2}n} \Pi_{\mathcal{M}_{2,\alpha}}$. Noting that

$$\frac{e^{-\kappa \delta t n/2}| \pm \alpha \rangle}{\|e^{-\kappa \delta t n/2}| \pm \alpha \rangle\|} = | \pm \alpha e^{-\kappa \delta t/2} \rangle,$$

the operators $E_{2k}$ and $F_{2k}$ map the coding subspace to $\mathcal{M}_{2, \alpha e^{-\kappa t/2}}$, while leaving the photon number parity unchanged.

Now, we note the convergence rate to $\mathcal{M}_{2,\alpha}$ due to the two-photon process scales with $|\alpha|^2$. This fact will be shown in Section 3.4. Following a discussion similar to the one detailed in Section 1.2.1, the time duration $\delta t$ between two corrections can be fixed as $\tau_m/|\alpha|^2$ with a constant time $\tau_m$ independent of the cat size (and scaling with the two-photon dissipation

rate). Therefore, The action of all error operators $F_{2k}$ and $E_{2k}$ send the code states $|\pm\alpha\rangle$ to $|\pm\alpha e^{-k\tau_m/2|\alpha|^2}\rangle$. We note that

$$\alpha e^{-k\tau_m/2|\alpha|^2} = \alpha - \frac{k\tau_m}{2|\alpha|} + \mathcal{O}\left(\frac{1}{|\alpha|^2}\right).$$

Therefore one can fix $\varepsilon > 0$ such that the coherent states $|\pm\alpha\rangle$ are sent to coherent states within $\varepsilon$ neighbourhood of the initial states. This indicates that such photon loss errors are corrected up to an $\mathcal{O}(-2|\alpha|^2)$. This indication relying on a descretization of the continuous error correction process is confirmed through the numerical simulations of Figure 7.

*Example: Gaussian displacement channel -* In such a model, we assume that the state of the harmonic oscillator undergoes a random displacement of value $\beta \in \mathbb{C}$ with $\beta$ a Gaussian random variable centered at the origin and with the standard deviation $\sqrt{\Gamma\delta t}$ proportional to the square root of a waiting time $\delta t$. It is therefore modelled by the Kraus map

$$\rho(t+\delta t) = \frac{1}{2\pi\delta t\Gamma} \int_{\beta\in\mathbb{C}} d^2\beta \, e^{-\frac{|\beta|^2}{2\delta t\Gamma}} D_\beta \rho(t) D_\beta^\dagger.$$

Once again, we note that the time $\delta t$ between error correction operations scales at $\tau_m/|\alpha|^2$. Therefore, the evolution between two correction steps is given by

$$\rho_+ = \frac{|\alpha|^2}{2\pi\tau_m\Gamma} \int_{\beta\in\mathbb{C}} d^2\beta \, e^{-\frac{|\alpha|^2|\beta|^2}{2\tau_m\Gamma}} D_\beta \rho D_\beta^\dagger.$$

For a function $c(\alpha) = \sqrt{|\alpha|}$ satisfying $c(\alpha)/|\alpha| \to 0$ as $|\alpha| \to \infty$, we have

$$\frac{|\alpha|^2}{2\pi\tau_m\Gamma} \int_{\beta\in\mathbb{C}} d^2\beta \, e^{-\frac{|\alpha|^2|\beta|^2}{2\tau_m\Gamma}} D_\beta |\alpha\rangle\langle\alpha| D_\beta^\dagger =$$
$$\frac{|\alpha|^2}{2\pi\tau_m\Gamma} \int_{|\beta|<c(\alpha)} d^2\beta \, e^{-\frac{|\alpha|^2|\beta|^2}{2\tau_m\Gamma}} D_\beta |\alpha\rangle\langle\alpha| D_\beta^\dagger + \frac{|\alpha|^2}{2\pi\tau_m\Gamma} \int_{|\beta|>c(\alpha)} d^2\beta \, e^{-\frac{|\alpha|^2|\beta|^2}{2\tau_m\Gamma}} D_\beta |\alpha\rangle\langle\alpha| D_\beta^\dagger.$$

While, the left term is corrected by $\mathbb{R}_2$ up to an $\mathcal{O}(\exp(-2|\alpha|^2))$, we have for the right term

$$\frac{|\alpha|^2}{2\pi\tau_m\Gamma} \left\| \int_{|\beta|>c(\alpha)} d^2\beta \, e^{-\frac{|\alpha|^2|\beta|^2}{2\tau_m\Gamma}} D_\beta |\alpha\rangle\langle\alpha| D_\beta^\dagger \right\| \leq \frac{|\alpha|^2}{2\pi\tau_m\Gamma} \int_{|\beta|>\sqrt{|\alpha|}} d^2\beta \, e^{-\frac{|\alpha|^2|\beta|^2}{2\tau_m\Gamma}} = e^{-\frac{|\alpha|^3}{2\tau_m\Gamma}}.$$

Therefore, this error channel is corrected up to an $\mathcal{O}(-2|\alpha|^2) + \mathcal{O}(-\tilde{\gamma}|\alpha|^3)$, with $\tilde{\gamma} = 1/2\tau_m\Gamma$. This indicates that the total error channel is corrected up to an $\mathcal{O}(-2|\alpha|^2)$.

*Example: Markovian photon dephasing-* The dynamics of an oscillator subject only to Markovian photon dephasing at rate $\kappa_\phi$ is well described the Lindblad master equation

$$\frac{d}{dt}\rho = \kappa_\phi \mathcal{D}[a^\dagger a]\rho.$$

For such a master equation the evolution of the system's density operator $\rho$ over a time interval $\delta t$, can be represented by the Kraus map

$$\rho(t+\delta t) = \frac{1}{\sqrt{2\pi\delta t\kappa_\phi}} \int_{-\infty}^{\infty} d\phi \, e^{-\frac{|\phi|^2}{2\delta t\kappa_\phi}} e^{i\phi a^\dagger a} \rho(t) e^{-i\phi a^\dagger a}.$$

Taking $\delta t = \tau_m/|\alpha|^2$, $\rho = |\alpha\rangle\langle\alpha|$ and $c(\alpha) = |\alpha|^{1-\eta}$ with $\eta > 0$ small, we have

$$\frac{|\alpha|}{\sqrt{2\pi\tau_m\kappa_\phi}}\int_{-\infty}^{\infty}d\phi\, e^{-\frac{|\alpha|^2|\phi|^2}{2\tau_m\kappa_\phi}}e^{i\phi a^\dagger a}|\alpha\rangle\langle\alpha|e^{-i\phi a^\dagger a} =$$

$$\frac{|\alpha|}{\sqrt{2\pi\tau_m\kappa_\phi}}\int_{|\phi|<c(\alpha)/|\alpha|}d\phi\, e^{-\frac{|\alpha|^2|\phi|^2}{2\tau_m\kappa_\phi}}e^{i\phi a^\dagger a}|\alpha\rangle\langle\alpha|e^{-i\phi a^\dagger a}+$$

$$\frac{|\alpha|}{\sqrt{2\pi\tau_m\kappa_\phi}}\int_{|\phi|>c(\alpha)/|\alpha|}d\phi\, e^{-\frac{|\alpha|^2|\phi|^2}{2\tau_m\kappa_\phi}}e^{i\phi a^\dagger a}|\alpha\rangle\langle\alpha|e^{-i\phi a^\dagger a}. \quad (15)$$

We have

$$e^{i\phi a^\dagger a}|\alpha\rangle = |e^{i\phi}\alpha\rangle,$$

and therefore, for $|\phi| < c(\alpha)/|\alpha| = 1/|\alpha|^\eta$, we have

$$|\alpha - e^{i\phi}\alpha| = |\alpha|\sqrt{2(1-\cos\phi)} < c(\alpha) = |\alpha|^{1-\eta} \qquad \text{in the limit } |\alpha| \to \infty.$$

Thus in (15), the first term is corrected by $\mathbb{R}_2$ up to an $\mathcal{O}(\exp(-2|\alpha|^2))$. For the second term, we have

$$\frac{|\alpha|}{\sqrt{2\pi\tau_m\kappa_\phi}}\left\|\int_{|\phi|>c(\alpha)/|\alpha|}d\phi\, e^{-\frac{|\alpha|^2|\phi|^2}{2\tau_m\kappa_\phi}}e^{i\phi a^\dagger a}|\alpha\rangle\langle\alpha|e^{-i\phi a^\dagger a}\right\| \leq \frac{|\alpha|}{\sqrt{2\pi\tau_m\kappa_\phi}}\int_{|\phi|>c(\alpha)/|\alpha|}d\phi\, e^{-\frac{|\alpha|^2|\phi|^2}{2\tau_m\kappa_\phi}}$$

$$= 1 - \text{erf}\left(\frac{|\alpha|^{1-\eta}}{\sqrt{2\tau_m\kappa_\phi}}\right).$$

Noting that $1 - \text{erf}(x) \leq e^{-x^2}$ for $x \geq 0$, we have

$$1 - \text{erf}\left(\frac{|\alpha|^{1-\eta}}{\sqrt{2\tau_m\kappa_\phi}}\right) \leq e^{-\frac{|\alpha|^{2-2\eta}}{2\tau_m\kappa_\phi}}.$$

We note that for $\kappa_\phi\tau_m \leq 1/4$ (meaning that the dephasing rate 4 times lower than the two-photon dissipation rate), the above error function is dominated by $e^{-2|\alpha|^{2-2\eta}}$. This indicates that the Markovian dephasing error is suppressed up to an $\mathcal{O}(e^{-2|\alpha|^{2-\epsilon}})$ for all $\epsilon > 0$.

*Example: phase noise due to dispersive coupling to a high-Q mode at non-zero temperature*- Let us consider that the mode $a$ is coupled to a mode $b$ through the cross-Kerr coupling $H_{\text{int}} = -\hbar\chi a^\dagger a b^\dagger b$. This mode $b$, coupled to a non-zero temperature bath, is in the thermal equilibrium $\rho_b^s = \sum_n p_n|n\rangle\langle n|$, associated to the mean photon number $n_{\text{th}} = \sum_n np_n$. In the expansion $H_{\text{int}} = -\hbar\chi a^\dagger a(b^\dagger b - n_{\text{th}}) - \hbar\chi n_{\text{th}}a^\dagger a$, we keep only the first term, as the second term induces a deterministic phase rotation that can be taken into account in the cat code pumping. Given an initial state of the form $\rho(0) = \rho_a \otimes \rho_b^s$, the state at time $\delta t$ is given by $\rho(\delta t) = e^{i\chi\delta t a^\dagger a(b^\dagger b - n_{\text{th}})}\rho(0)e^{-i\chi\delta t a^\dagger a(b^\dagger b - n_{\text{th}})}$. The state of the mode $a$ at time $\delta t$ reads $\rho_a(\delta t) = \text{tr}_b[\rho(\delta t)]$, i.e

$$\rho_a(\delta t) = \sum_n p_n e^{i\chi\delta t(n-n_{\text{th}})a^\dagger a}\rho_a e^{-i\chi\delta t(n-n_{\text{th}})a^\dagger a}.$$

The set of errors associated to this map is the set of unitary errors $\{E_n = \sqrt{p_n}e^{i\chi\delta t(n-n_{\text{th}})a^\dagger a}\}$. Once again, we note that the time $\delta t$ between error correction operations scales at $\tau_m/|\alpha|^2$. Therefore,

$$E_n|\alpha\rangle = \sqrt{p_n}|\alpha e^{i\chi\tau_m(n-n_{\text{th}})/|\alpha|^2}\rangle.$$

Now calling $\theta_n = \chi \tau_m (n - n_{\text{th}})/|\alpha|^2$, we have

$$|\alpha e^{i\theta_n} - \alpha|^2 = 2|\alpha|^2 (1 - \cos\theta_n).$$

We note that, for $c(\alpha) = |\alpha|^{1-\eta}$, and in the limit of large $|\alpha|$, $|\alpha e^{i\theta_n} - \alpha| < c(\alpha)$ is equivalent to

$$|n - n_{\text{th}}| < |\alpha|^{2-\eta}/\chi \tau_m.$$

Therefore, writing

$$
\begin{aligned}
\boldsymbol{\rho}_a(\delta t) &= \sum_n p_n e^{i\chi\delta t(n-n_{\text{th}})\boldsymbol{a}^\dagger \boldsymbol{a}} \boldsymbol{\rho}_a e^{-i\chi\delta t(n-n_{\text{th}})\boldsymbol{a}^\dagger \boldsymbol{a}} \\
&= \sum_{n \leq n_{\text{th}}+|\alpha|^{2-\eta}/\chi\tau_m} p_n |\alpha e^{i\theta_n}\rangle\langle\alpha e^{i\theta_n}| + \sum_{n > n_{\text{th}}+|\alpha|^{2-\eta}/\chi\tau_m} p_n |\alpha e^{i\theta_n}\rangle\langle\alpha e^{i\theta_n}|,
\end{aligned}
\tag{16}
$$

the left hand term is corrected by $\mathbb{R}_2$ up to an $\mathcal{O}(\exp(-2|\alpha|^2))$. For the right hand term, we have

$$\left\| \sum_{n > n_{\text{th}}+|\alpha|^{2-\eta}/\chi\tau_m} p_n |\alpha e^{i\theta_n}\rangle\langle\alpha e^{i\theta_n}| \right\| \leq \sum_{n > n_{\text{th}}+|\alpha|^{2-\eta}/\chi\tau_m} p_n.$$

Noting that for a thermal distribution $p_n = r^n/(1+n_{\text{th}})$ with $r = n_{\text{th}}/(1+n_{\text{th}})$, we have

$$\sum_{n > n_{\text{th}}+|\alpha|^{2-\eta}/\chi\tau_m} p_n \leq r^{n_{\text{th}}+|\alpha|^{2-\eta}/\chi\tau_m} = e^{n_{\text{th}}\log r + |\alpha|^{2-\eta}\log r/\chi\tau_m}.$$

Therefore taking $\chi\tau_m/|\log r| \leq 1/2$ (meaning $\chi \leq \kappa_2 |\log r|/2$), the right hand term in (16) is dominated in norm by an $\mathcal{O}(e^{-2|\alpha|^{2-\eta}})$ for all $\eta > 0$.

Before ending this section, we provide a numerical simulation illustrating such an exponential suppression in presence of single photon loss and Markovian dephasing. Other effects could be added to these numerical simulations. In order to understand these simulations, we note that the cat qubit codespace is a subspace of the infinite dimensional Hilbert space of a quantum harmonic oscillator. Various noise mechanisms could therefore lead to a leakage out of the code space, even though the two-photon dissipation mechanism tends to steer the state back to the codespace. One way to take into account this leakage in the calculations of the qubit properties (e.g. bit-flip and phase-flip errors) is to specify the qubit state through observables of the quantum harmonic oscillator. Therefore, whole subspaces of the harmonic oscillator state space are associated to a logical qubit value in a way that makes sense compared to the typical measurements. In other words, the logical qubit value is uniquely defined from Pauli expectation values $\langle\hat{\sigma}_i\rangle = \text{Tr}(\boldsymbol{J}_i\boldsymbol{\rho})$ with $i = x, y, z$. In the presence of the two photon dissipation, the most natural choice for these observables $\boldsymbol{J}_i$ are the invariants as introduced before.

In Figure 7, we simulate the master equation

$$\frac{d}{dt}\boldsymbol{\rho} = \kappa_2 \mathcal{D}[\boldsymbol{a}^2 - \alpha^2]\boldsymbol{\rho} + \kappa_1 \mathcal{D}[\boldsymbol{a}]\boldsymbol{\rho} + \kappa_\phi \mathcal{D}[\boldsymbol{a}^\dagger \boldsymbol{a}]\boldsymbol{\rho}.$$

We initialize the system at $|0\rangle_c \approx |\alpha\rangle$ and we calculate the value of the invariant $\boldsymbol{J}_Z$ after a time of $1/\kappa_2$. This corresponds to an effective bit-flip error probability after this time. The simulations show clearly an exponential suppression of bit-flips as predicted through the above analysis at a rate proportional to $\exp(-2|\alpha|^2)$.

Note also that, in the discussion of this section we have only studied an effective model associated to the two-photon driven dissipation. In the next chapter, we discuss how to effectively achieve such a process.

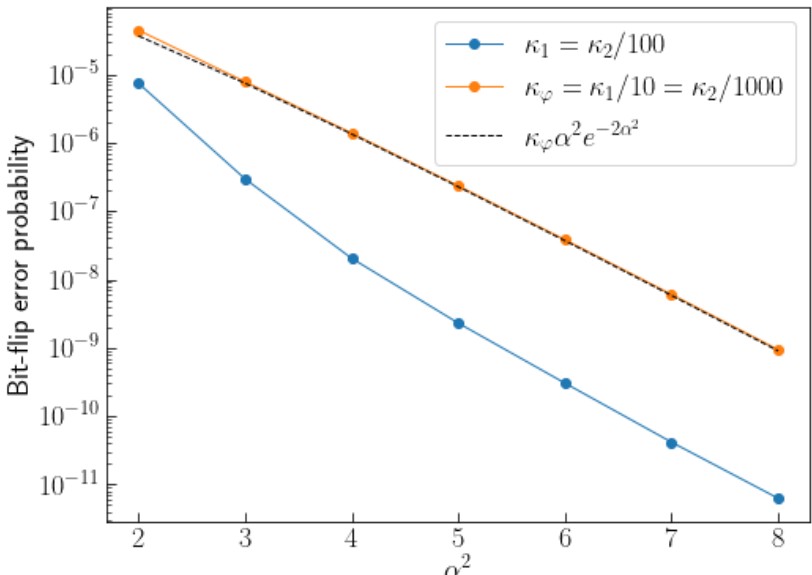

Figure 7: The probability of bit-flip after a time of order $1/\kappa_2$ in presence of single photon loss (blue curve) and both single photon loss and Markovian photon dephasing (orange curve).

## 3 Realizing two-photon driven dissipation

In this chapter, we go through approaches to implement multi-photon driven dissipative processes using the coupling of cavity modes to a circuit based on Josephson junctions. The so-called *parametric methods,* let us to combine a microwave driving pump satisfying a frequency matching condition, and a certain non-linear interaction term, to engineer un-natural linear or nonlinear effective Hamiltonians. In the past, such methods have been used, for instance, to achieve frequency conversion [41], quantum-limited amplification [42], two-mode squeezing [43], transverse readout of a qubit [44], and multi-photon exchanges between two modes [18].

In the context of quantum superconducting circuits, the required non-linear interactions are provided by the Josephson junctions. While, in the standard approaches to quantum information processing, the Josephson junctions are directly used to encode the information as an artificial atom, here they merely play the role of a nonlinear crystal providing multi-wave mixing. Note however that, compared to nonlinear crystals in the optical regime, Josephson circuits have a much larger ratio between multi-wave mixing and decoherence rates [45–47]. Therefore, they operate at regimes that have never been accessible to quantum optics experiments.

We will first focus on the case of two-photon driven dissipative processes. After providing the general picture, we will start by presenting the simplest implementation of a two-photon exchange Hamiltonian and will discuss its limitations. We will next present some alternative circuits that potentially remove some of these limitations. Next, we will show how from the effective two-photon exchange Hamiltonian, and through adiabatic elimination techniques, one achieves a two-photon dissipative process. We will also provide the corrections to this picture by higher order terms in such an approximation.

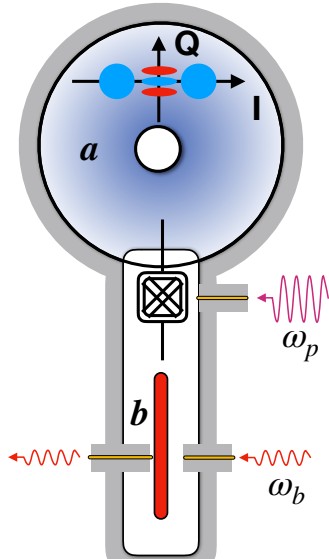

Figure 8: Lay-out of a possible implementation of two-photon driven dissipative process. A high-Q 3D cylindrical post-cavity [49] (the blue disk represented by the annihilation operator $a$) is coupled through a Josephson circuit (double box with double crosses) to a low-Q strip-line resonator (red strip represented by the annihilation operator $b$). The Josephson circuit is strongly driven by a microwave pump drive at a well-chosen frequency $\omega_p$ to effectively engineer an interaction Hamiltonian of the form $H_2/\hbar = g_2 a^2 b^\dagger + g_2^* a^{2\dagger} b$. The strip-line resonator is driven at its resonance (frequency $\omega_b$) and loses its photons through the strongly coupled port on the left.

## 3.1 General picture

Using the coupling of cavity modes to a Josephson junction (JJ), single photon dissipation, and coherent drives, we aim to produce effective dynamics in the form of Eq. (5). These are the same tools used in the Josephson Bifurcation Amplifier (JBA) to produce a squeezing Hamiltonian [48] and here we will show that, by selecting a particular pump frequency, we can achieve a two photon driven dissipative process. The general picture of a device implementing an effective two-photon driven dissipative process is presented in Fig. 8.

In a parametric approach, and through driving a non-linear coupling between two modes $a$ and $b$ at a well-chosen frequency $\omega_p$, we engineer a two-photon interaction Hamiltonian of the form

$$\frac{H_2}{\hbar} = g_2 a^2 b^\dagger + g_2^* a^{2\dagger} b, \tag{17}$$

where the complex coupling amplitude $g_2$ is controlled by the phase and amplitude of the pumping drive. The mode $a$ is assumed to be high-Q and for now we will neglect its loss. The mode $b$ is however intentionally lossy and loses photons at a rate $\kappa_b$. Finally, this low-Q mode is driven at its resonance $\omega_b$. The master equation in the rotating frame of the two harmonic oscillators is given by

$$\frac{d}{dt}\rho = -i[g_2 a^2 b^\dagger + g_2^* a^{2\dagger} b, \rho] - i[\epsilon_d b^\dagger + \epsilon_d^* b, \rho] + \kappa_b \mathcal{D}[b]\rho, \tag{18}$$

where the complex amplitude $\epsilon_d$ represents the amplitude and phase of the resonant drive at frequency $\omega_b$. This master equation can also be written in the form

$$\frac{d}{dt}\rho = -i[g_2(a^2 - \alpha^2)b^\dagger + g_2^*(a^2 - \alpha^2)^\dagger b, \rho] + \kappa_b \mathcal{D}[b]\rho,$$

where

$$\alpha = \sqrt{-\frac{\epsilon_d}{g_2}}.$$

In particular, for a fixed two-photon interaction strength $|g_2|$, the magnitude of $\alpha$ is simply controlled by the amplitude of the resonant drive $|\epsilon_d|$. As we will see in Section 3.4, assuming $\kappa_b$ to be large enough compared to $|g_2|$, we can adiabatically eliminate the dynamics of the mode $b$ to achieve an effective two-photon driven dissipative dynamics for the mode $a$ represented by the master equation (5).

## 3.2 Engineering a two-photon exchange Hamiltonian

In this section and the next one, we focus on the engineering of the two-photon interaction Hamiltonian (17).

Here, we start by the simplest possible implementation where the Josephson circuit in Fig. 8 is a single Josephson junction with two antennas coupling on one side to the high-Q 3D cavity mode (called the storage mode) and on the other side to the low-Q strip-line resonator (called the dump mode). This is the case in the experimental implementations [18,50].

We assume the dump mode to be driven at two frequencies $\omega_p$ and $\omega_d$. The frequency $\omega_p$ is chosen close to $2\omega_a - \omega_b$ and the frequency $\omega_d$ is close to $\omega_b$. The total Hamiltonian is given by [51]

$$\frac{H}{\hbar} = \tilde{\omega}_a a^\dagger a + \tilde{\omega}_b b^\dagger b - \frac{E_J}{\hbar}\left(\cos(\varphi) + \frac{\varphi^2}{2}\right) + \left(\epsilon_p e^{-i\omega_p t} + \epsilon_p^* e^{i\omega_p t} + \epsilon_d e^{-i\omega_d t} + \epsilon_d^* e^{i\omega_d t}\right)(b^\dagger + b). \quad (19)$$

The bare frequencies $\tilde{\omega}_{a,b}$ are shifted towards the measured frequencies $\omega_{a,b}$ due to the Lamb shift induced by the contribution of the Josephson Hamiltonian. This Hamiltonian is represented by the cosine term, where we have subtracted the quadratic terms, already taken into account in the first two terms. Furthermore, $E_J$ represents the Josephson energy associated to the junction, and $\varphi$ is the phase across the junction. Here, $\varphi$ can be decomposed as a linear combination of various modes participating to this phase across the junction

$$\varphi = \varphi_a(a + a^\dagger) + \varphi_b(b^\dagger + b),$$

where $\varphi_{a,b}$ denote the contribution of the modes $a$ and $b$ to the zero point fluctuations of $\varphi$. Note that, other modes could participate to this phase (e.g. the Junction's mode will contribute significantly). However, here we assume that these modes are not driven and are therefore in their vacuum state. Therefore, they only contribute to a renormalisation of the Josephson energy $E_J$. Finally, $\epsilon_{p,d}$ and $\omega_{p,d}$ represent the complex amplitudes and frequencies of the two microwave drives irradiating the dump mode.

Now, we make a change of frame consisting in going to an appropriate rotating frame for the modes $a$ and $b$ and furthermore displacing the $b$ mode to take into account the effect of the pump $\epsilon_p$. This change of frame is given by the unitary

$$U(t) = e^{i\omega_d t b^\dagger b} e^{i\frac{\omega_d+\omega_p}{2}t a^\dagger a} e^{-\tilde{\xi}_p b^\dagger + \tilde{\xi}_p^* b},$$

$$\frac{d}{dt}\tilde{\xi}_p = -i\tilde{\omega}_b \tilde{\xi}_p - i\left(\epsilon_p e^{-i\omega_p t} + \epsilon_p^* e^{i\omega_p t}\right) - \frac{\kappa_b}{2}\tilde{\xi}_p.$$

Note that the displacement value $\tilde{\xi}_p(t)$ converges, after a transient regime over a time-scale of order $1/\kappa_b$, to $\tilde{\xi}_p(t) = \xi_p e^{-i\omega_p t}$ with

$$\xi_p = -i\frac{\epsilon_p}{\kappa_b/2 + i(\tilde{\omega}_b - \omega_p)}.$$

The Hamiltonian in this new frame is given by

$$
\frac{\widetilde{H}}{\hbar} = \left( \tilde{\omega}_a - \frac{\omega_p + \omega_d}{2} \right) \boldsymbol{a}^\dagger \boldsymbol{a} + (\tilde{\omega}_b - \omega_d) \boldsymbol{b}^\dagger \boldsymbol{b} - \frac{E_J}{\hbar} \left( \cos(\tilde{\varphi}) + \frac{\tilde{\varphi}^2}{2} \right)
$$
$$
+ \left( \epsilon_d e^{-i\omega_d t} + \epsilon_d^* e^{i\omega_d t} \right) \left( e^{-i\omega_d t} \boldsymbol{b} + e^{i\omega_d t} \boldsymbol{b}^\dagger \right),
$$

where

$$
\tilde{\varphi} = \varphi_a \left( e^{-i\frac{\omega_p + \omega_d}{2} t} \boldsymbol{a} + e^{i\frac{\omega_p + \omega_d}{2} t} \boldsymbol{a}^\dagger \right) + \varphi_b \left( e^{-i\omega_d t} \boldsymbol{b} + e^{i\omega_d t} \boldsymbol{b}^\dagger \right) + \varphi_b \left( \tilde{\xi}_p(t) + \tilde{\xi}_p^*(t) \right).
$$

Assuming $|\epsilon_d| \ll \omega_d$, and $\|\tilde{\varphi}\| \ll 1$, we expand the cosine up to fourth order and perform rotating-waves approximation to reach an effective Hamiltonian

$$
H_{\text{eff}} = H_{\text{detuning}} + H_{\text{Kerr}} + H_2, \tag{20}
$$

where

$$
\frac{H_{\text{detuning}}}{\hbar} = \left( \omega_a - \frac{\omega_p + \omega_d}{2} - \chi_{ab}|\xi_p|^2 \right) \boldsymbol{a}^\dagger \boldsymbol{a} + (\omega_b - \omega_d - 2\chi_{aa}|\xi_p|^2) \boldsymbol{b}^\dagger \boldsymbol{b},
$$
$$
\frac{H_{\text{Kerr}}}{\hbar} = -\frac{\chi_{aa}}{2} \boldsymbol{a}^{\dagger 2} \boldsymbol{a}^2 - \frac{\chi_{bb}}{2} \boldsymbol{b}^{\dagger 2} \boldsymbol{b}^2 - \chi_{ab} \boldsymbol{a}^\dagger \boldsymbol{a} \boldsymbol{b}^\dagger \boldsymbol{b},
$$
$$
\frac{H_2}{\hbar} = g_2 \boldsymbol{a}^2 \boldsymbol{b}^\dagger + g_2^* \boldsymbol{a}^{\dagger 2} \boldsymbol{b} + \epsilon_d \boldsymbol{b}^\dagger + \epsilon_d^* \boldsymbol{b}.
$$

Here, the frequencies $\omega_{a,b}$ differ from the bare frequencies $\tilde{\omega}_{a,b}$ by the Lamb shift arising from operator ordering in $H_{\text{Kerr}}$. Moreover these frequencies are further shifted down by a term proportional to $|\xi_p|^2$, which corresponds to the AC Stark shift induced by the pump. The linear dependence of this shift versus the pump power is only a first order approximation where higher order contributions of the cosine potential are neglected. The second term $H_{\text{Kerr}}$ corresponds to self-Kerr and cross-Kerr terms. In a first order approximation, we have:

$$
\chi_{aa} = \frac{E_J}{\hbar} \frac{\varphi_a^4}{2}, \qquad \chi_{bb} = \frac{E_J}{\hbar} \frac{\varphi_b^4}{2}, \qquad \chi_{ab} = \frac{E_J}{\hbar} \varphi_a^2 \varphi_b^2.
$$

Note that, in a higher order approximation, these terms also depend on the pump power and one can observe shifts similar to the AC Stark shift.

Finally, the last term $H_2$ is precisely the Hamiltonian that we are intending to engineer. The second term simply represents the near-resonant driving of the dump mode. The first term of this Hamiltonian models a non-linear interaction between the storage and the dump mode: two photons from the storage mode can swap with a single photon in the dump. This term is induced by the four-wave mixing of the pump, the dump and the storage modes. Up to first oder approximations, the coupling strength is given by

$$
g_2 = \chi_{sr} \frac{\xi_p^*}{2}.
$$

The amplitude and phase of this two-photon interaction is therefore controlled by the amplitude and phase of the pumping drive at frequency $\omega_p$. In particular, it is tempting to think that one can increase this interaction strength linearly with the pump amplitude. While this assumption is true for small enough pump amplitudes, it has been experimentally observed that the scaling rapidly stops to be true (see e.g. [18]). More precisely, as soon as the system is driven too strongly, other terms that are not included in the above approximations dominate the dynamics of the system, therefore limiting the performance of the two-photon process. In

practice, the maximal value of $|\xi_p|$ achieved in experiments [18, 50] has been around 1. Note that in all these experiments $\varphi_b \ll 1$ and therefore a value $|\xi_p|$ of order one, still corresponds to small excursions at the bottom of the cosine potential. One would therefore expect the above assumptions to be still holding.

In a recent theoretical analysis [52] and a parallel experimental validation work [53], we have investigated these limitations. In a numerical analysis which gets out of the scope of the current notes, we argue that such out-of-equilibrium nonlinear systems are easily plagued by complex dynamics leading to instabilities. These instabilities are the quantum reminiscent of the classical chaotic behaviour of driven Josephson junctions studied decades ago. The same study suggests that shunting a Josephson junction in the transmon regime with an appropriate inductance removes such instabilities and enables us to operate over a much larger range of pump powers.

Finally, note that, even in the absence of such instabilities one necessarily has to deal with undesired terms such as the self-Kerr and cross-Kerr terms in the above effective Hamiltonian. As argued in the previous chapter, whenever these undesired terms are weak enough (compared to the effective two-photon dissipation rate), their effect can be neglected. It is therefore important to come up with circuit designs that diminish the strength of such terms. This is the topic of the next section.

## 3.3 Alternative circuits to suppress undesired interactions

As argued in the previous section, the engineering of an effective two-photon interaction (17) with a single Josephson junction is necessarily accompanied with undesired cross-Kerr interaction terms that could potentially limit the performance of the two-photon process. In this section, we propose alternative Josephson circuits that circumvent such a limitation.

*Asymmetric Josephson ring modulator.* Inspired by the design of the Josephson ring modulator [54, 55], which ensures an efficient three-wave mixing, we present here a design which can potentially induce the above two-photon interaction Hamiltonian while avoiding the addition of extra undesirable interactions (this design was proposed in [10]). The Josephson ring modulator (Fig. 9(a)) provides a coupling between the three modes (as presented in Fig. 9(c)) of the form

$$H_{\text{JRM}} = \frac{E_L}{4}\left(\varphi_X^2 + \varphi_Y^2 + \frac{\varphi_Z^2}{2}\right)$$
$$- 4E_J\left[\cos\frac{\varphi_X}{2}\cos\frac{\varphi_Y}{2}\cos\frac{\varphi_Z}{2}\cos\varphi_{\text{ext}} + \sin\frac{\varphi_X}{2}\sin\frac{\varphi_Y}{2}\sin\frac{\varphi_Z}{2}\sin\varphi_{\text{ext}}\right],$$

where $E_L = \phi_0^2/L$, $\varphi_{X,Y,Z} = \Phi_{X,Y,Z}/\phi_0 = \varphi_{X,Y,Z}(a_{X,Y,Z} + a_{X,Y,Z}^\dagger)$ and $\varphi_{\text{ext}} = \phi_{\text{ext}}/\phi_0$ is the dimensionless external flux threading each of the identical four loops of the device (here $\phi_0 = \hbar/2e$ represents the reduced flux quantum). Furthermore, the three spatial mode amplitudes $\varphi_X = \varphi_3 - \varphi_1$, $\Phi_Y = \varphi_4 - \varphi_2$ and $\Phi_Z = \varphi_2 + \varphi_4 - \varphi_1 - \varphi_3$ are gauge invariant orthogonal linear combinations of the superconducting phases of the four nodes of the ring (Fig. 9(c)).

In the same manner the design of Fig. 9(b), for a dimensionless external flux of $\varphi_{\text{ext}} = \pi/4$ on the small loops and $3\varphi_{\text{ext}} = 3\pi/4$ on the big loops, induces an effective interaction Hamiltonian of the form

$$H'_{\text{JRM}} = \frac{E_L}{4}\left(\varphi_X^2 + \varphi_Y^2 + \frac{\varphi_Z^2}{2}\right) - 2\sqrt{2}E_J\sin\frac{\varphi_X}{2}\sin\frac{\varphi_Y}{2}\left[\sin\frac{\varphi_Z}{2} + \cos\frac{\varphi_Z}{2}\right]. \tag{21}$$

Similarly to [55], by decreasing the inductances $L$ and therefore increasing the associated $E_L$, one can keep the three modes of the device stable for such a choice of external fluxes. This however comes at the expense of diluting the nonlinearity.

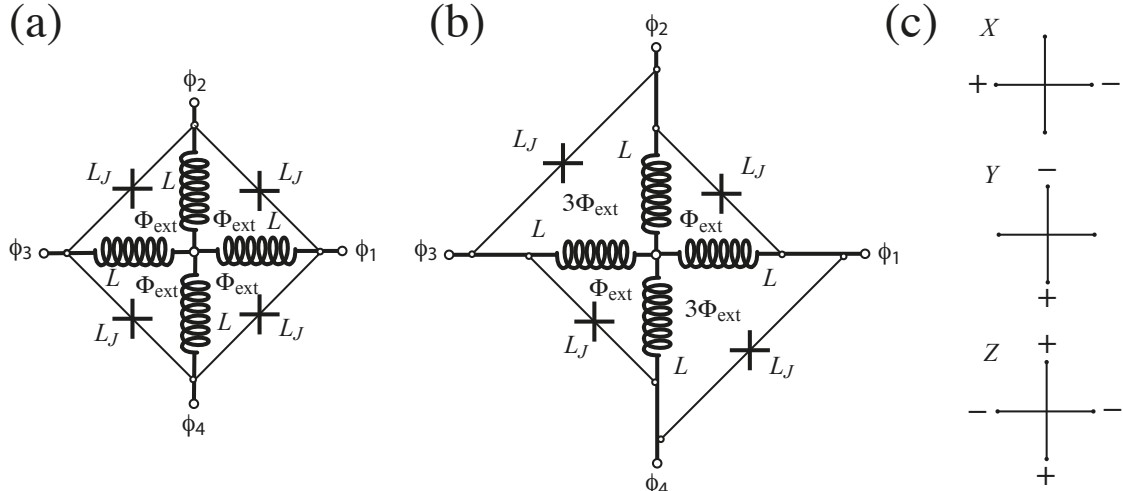

Figure 9: Josephson ring modulators (JRM) providing desired interactions between field modes. **(a)** JRM developed to ensure quantum limited amplification of a quantum signal or to provide frequency conversion between two modes; The signal and idler are respectively coupled to the $X$ and $Y$ modes, as represented in **(c)** and the pump drive is applied on the $Z$ mode. **(b)** A modification of the JRM to ensure an interaction of the form Eq. (17). Reproduced with permission from Ref. [10].

Now, we couple the $Z$ mode of the device to the high-Q storage mode $\boldsymbol{a}$, its $Y$ mode to the low-Q dump mode $\boldsymbol{b}$ mode, and we drive the $X$ mode by a pump of frequency $2\omega_a - \omega_b$ ($\omega_a$ and $\omega_b$ are the effective frequencies of the modes $\boldsymbol{a}$ and $\boldsymbol{b}$). By expanding the Hamiltonian of Eq. (21) up to fourth order terms in $\varphi = (\varphi_X, \varphi_Y, \varphi_Z)$, the only non-rotating term will be of the form

$$\boldsymbol{H}_{\text{eff}} = -\frac{\sqrt{2}}{16}\sqrt{n_{\text{pump}}}E_J\varphi_Z^2\varphi_Y\varphi_X\left(e^{i\phi_{\text{pump}}}\boldsymbol{a}^2\boldsymbol{b}^\dagger + e^{-i\phi_{\text{pump}}}\boldsymbol{a}^{\dagger 2}\boldsymbol{b}\right),$$

where $\phi_{\text{pump}}$ is the phase of the pump drive and $n_{\text{pump}}$ is the average number of circulating photons at pump frequency [56].

Let us now present another circuit design which has been used in a recent experiment demonstrating for the first time the exponential suppression of bit-flips for dissipatively stabilized cat qubits.

*The Asymmetrically Threaded SQUID (ATS)*. In order to circumvent the limitations of the experiments [18, 50], a novel non-linear circuit element, called the Asymmetrically Threaded SQUID (ATS) was realized in [39] to implement the two-to-one photon conversion Hamiltonian (17). It consists in a SQUID (Superconducting Quantum Interference Device) shunted in its center by a large inductance, thus forming two loops. Before we detail how the two-photon exchange Hamiltonian is obtained from the interaction Hamiltonian between the dump mode and the storage mode, let us recall the properties of the ATS. The potential energy of this element alone depends on only one degree of freedom, the phase $\varphi$ across the inductor, and is given by

$$U(\varphi) = \frac{1}{2}E_{L,b}\varphi^2 - E_{J,1}\cos(\varphi + \varphi_{\text{ext},1}) - E_{J,2}\cos(\varphi + \varphi_{\text{ext},2}).$$

Denoting $E_{J,1} = E_J + \Delta E_J$ and $E_{J,1} = E_J - \Delta E_J$, the potential energy can be written

$$U(\varphi) = \frac{1}{2}E_{L,b}\varphi^2 - 2E_J\cos(\varphi_\Sigma)\cos(\varphi + \varphi_\Delta) + 2\Delta E_J\sin(\varphi_\Sigma)\sin(\varphi + \varphi_\Delta),$$

where $\varphi_\Sigma = \frac{1}{2}(\varphi_{\text{ext},1} + \varphi_{\text{ext},2})$ and $\varphi_\Delta = \frac{1}{2}(\varphi_{\text{ext},1} - \varphi_{\text{ext},2})$. Setting $\varphi_\Sigma = \frac{\pi}{2} + \epsilon(t) = \frac{\pi}{2} + \epsilon_0\cos(\omega_p t)$

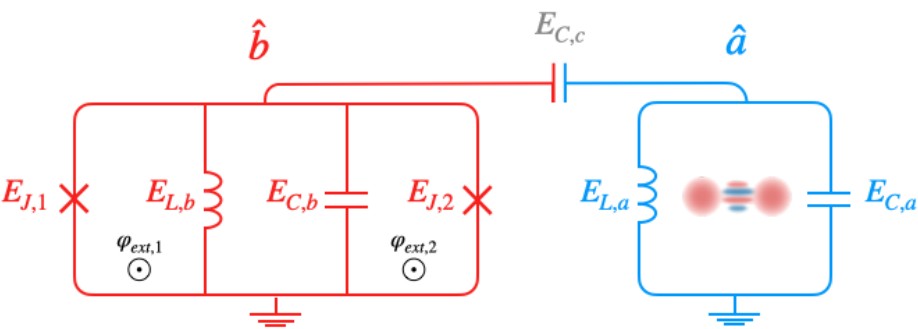

Figure 10: Circuit representation of the low Q mode of an ATS (red) coupled capacitively to a high-Q storage mode $a$ (red) hosting the cat qubit. The ATS is DC biased at the asymmetric flux bias point $\varphi_\Sigma = \frac{1}{2}(\varphi_{\text{ext},1} + \varphi_{\text{ext},1}) = \pi/2 + \epsilon_0 \cos(\omega_p t)$, $\varphi_\Delta = \frac{1}{2}(\varphi_{\text{ext},1} - \varphi_{\text{ext},1}) = \pi/2$ to mediate the required two-photon exchange between the memory and the buffer mode (see main text).

and $\varphi_\Delta = \pi/2$, where $\epsilon_0$ is small, the time-dependent potential energy becomes (to the first order in $\epsilon(t)$)

$$U(\varphi) = \frac{1}{2}E_{L,b}\varphi^2 - 2E_J\epsilon(t)\sin(\varphi) + 2\Delta E_J\cos(\varphi).$$

Now, because of the shunt, this potential is unbounded which prevents the system to escape to unconfined state like in the case of an unshunted Josephson junction. In practice, the inductance can be replaced by an array of N Josephson junctions for which the potential energy $\frac{1}{2}E_{L,b}\varphi^2$ is replaced by $NE_{J,L}\cos(\varphi/N)$ where $E_{J,L}$ is the Josephson of each individual junction of the array; which is now a bounded potential, but for which the number of junctions N can be chosen such that the depth of the potential is high enough ($NE_{J,L} \gg 2E_J\epsilon_0$) so that the state remains confined in the parabolic part of the cosine potential.

The two modes we consider are a high Q mode (the storage $a$) and the (low Q) mode of the ATS. When these two modes are coupled using the ATS as in Figure 10, and a weak resonant drive is applied on the buffer, the Hamiltonian governing the dynamics is given by

$$H/\hbar = \omega_a a^\dagger a + \omega_b b^\dagger b - 2E_J\epsilon(t)\sin(\varphi) + (\epsilon_d e^{-i\omega_d t} + \epsilon_d^* e^{i\omega_d t})(b + b^\dagger),$$

where $\varphi = \varphi_a + \varphi_b = \varphi_a(a + a^\dagger) + \varphi_b(b + b^\dagger)$ is the global phase across the ATS dipole, written as the sum of the phase across each of the two modes $a$ and $b$ weighted by the contribution $\varphi_{a,b}$ of each mode to the zero point fluctuations of $\varphi$.

Expanding the sine up to the third order, the Hamiltonian reads

$$H/\hbar = \omega_a a^\dagger a + \omega_b b^\dagger b - 2E_J\epsilon(t)\varphi_a - 2E_J\epsilon(t)\varphi_b + \frac{1}{3}E_J\epsilon(t)(\varphi_a + \varphi_b)^3$$
$$+ \left(\epsilon_d e^{-i\omega_d t} + \epsilon_d^* e^{i\omega_d t}\right)(b + b^\dagger).$$

The rest of the derivation proceeds as in the case of the single junction. By going to the frame displaced by $\xi_a(t) = \xi_a e^{-i\omega_p t}$ for $a$ and $\xi_b(t) = \xi_b e^{-i\omega_p t}$ for $b$, to the frame rotating at frequency $(\omega_p + \omega_d)/2$ for $a$ and $\omega_d$ for $b$ and keeping only the non rotating terms, the Hamiltonian reads

$$H = H_{\text{shift}} + H_{\text{int}},$$

where

$$H_{\text{shift}} = (\bar{\omega}_b - \omega_d - \Delta_b)b^\dagger b + \left(\bar{\omega}_a - \frac{\omega_p + \omega_d}{2} - \Delta_a\right)a^\dagger a,$$

$$H_{\text{int}} = g_2^* a^2 b^\dagger + g_2 a^{\dagger 2} b + \epsilon_d b^\dagger + \epsilon_d^* b.$$

The AC Stark Shift induced by the pump is now given by $\Delta_{a,b}/\hbar = \frac{1}{3}E_J\varphi_{a,b}^2(\Re(\xi_a)\varphi_a) + \Re(\xi_b)\varphi_b$ and the coupling strength is given by $\hbar g_2 = \frac{1}{2}E_J\epsilon_0\varphi_a^2\varphi_b$.

Crucially, unlike in the case of a single Josephson junction, the only leading order non-rotating term is precisely the desired Hamiltonian, without all the Kerr-like terms that resulted in spurious effects. This led to the first observation of the exponential suppression of the bit-flip error rate with the size of the cat qubit, at the cost of a linear increase of the phase-flip error rate [39]. More precisely, the parameters achieved in this experiment are $\kappa_1/2\pi = 53\text{kHz}$ and $\kappa_2/2\pi = 40\text{kHz}$, thus the ratio is of the two rates of dissipation is of order 1. This ratio is expected to be greatly improved by using a long-lived 3D cavity for the storage mode. However, because of the absence of Kerr-terms, it was possible to witness the exponential increase of lifetime with respect to bit-flip errors: for each added photon in the cat qubit, the bit-flip time is multiplied by 4.2, up to a point where it saturates around 1ms (300 times longer than the $T_1 = 3\mu s$). In this experiment, the saturation of the bit-flip exponential suppression was caused by the thermal occupation of the transmon used for the Wigner tomography of the cat qubit, which contaminated the storage mode through the dispersive coupling between the transmon and the cavity mode used for the readout.

## 3.4 Two-photon dissipation: Adiabatic elimination

In previous sections, we showed how to engineer an interaction between two modes $a$ and $b$ to effectively achieve a master equation of the form (18). As stated in Section 3.1, this master equation can also be written in the following form

$$\frac{d}{dt}\rho = -i\left[g_2(a^2 - \alpha^2)b^\dagger + g_2^*(a^2 - \alpha^2)^\dagger b, \rho\right] + \kappa_b\mathcal{D}[b]\rho, \qquad \alpha = \sqrt{-\frac{\epsilon_d}{g_2}}. \qquad (22)$$

In this section, assuming $|g_2| \ll \kappa_b$ we will show how one can perform an adiabatic elimination of the mode $b$ to effectively achieve the master equation (5) for the mode $a$. This is the two-photon driven dissipative process that we were looking to achieve.

One way is to pursue the approach of [57, Section 12.1]. Calling

$$\varepsilon = |g_2|/\kappa_b \ll 1,$$

the above master equation is of the following form

$$\frac{d}{dt}\rho = -i\varepsilon\left[\frac{H_{\text{int}}}{\hbar}, \rho\right] + \kappa_b\mathcal{D}[b]\rho,$$

Here, restricting the dynamics to a finite dimensional subspace of the harmonic oscillator's Hilbert space, the Hamiltonian $H_{\text{int}}/\hbar$ admits a norm of order $\kappa_b$, similar to the dissipation rate. The idea consists in taking as ansatz the solution of the form

$$\rho = \rho_{00} \otimes |0_b\rangle\langle 0_b| + \varepsilon(\rho_{01} \otimes |0_b\rangle\langle 1_b| + \rho_{10} \otimes |1_b\rangle\langle 1_b|)$$
$$+ \varepsilon^2(\rho_{11} \otimes |1_b\rangle\langle 1_b| + \rho_{02} \otimes |0_b\rangle\langle 2_b| + \rho_{20} \otimes |2_b\rangle\langle 0_b|) + \mathcal{O}(\varepsilon^3). \qquad (23)$$

Here, the operators $\rho_{jk}$ live on the Hilbert space of the mode $a$ and we are interested in the reduced dynamics of

$$\rho_s = \text{tr}_b(\rho) = \rho_{00} + \varepsilon^2\rho_{11} + \mathcal{O}(\varepsilon^3).$$

We can also perform this adiabatic elimination in a more systematic way that enables us to apply it to other similar bipartite systems (we borrow the recipe proposed in [58]). Indeed, we are dealing with a bipartite quantum system composed of two subsystems A and B. The subsystem B is strongly dissipative and the subsystems A and B are weakly coupled through the Hamiltonian interaction.

In a general form, we have the following type of dynamics:

$$\frac{d}{dt}\boldsymbol{\rho} = -i\varepsilon[\boldsymbol{H}_{\text{int}}, \boldsymbol{\rho}] + \mathcal{L}_B(\boldsymbol{\rho}), \tag{24}$$

where $\mathcal{L}_B$ is a Lindblad super-operator acting only on the Hilbert space $\mathcal{H}_B$. Also the two systems A and B are weakly coupled through the interaction Hamiltonian $\boldsymbol{H}_{\text{int}}$. For $\varepsilon = 0$, the solutions stay separable for all times, namely for $\boldsymbol{\rho}(0) = \boldsymbol{\rho}_A \otimes \boldsymbol{\rho}_B$, we have $\boldsymbol{\rho}(t) = \boldsymbol{\rho}_A(0) \otimes \boldsymbol{\rho}_B(t)$. Here, we assume that the Lindbladian $\mathcal{L}_B$ is strongly dissipative and relaxes fast to a unique steady state $\bar{\boldsymbol{\rho}}_B$.

We seek a solution summarizing the effect of the coupling by viewing it as a perturbation in $\varepsilon$ on this uncoupled situation. This perturbation should leave an invariant subspace of the same dimensionality as $\mathcal{H}_B$, and we postulate to model it by a density operator $\boldsymbol{\rho}_s$ on $\mathcal{H}_s$. The postulate is thus that, beyond linear systems perturbation, the reduced dynamics for the perturbed system can satisfy the structure of a quantum system, with dynamics

$$\frac{d}{dt}\boldsymbol{\rho}_s = \mathcal{L}_{s,\varepsilon}(\rho_s), \tag{25}$$

and embedded in the overall system via a Kraus map (completely positive trace-preserving map)

$$\boldsymbol{\rho} = \mathbb{K}_\varepsilon(\boldsymbol{\rho}_s) = \sum_\mu \boldsymbol{M}_\mu \boldsymbol{\rho}_s \boldsymbol{M}_\mu^\dagger. \tag{26}$$

This will indeed be a solution of the system if it satisfies the invariance equation:

$$\mathcal{L}(\mathbb{K}_\varepsilon(\boldsymbol{\rho}_s)) = \mathbb{K}_\varepsilon(\mathcal{L}_{s,\varepsilon}(\boldsymbol{\rho}_s)) \quad \text{for all } \boldsymbol{\rho}_s. \tag{27}$$

Solving (25)-(27) exactly is difficult in general, so we consider a series expansion in $\varepsilon$. Writing

$$\mathcal{L}_{s,\varepsilon}(\boldsymbol{\rho}_s) = \sum_{k=1}^\infty \varepsilon^k \tilde{\mathcal{L}}_{s,k}(\boldsymbol{\rho}_s),$$

$$\mathbb{K}_\varepsilon(\boldsymbol{\rho}_s) = \sum_{k=0}^\infty \varepsilon^k \mathbb{K}_k(\boldsymbol{\rho}_s),$$

we plug this into the invariance equation (27) and identify terms of equal powers in $\varepsilon$. Solving the resulting equations up to some order $\varepsilon^k$ then provides the relevant approximation.

A challenge in this method is to prove that the resulting finite sums of linear superoperators, indeed take a Lindblad equation and Kraus map form respectively. Note that the $\mathcal{L}_k$ and $\mathbb{K}_k$ individually are not imposed Lindbladian and Kraus maps; e.g. for the sum to be trace-preserving, the $\mathbb{K}_k$ for $k > 0$ must actually provide a zero trace. For this it can be necessary to exploit some freedom in the coordinate choice left by (25),(26), depending on the system at hand.

In this section, we provide a recipe to find at least the first terms of such a series. We start by writing the interaction Hamiltonian in the generic form

$$\boldsymbol{H}_{\text{int}} = \sum_r \boldsymbol{A}_r \otimes \boldsymbol{B}_r^\dagger, \tag{28}$$

where $A_r$ and $B_r$ are Hamiltonians acting on Hilbert spaces $\mathcal{H}_A$ and $\mathcal{H}_B$ respectively. We also remind that the unique steady state of the Lindbladian $\mathcal{L}_B$ is assumed to be $\bar{\rho}_B$. In the particular case of the two photon exchange Hamiltonian, we have

$$A_1 = \sqrt{\kappa_b}e^{i\phi}(a^2 - \alpha^2), \quad A_2 = \sqrt{\kappa_b}e^{-i\phi}(a^2 - \alpha^2)^\dagger, \quad B_1 = \sqrt{\kappa_b}e^{i\phi}b, \quad B_2 = \sqrt{\kappa_b}e^{-i\phi}b^\dagger,$$

where $e^{i\phi} = g_2/|g_2|$.

Also the Lindbladian is simply given by

$$\mathcal{L}_B(\rho_B) = \kappa_b \mathcal{D}[b]\rho_B,$$

and therefore its unique steady state is the vacuum state of the mode $b$:

$$\bar{\rho}_B = |0_B\rangle\langle 0_B|.$$

*First order reduced model.* The first order dynamics is purely Hamiltonian:

$$\mathcal{L}_{s,\varepsilon}(\rho_s) = -i\varepsilon[H_{s,1}, \rho_s] + \mathcal{O}(\varepsilon^2).$$

An effective Hamiltonian is calculated as

$$H_{s,1} := \sum_r \mathrm{Tr}\left(B_r\bar{\rho}_B\right)A_r. \tag{29}$$

In the case of the two-photon process, this gives $H_{s,1} = 0$.

*Second order reduced model.* It is computed by pursuing the following steps:

1. (Bottleneck step) for $r = 1$ to $\bar{r}$, find $R_r$ the unique solution of

$$\mathcal{L}_B(R_r) = i(B_r\bar{\rho}_B - \mathrm{Tr}\left(B_r\bar{\rho}_B\right)\bar{\rho}_B), \quad \text{with} \quad \mathrm{Tr}(R_r) = 0.$$

   This operator $R_r$ can always be written in the form $K_r\bar{\rho}_B$ but not in a unique manner.

2. Compute $X$ and $Y$ the matrices with entries

$$(X)_{r,r'} = i\,\mathrm{Tr}\left(B_r^\dagger R_{r'} - R_r^\dagger B_{r'}\right) \text{ and } (Y)_{r,r'} = \frac{1}{2}\mathrm{Tr}\left(B_r^\dagger R_{r'} + R_r^\dagger B_{r'}\right).$$

   The matrix $X$ is semi-definite positive and, we take $\Lambda$ any matrix satisfying $X = \Lambda\Lambda^\dagger$.

3. Compute

$$\begin{aligned}
H_{s,1} &= \sum_r \mathrm{Tr}\left(B_r\bar{\rho}_B\right)A_r^\dagger, \\
H_{s,2} &= \sum_{r,r'}(Y)_{r,r'}A_rA_{r'}^\dagger, \\
L_{s,2}^r &= \sum_{r'}(\Lambda)_{r',r}^\dagger A_{r'}^\dagger.
\end{aligned} \tag{30}$$

The second order reduced Lindbladian is given by

$$\mathcal{L}_{s,\varepsilon}(\rho_s) = -i\varepsilon[H_{s,1}, \rho_s] - i\varepsilon^2[H_{s,2}, \rho_s] + \varepsilon^2\sum_r \mathcal{D}[L_{s,2}^r]\rho_s + \mathcal{O}(\varepsilon^3). \tag{31}$$

Let us apply this algorithm to the case of the two-photon process.

1. Taking $\boldsymbol{B}_1 = \sqrt{\kappa_b}e^{i\phi}\boldsymbol{b}$ and $\bar{\boldsymbol{\rho}}_B = |0\rangle\langle 0|$, we have

$$\mathcal{L}_B(\boldsymbol{R}_1) = 0, \quad \text{with} \quad \boldsymbol{R}_1 = \boldsymbol{K}_1|0\rangle\langle 0|, \ \langle 0|\boldsymbol{R}_1|0\rangle = 0.$$

The unique solution to this equation is $\boldsymbol{R}_1 = 0$.

Also taking $\boldsymbol{B}_2 = \sqrt{\kappa_b}e^{-i\phi}\boldsymbol{b}^\dagger$, we have

$$\mathcal{L}_B(\boldsymbol{R}_2) = i\sqrt{\kappa_b}e^{-i\phi}|1\rangle\langle 0|, \quad \text{with} \quad \boldsymbol{R}_2 = \boldsymbol{K}_2|0\rangle\langle 0|, \ \langle 0|\boldsymbol{R}_2|0\rangle = 0.$$

The unique solution to this equation is

$$\boldsymbol{R}_2 = -\frac{2i}{\sqrt{\kappa_b}}e^{-i\phi}|1\rangle\langle 0|.$$

2. We have

$$X = \begin{pmatrix} 0 & 0 \\ 0 & 4 \end{pmatrix}, \qquad Y = \begin{pmatrix} 0 & 0 \\ 0 & 0 \end{pmatrix}.$$

We choose $\Lambda = \begin{pmatrix} 0 & 0 \\ 0 & 2 \end{pmatrix}$.

3. We have

$$\begin{aligned}
&\boldsymbol{H}_{s,1} = \boldsymbol{H}_{s,2} = 0, \\
&\boldsymbol{L}_{s,2}^1 = 0, \\
&\boldsymbol{L}_{s,2}^2 = 2\boldsymbol{A}_2^\dagger = 2\sqrt{\kappa_b}e^{i\phi}(\boldsymbol{a}^2 - \alpha^2).
\end{aligned}$$

The second order Lindblad equation is therefore given by

$$\frac{d}{dt}\boldsymbol{\rho}_s = 4\varepsilon^2\kappa_b \mathcal{D}[\boldsymbol{a}^2 - \alpha^2]\boldsymbol{\rho}_s.$$

This is the driven two-photon process with an effective two-photon dissipation rate

$$\kappa_2 \approx 4\varepsilon^2\kappa_b = 4\frac{g_2^2}{\kappa_b}.$$

The calculations for higher order terms are much more involved and here we skip them. It is however important to answer one question. Is the bit-flip suppression induced by the two-photon process affected by higher order terms in this adiabatic elimination? The answer fortunately is no. In order to see this, one can note that the states $\{|\pm\alpha\rangle \otimes |0\rangle\}$ are precise steady states of the two-mode system (22). Indeed, the protection of the coherent states $|\pm\alpha\rangle$ against local shifts is ensured at all orders because of this stability. The rate of convergence to these states (which can be seen as the rate of protection against such excursions in the phase space) can however be modified when considering higher-order terms. Note that it is important that such a convergence (correction) rate exceeds the diffusion rate induced by local error mechanisms. Let us discuss this convergence rate and higher-order corrections to it through the following few paragraphs.

One can consider the Lyapunov function [59]

$$\mathcal{V}(\boldsymbol{\rho}_s) = \text{Tr}\left((\boldsymbol{a}^2 - \alpha^2)\boldsymbol{\rho}_s(\boldsymbol{a}^2 - \alpha^2)^\dagger\right).$$

It is clear that $\boldsymbol{V}(\boldsymbol{\rho}_s) = 0$ if and only if $\boldsymbol{\rho}_s \in \text{span}\{|\pm\alpha\rangle\}$. Considering the above second-order reduced dynamics, it was shown in [59] that

$$\frac{d}{dt}\mathcal{V}(\boldsymbol{\rho}_s) \leq -8\varepsilon^2\kappa_b \mathcal{V}(\boldsymbol{\rho}_s).$$

More strongly, simple calculations show that for a coherent state $\boldsymbol{\rho}_s = |\beta\rangle\langle\beta|$, we have

$$\frac{d}{dt}\mathcal{V}(\boldsymbol{\rho}_s) \approx -8\varepsilon^2\kappa_b\left(1 + 2|\beta|^2\right)\mathcal{V}(\boldsymbol{\rho}_s).$$

Therefore, locally around the states $|\pm\alpha\rangle$, the convergence rate is given by $8\varepsilon^2\kappa_b(2|\alpha|^2 + 1)$. The question is if this convergence rate is significantly modified at higher orders. The higher order calculations are complicated. Such calculations up to fourth order have been recently performed in [60]. Skipping the details, here we provide the reduced master equation:

$$\frac{d}{dt}\boldsymbol{\rho}_s = 4\varepsilon^2\kappa_b\mathcal{D}\left[\left(1 - 2\varepsilon^2(A^\dagger A + AA^\dagger)\right)A\right]\boldsymbol{\rho}_s + 32\varepsilon^4\kappa_b\mathcal{D}[A^2]\boldsymbol{\rho}_s + \mathcal{O}(\varepsilon^5), \qquad A = \boldsymbol{a}^2 - \alpha^2.$$

First, we note that, as expected, the two coherent states $|\pm\alpha\rangle$ are still steady states of this system. Considering the same Lyapunov function and considering $\boldsymbol{\rho}_s = |\beta\rangle\langle\beta|$, we have

$$\frac{d}{dt}\mathcal{V}(\boldsymbol{\rho}_s) \approx -8\varepsilon^2\kappa_b(1 + 2|\beta|^2)\left(1 - 4\varepsilon^2(1 + 2|\beta|^2)\right)\mathcal{V}(\boldsymbol{\rho}_s).$$

Therefore the above local convergence rate reduces by a factor of $\left(1 - 4\varepsilon^2(1 + 2|\alpha|^2)\right)$. This calculation provides the insight that, fixing a maximum amplitude $|\alpha_{\max}|$, we need to design the parameters in a deep enough adiabatic regime. More precisely, we need to ensure that $4\varepsilon^2(1 + 2|\alpha_{\max}|^2)$ remains small with respect to 1.

## 4 Bias-preserving gates

The purpose of this section is to describe how the quantum bit of information stored in a cat qubit can be processed. As mentioned earlier, in these notes we focus on the potential of cat qubits as biased noise qubits. It is therefore crucial that the noise bias remains preserved *during* the processing of the information encoded in the cat qubit. There are actually two distinct conditions to fulfil to successfully design a bias-preserving operation, that are discussed in the next subsections.

### 4.1 Bias-preserving gates: two conditions to satisfy

The first condition concerns the operation itself, as operators that *convert* the $Z$ operator into an $X$ (or $Y$) operators are automatically non bias-preserving. The textbook example of such an operation is the Hadamard gate $H$, that converts a Pauli $Z$ operator into $X$ (and vice-versa). Applying a Hadamard gate to a cat qubit converts a phase-flip error that occurs just before the gate to a bit-flip error. Thus, even though bit-flips occur with an exponentially small probability, the application of a Hadamard gate on a cat qubit re-introduces bit-flips with a probability that is similar to the phase-flip error probability. This observation can be generalized to other gates such as the $S$ gate or the controlled-$H$ gate, etc. The gates that commute with the $Z$ operator are readily acceptable candidates. Gates that do not commute with $Z$ may still be acceptable, as long as the error produced by the propagation of the $Z$ error through the gate remains of the phase-flip type. Note that in this regard, we only require that $Z$ errors are not converted to other types of errors, while $X$ or $Y$ errors that occur with exponentially small probability can be converted to other types of errors.

*Single-qubit gates.* Consider the case of a unitary operator $U$ acting on a single qubit. For the purpose of this discussion, one can disregard the global phase and identify the unitary to a rotation on the Bloch sphere of an angle $\theta$ around the axis specified by the real valued unitary vector $\vec{n} = (n_x, n_y, n_z)$

$$U = \mathcal{R}_{\vec{n}}(\theta) = e^{-i\frac{\theta}{2}\vec{n}\cdot\vec{\sigma}} = \cos\frac{\theta}{2} - i\sin\frac{\theta}{2}(n_x X + n_y Y + n_z Z).$$

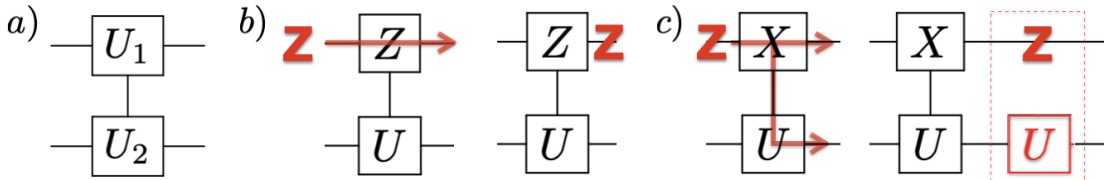

Figure 11: a) Circuit representation of a general $U_1$-controlled-$U_2$ gate. (b-c) A Pauli $Z$ error commutes with a $Z$-controlled-$U$ operation, but produces an additional $U$ error by propagating through an $X$-controlled-$U$ gate.

where $\vec{\sigma} = (X, Y, Z)$. A phase-flip error $Z$ that occurred before the gate $\mathcal{R}_{\vec{n}}(\theta)$ propagates through the gate as a $Z$ error together with an additional unitary error $\mathcal{E}_Z(\vec{n}, \theta)$

$$\mathcal{R}_{\vec{n}}(\theta)Z = Z\mathcal{E}_Z(\vec{n}, \theta)\mathcal{R}_{\vec{n}}(\theta),$$

where the additional error $\mathcal{E}_Z(\vec{n}, \theta)$ is given by

$$\mathcal{E}_Z(\vec{n}, \theta) = \left(\cos\theta + 2\sin^2\frac{\theta}{2}n_z^2\right)I - i\left(\sin\theta\, n_x + 2\sin^2\frac{\theta}{2}n_y n_z\right)X - i\left(\sin\theta\, n_y - 2\sin^2\frac{\theta}{2}n_x n_z\right)Y.$$

Thus, the unitary $\mathcal{R}_{\vec{n}}(\theta)$ does not convert $Z$ errors into $X$ or $Y$ error if and only if the following conditions are satisfied

$$\begin{cases} \sin\theta\, n_x + 2\sin^2\frac{\theta}{2}n_y n_z = 0, \\ \sin\theta\, n_y - 2\sin^2\frac{\theta}{2}n_x n_z = 0. \end{cases} \tag{32}$$

As one could expect, this includes the rotations around the $Z$ axis of the Bloch sphere of an arbitrary angle $Z(\theta)$, which commute with the $Z$ operator (the $\pi$ rotation around the $Z$-axis) as confirmed by checking the above conditions are satisfied for $n_z = 1$ (and hence $n_x = n_y = 0$) and any angle $\theta$. Note that the set of rotations $\{Z(\theta)\}_{\theta\in[0,2\pi[}$ actually contains gates from arbitrarily high levels of the Clifford hierarchy [61, 62], as one can check that the rotation around a *cardinal axis* $+X, +Y, +Z$ of an angle $\theta \in \{k\pi/2^{n-1}\}_{k\in\mathbb{Z}_{2^n}}$ is in the $n$-th level of the Clifford hierarchy.

The conditions (32) can also be satisfied for unitaries that do not commute with the $Z$ operator. For instance, they are satisfied by the $\pi$-rotations around any axis in the $(X, Y)$ plane, $\theta = \pi$ and $(n_x, n_y, n_z) = \vec{n}_{X,Y}(\varphi) = (\cos(\varphi), \sin(\varphi), 0)$, producing the unitary

$$U = \mathcal{R}_{\vec{n}_{X,Y}(\varphi)}(\pi) = \cos(\varphi)X + \sin(\varphi)Y.$$

Note that this set also contains gates from arbitrary levels of the Clifford hierarchy, following from the fact that $\mathcal{R}_{\vec{n}_{X,Y}(\varphi)}(\pi)$ is in the $(n-1)$-th level if $\varphi \in \{k\pi/2^{n-1}\}_{k\in\mathbb{Z}_{2^n}}$. This can be checked *e.g* from the above fact using the decomposition

$$\mathcal{R}_{\vec{n}_{X,Y}(\varphi)}(\pi) = Z(\varphi)XZ(-\varphi).$$

The conditions (32) also automatically rule out some gates. One can check, as expected, that the Hadamard gate ($\theta = \pi$, $\vec{n} = (1, 0, 1)/\sqrt{2}$) does not match the criteria, or that the only rotations around the $X$ or $Y$ axis that are allowed are those of an angle $\pi$, that is the Pauli rotations.

*Entangling gates.* The same analysis can be carried through for entangling gates. For two qubits (or more generally, two subsystems), an entangling gate $U$ is a gate that cannot be

factorized in the form $U = U_1 \otimes U_2$, where $U_{1,2}$ are gates acting each on one of the two qubits (or two subsystems).

Here, we investigate the bias-preserving compatibility of a specific subset of entangling gates composed of the "controlled" gates. A two-qubit controlled gate is built using two single-qubit unitary operators (different from the identity) $U_1$ and $U_2$, where one of the two (say $U_1$) has to be Hermitian. The resulting entangling gate, called the "$U_1 -$controlled$- U_2$" gate, acts as follows. The unitary operator $U_1$ being Hermitian (and non-trivial), it has exactly two eigenvalues: $\pm 1$. The two associated eigenspaces split the Hilbert space of the first qubit in two. The $U_1$-controlled-$U_2$ consists in applying the unitary $U_2$ to the second qubit, called the *target* qubit, whenever the state of the first qubit, called the *control* qubit, is in the $-1$ eigenspace, and applying identity otherwise. The circuit representation of such a gate is depicted in Figure 11 a). In general, it suffices that only one of the two unitaries $U_1$ or $U_2$ be Hermitian to define such an operation, where the qubit corresponding to the Hermitian unitary is taken as the control qubit. Interestingly, when both unitaries are Hermitian, choosing either one of the qubits as the control qubit produces the same quantum gate. Note that in the literature, a $Z$-controlled-$U$ gate is simply referred to as a controlled-$U$ operation, because the $\pm 1$ eigenstates of the $Z$ operator are the computational $|0\rangle, |1\rangle$ states and the $Z$ control is represented by the symbol $\bullet$ in circuit notation to emphasize the classical analogy. This definition is readily generalized to multi-qubit unitaries.

Here, we focus on a specific subset of multi-qubit controlled gates where the basic unitaries used to construct entangling gates are only $X$ and $Z$ Pauli operators. This includes, for example, the two-qubit "controlled-NOT" gate ($Z$-controlled-$X$), denoted CNOT or CX, the two-qubit "controlled-Z" gate ($Z$-controlled-$Z$) gate, denoted CZ, or the three-qubit Toffoli gate ($Z$-controlled-$Z$-controlled-$X$), denoted CCX.

As with the single-qubit unitaries, we are here interested in two things. First, one can check that an $n$-qubit controlled gate where each of the involved unitaries is a (non-trivial) Pauli operator is in the $n$-th level of the Clifford hierarchy. Consider first the three two-qubit controlled gates that can be formed using $X$ and $Z$ operators

$$\{U_1 - \text{controlled} - U_2, \ U_{1,2} \in \{X, Z\}\}.$$

We are interested in how $Z$ errors propagate through such gates. As $Z$ trivially commutes with $Z$ but anti-commutes with $X$, it is clear that the $\pm 1$ eigenspaces of the $Z$ operator are not disturbed by a $Z$ error, while the $\pm 1$ eigenspaces of $X$ are swapped. Thus, as depicted in Figure 11 (b-c), a $Z$ error acting on a "$Z$-controlled-$U$" commutes with the gate, while a $Z$ error acting on a "$X$-controlled-$U$" produces an additional error $U$ on the corresponding qubit. From this observation, it is clear that a multi-qubit controlled gate built with $X$ and $Z$ operators does not convert $Z$ errors if and only if $U$ is composed of $Z$ operators only. In other words, the eligible gates cannot contain more than a single $X$ control: the CZ gate and the CNOT gate are not forbidden by our bias-preserving definition, while the "$X$-controlled-$X$" gate is.

The same analysis carries through straightforwardly to a higher number of qubits. Using only $X$ and $Z$ operators, it is necessary to use at least three qubits to construct a non-Clifford gate (gates that do not belong to the first two levels of the Clifford hierarchy). Out of the four gates of the form

$$\{U_1 - \text{controlled} - U_2 - \text{controlled} - U_3, \ U_{1,2,3} \in \{X, Z\}\},$$

only those that contain zero (the CCZ) or one (the CCX gate) $X$ operator do not convert $Z$ type errors into $X$ type errors.

The second condition that needs to be fulfilled by a bias-preserving gate in addition to not convert $Z$-type errors into $X$ or $Y$-type errors is that it should be *implementable* in a

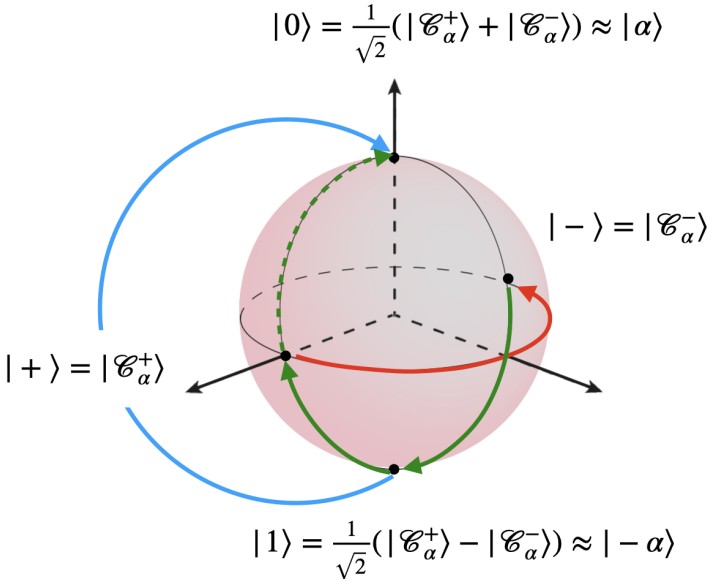

Figure 12: Schematic illustration of the no-go theorem for a bias-preserving X gate on a two-level system, and of the trick used to work around this no-go in the case of the cat qubit.

bias-preserving manner. While the first condition is agnostic to the specific technology implementing the qubit but rather only depends on the structure of the unitary itself, this second condition can only be checked on the description of the process actually implementing the gate on a given physical platform. Let us consider, for instance, rotations around the $X$ (or $Y$ axis). As discussed above, only the $\pi$-rotation around the $X$ axis is a viable candidate for a bias-preserving implementation. The authors of [63] rightfully noted that the structure of the noise *induced* by the implementation of the gate may have no reason to be highly biased: in the case of a $\pi$-rotation around the $X$-axis, for instance, the effect of a slight over-rotation or under-rotation may introduce an error proportional to $X$ rather than $Z$, thus re-introducing bit-flip errors that are not exponentially unlikely.

We argue in the next subsections that all the candidates introduced above can indeed be implemented in a bias-preserving manner on cat qubits. Because the exponential bias in the noise structure comes from the distance in the phase-space between the two computational states, one general guiding principle that needs to be followed when designing such implementations is that this distance should never be decreased during the process implementing a gate.

This guiding principle is necessary, but sufficient only for the gates that commute with the $Z$ errors *at all times* during the execution of the gates. It has been shown in [10] how such gates, that include arbitrary rotations around the $Z$-axis or the $CZ$ gate, can be implemented using a weak Hamiltonian in presence of the strong two-photon dissipative dynamics. The effect of the weak Hamiltonian implementing the gate is to induce a slow evolution in the two-dimensional stable manifold of the cat qubits. The precise bias-preserving implementation of these gates is discussed in subsection 4.2.

The gates that do not commute with the $Z$ error pose additional challenges, and were usually discarded from general hardware agnostic studies of computing with biased noise qubits (see *e.g* [63, 64]). Indeed, considering again the $\pi$-rotation around the $X$-axis of the Bloch sphere, it is actually *impossible* to design a bias-preserving implementation without leaving

the code subspace. The intuition of why an X gate cannot be performed in a bias-preserving manner without leaving the code space is illustrated in Figure 12, taken from [65]. A continuous process that rotates the state $|-\alpha\rangle$ to $|\alpha\rangle$ (and vice-versa) without leaving the code space takes the state through a path on the surface of the Bloch sphere (green arrows in Figure 12). If a phase-flips occurs, say, at the middle of this evolution (red arrow), then the remaining part of the process (green arrow) results in a bit-flip after the gate is executed. The way around this no-go theorem is to rather design an implementation that takes the state outside the code space (blue arrow) during the whole evolution, in such a way that the errors that occur during the evolution cannot introduce bit-flip errors. The idea was originally introduced to perform a bias-preserving CNOT gate in the context of Kerr-cat qubits [66]. The gates that are implemented in this manner are also discussed in subsection 4.2. The rest of the section is organized as follows.

In subsection 4.2, we describe the precise implementations of all the operations in the set

$$\mathcal{S}' = \{\mathcal{P}_{|+\rangle}, \mathcal{P}_{|0\rangle}, \mathcal{M}_X, \mathcal{M}_Z\} \cup \{Z(\theta), ZZ(\theta), ZZZ(\theta)\} \cup \{X, \mathrm{CX}, \mathrm{SWAP}, \mathrm{CCX}\},$$

with a particular focus on the bias-preserving property of the implementation. This set is split in three subsets, corresponding to three different ways to achieve a bias-preserving implementation, and discussed separately in subsection 4.2. The first one concerns state preparation and measurement. Here, the bias-preserving property is either trivial or comes from considerations very specific to the realization of the operation. The second subset contains the gates that are realized through the quantum Zeno effect, by using a weak Hamiltonian that triggers the accumulation of the desired "dynamical phase" in the cat qubit subspace. Here, the bias-preserving property is ensured by the fact that phase-flips commutes with the continuous process implementing the gate, and by the fact that the two-photon dissipation is always turned on. The last subset is composed of the gates that are implemented using a continuous deformation of the code space that impart a topological $\pi$ phase around the $X$ axis of the Bloch sphere. The bias-preserving implementation of these gates is decomposed in two parts: first, the two-photon dissipative scheme is made time-dependent, and for multi-qubit gates, conditional, in order to implement the required code deformation. Additionally, we argue that the fidelity of these gates is greatly improved by adding a Hamiltonian during the gate execution. We emphasize that these (optional) Hamiltonians are not required for the gate implementation, nor for the bias-preserving property of the gates, but merely to greatly reduce the phase-flip errors induced by the non-adiabaticity (finite time) of these gates.

Then, in subsection 4.3, we either give or derive explicitly analytical error models for the dominant phase-flip error probabilities, As will become clear upon inspection of the error models, the phase-flip errors occurring during the execution of the gates come from two different sources that we both take into account. The first are the phase-flip errors induced by the main error channel of the quantum harmonic oscillator, namely the photon loss, characterized by the single photon dissipation rate $\kappa_1$. The second source of phase-flip errors is the finite time of the gates.

Last, in subsection 4.4, we discuss how all the proposed implementations can be realized within the framework of circuit QED. We describe how the weak Hamiltonians required for the Zeno gates have been realized [50] or could be realized. Last, we discuss the realization of the topological gates. This can be split in two parts: the (required) implementation of the time dependence of the two-pumping scheme that realized the topological deformation of the cat qubit code space, and the (optional) implementation of the feed-forward Hamiltonians that might be added during the gates execution to reduce the phase-flip errors induced by non-adiabaticity.

## 4.2 Bias-preserving implementations

**State preparation and measurement.**

*Measurement of the X operator.* The only measurement on the cat qubit required in the construction of the scheme is the measurement of the $X$ operator, whose eigenstates are the cat states $|\mathcal{C}_\alpha^\pm\rangle$. Because these states have a well-defined photon-number parity, the measurement of $X$ can be realized by a photon-number parity measurement. Here, the "bias-preserving" property of this operation is trivially ensured by the fact that, every time an $X$ measurement is needed in our circuit, it is performed on an ancilla qubit whose state is discarded after the measurement and prepared again in a fresh state. The measurement of the $X$ operator could be either destructive or quantum non-demolition (QND) as it is only used on ancilla cat qubits, which are discarded after the measurement, except at the very end of the execution of the quantum algorithm where the data cat qubits are also measured (destructively) to get the output of the algorithm. The QND parity measurement proposed in [67] and realized in [31,68] is perfectly suitable for our scheme. The main idea behind this protocol is to couple to an ancilla (transmon) qubit to the mode whose photon-number parity is to be measured via the dispersive interaction Hamiltonian

$$H = -\chi |e\rangle\langle e| a^\dagger a.$$

The unitary evolution generated by this Hamiltonian on a time interval $T = \pi/\chi$ is given by

$$U = |g\rangle\langle g| I + |e\rangle\langle e| e^{i\pi a^\dagger a},$$

entangling the state of the ancilla with the parity of the state of the cavity. Preparing the ancilla qubit in a superposition state $|+\rangle = \frac{1}{\sqrt{2}}(|g\rangle + |e\rangle)$, the effect of the unitary $U$ is to flip the ancilla to the state $|-\rangle = \frac{1}{\sqrt{2}}(|g\rangle - |e\rangle)$ when the cavity contains an odd number of photons and to leave it unchanged otherwise. A measurement of the $\sigma_x$ operator of the qubit thus reveals the parity of the cavity state.

Note that in order to perform such a parity measurement, the two-photon driven dissipation on the measured cat qubit has to be turned off. However, given that these measurements are performed on ancilla cat qubits that are thrown out after each measurement, the absence of protection during the measurement merely affects the measurement fidelity and does not have any consequence on the rest of the circuit. Fidelities of photon-number parity measurement of about 98.5% have been previously achieved using this protocol [32].

*Measurement of the Z operator.* The measurement of the Pauli $Z$ operator of the cat qubits is not required to obtain a universal set of gates at the logical level [9]. Yet, it might be required to design new logical operations, or to simplify some of the logical circuits. Note that the eigenstates of the $Z$ operator are (exponentially close to) the coherent states $|\pm\alpha\rangle$, such that a destructive measurement of the $Z$ operator can be implemented *e.g* the protocol used to measure the phase of a coherent state of [69].

*Preparation of the cat states $|\mathcal{C}_\alpha^\pm\rangle$.* The preparation of the eigenstates of the $X$ operator is trivially compatible with the noise bias since a bit-flip does not affect these states, as noted in [63]. Indeed, because the cat states $|\mathcal{C}_\alpha^\pm\rangle$ have equal population on the $|\pm\alpha\rangle$ states, the bit-flip operator $X$ cannot modify these population. One way to prepare the even cat state $|+\rangle = |\mathcal{C}_\alpha^+\rangle$ is performed by initializing the quantum harmonic oscillator in the vacuum state $|0\rangle$ and turning on the driven two-photon dissipation [10]. Indeed, the two-photon driven dissipation conserves the photon-number parity, such that unique steady state of the system is given by the even cat state. Such a state preparation has already been realized experimentally [18] and the fidelity of this operation is set by the ratio between the two-photon dissipation rate $\kappa_2$, setting the rate of convergence to the cat state, and the undesired single-photon loss

rate $\kappa_1$, setting the parity jump rates (equivalent to phase-flip errors) mixing the even cat with the odd one. Then, the odd cat state $|-\rangle = |\mathcal{C}_\alpha^-\rangle$ can be prepared from the even cat state by applying the $Z$ described later in this subsection. The preparation of the cat states can also be performed using an active protocol rather than relying on the passive two-photon dissipation. Such protocol, like the mapping of an arbitrary state of a transmon to a cat qubit [70], have the advantage to be faster and to produce states with higher fidelity. A fast and reliable operation that prepares a given state on the cat qubit is immediately followed by the activation of the stabilization scheme. In particular, in the experiment [50], the state $|\mathcal{C}_\alpha^+\rangle$ was generated using optimal control techniques which can significantly improve the fidelity with respect to a passive preparation with two-photon driven dissipation.

Once again in order to construct a universal set of fault-tolerant gates at the logical level, one only requires that the physical cat qubits can be initialized in the states $|\mathcal{C}_\alpha^\pm\rangle$ [9]. However, the preparation of a cat qubit in the coherent state $|0\rangle \approx |\alpha\rangle$ could be useful for further logical circuit implementations.

*Preparation of the coherent state $|\alpha\rangle$.* The eigenstates of the $Z$ operator of the cat qubit are exponentially close to the coherent states $|\pm\alpha\rangle$. A fast and reliable preparation of these states is realized by applying a strong microwave pulse to the oscillator initialized in the vacuum state to generate a displacement $\mathcal{D}(\pm\alpha)$, and to turn on the two-photon driven-dissipative stabilization immediately after the displacement. Note that unlike the cat states $|\mathcal{C}_\alpha^\pm\rangle$, the $|\pm\alpha\rangle$ are not intrinsically robust to bit-flip errors, as the bit-flip error operator induces population transfer between $|\alpha\rangle$ and $|-\alpha\rangle$.

Here, the preparation of the state $|\alpha\rangle$ is only bias-preserving in the sense that the phase of the microwave pulse applied to displace the oscillator state from the vacuum to the coherent state $|\pm\alpha\rangle$ can be made very precise, such that the state of the oscillator after this displacement is in a certain coherent state $|\tilde{\alpha}\rangle$ in the neighbourhood of $|\alpha\rangle$, where $|\tilde{\alpha}\rangle$ is in general slightly different from $|\alpha\rangle$ to account for the small imprecision in the displacement. Then, the two-photon pumping is activated just after the displacement, such that the state $|\tilde{\alpha}\rangle$ relaxes to the coherent state $|\alpha\rangle$. The resulting bit-flip probability (*i.e*, the probability to be in the state $|-\alpha\rangle$ after a displacement $\mathcal{D}(\alpha)$ has been applied to the vacuum) can be very small. Indeed, the population of the state $|-\alpha\rangle$ at the end of this protocol is (roughly) given by the probability that a phase error of at least $\pi$ has occurred in the displacement, which can be sufficiently small. However, because here the "bias-preserving" is ensured solely by the fact that the phase of microwave pulses is very well controlled, it is specific to our circuit QED implementation of the scheme and it is important to check that the probability of a bit-flip error occurring during this protocol is of the same order as the exponentially suppressed bit-flip error of the cat qubit.

**Dynamical phase gates with the Quantum Zeno Effect.**

*$Z(\theta)$ gate.* The $Z(\theta)$ gate is the rotation of an arbitrary angle $\theta$ around the $Z$ axis of the Bloch sphere of the cat qubit:

$$Z(\theta) = e^{-i\frac{\theta}{2}Z_\alpha} = \cos\frac{\theta}{2}I_\alpha - i\sin\frac{\theta}{2}Z_\alpha. \tag{33}$$

It was first proposed in [10] and realized experimentally in [50]. The subscript $\alpha$ in the Pauli operators $I_\alpha$ and $Z_\alpha$ are here to emphasize that these are operators the Pauli operators acting on the cat qubit. They can be expressed as

$$I_\alpha = |\mathcal{C}_\alpha^+\rangle\langle\mathcal{C}_\alpha^+| + |\mathcal{C}_\alpha^-\rangle\langle\mathcal{C}_\alpha^-|,$$
$$Z_\alpha = |\mathcal{C}_\alpha^+\rangle\langle\mathcal{C}_\alpha^-| + |\mathcal{C}_\alpha^-\rangle\langle\mathcal{C}_\alpha^+|.$$

In the rest of this section, the subscript $\alpha$ is dropped and the operators acting on the cat qubit are simply written using the usual qubit notations. The $Z(\theta)$ gate is realized by applying a weak resonant drive described (in the rotating frame of the cavity mode) by the Hamiltonian

$H = \epsilon_Z \boldsymbol{a} + \epsilon_Z^* \boldsymbol{a}^\dagger$ in the presence of the two-photon driven dissipation modelled by the Lindblad super-operator $\kappa_2 \mathcal{D}[\boldsymbol{a}^2 - \alpha^2]$, with $|\epsilon_Z|$ small with respect to $\kappa_2$. The combination of these two dynamics implements the gate as follows. The fast two-photon driven-dissipative part of the dynamics confines the state in the cat qubit manifold, while the single photon drive induces a change of the photon number parity. If the cat qubit is initialized in the state $|\mathcal{C}_\alpha^+\rangle$, of even photon number parity, the effect of the weak drive is to induce Rabi oscillations between the even cat $|\mathcal{C}_\alpha^+\rangle$ and the odd cat $|\mathcal{C}_\alpha^-\rangle$. The rate of these Rabi oscillations is set by the first order perturbation induced by the Hamiltonian, given by the projection of the Hamiltonian on the cat qubit subspace

$$\left(|\mathcal{C}_\alpha^+\rangle\langle\mathcal{C}_\alpha^+| + |\mathcal{C}_\alpha^-\rangle\langle\mathcal{C}_\alpha^-|\right)\left(\epsilon_Z \boldsymbol{a} + \epsilon_Z^* \boldsymbol{a}^\dagger\right)\left(|\mathcal{C}_\alpha^+\rangle\langle\mathcal{C}_\alpha^+| + |\mathcal{C}_\alpha^-\rangle\langle\mathcal{C}_\alpha^-|\right) = 2\Re[\alpha\epsilon_Z]Z + \mathcal{O}(e^{-2|\alpha|^2}).$$

The fact that the effective dynamics, up to the first order in the small parameter $\epsilon_Z/\kappa_2$, is given by the projection of the perturbative Hamiltonian is the well known quantum Zeno effect. A rigorous mathematical derivation proving this fact can be found *e.g* in [71].

The oscillation rate is maximized when the phase of the drive is opposite to the phase of $\alpha$ such that $\alpha\epsilon_Z$ is a real number, and the rotation of an angle $\theta$ is obtained by applying the weak drive during a time

$$T = \frac{\theta}{4\alpha\epsilon_Z} = \frac{\theta}{4\sqrt{\bar{n}}|\epsilon_Z|}.$$

*$ZZ(\theta)$ gate and CZ gate.* The same recipe can be readily applied to construct the two qubit entangling gate [10]

$$Z_1 Z_2(\theta) = e^{-i\frac{\theta}{2}Z_1 Z_2} = \cos\frac{\theta}{2}I_1 I_2 - i\sin\frac{\theta}{2}Z_1 Z_2,$$

where the subscript (1,2) label the two cat qubits. This gate is realized by applying a weak beam-splitter Hamiltonian

$$H = \epsilon_{Z_1 Z_2}\boldsymbol{a}_1\boldsymbol{a}_2^\dagger + \epsilon_{Z_1 Z_2}^*\boldsymbol{a}_1^\dagger\boldsymbol{a}_2,$$

in the presence of the two-photon driven dissipation on both of the cat qubits. Here and for all the multi-qubit gates involved in this work, we always assume that the same $\alpha$ is used for the two (or more) cat qubits. This assumption is made for the sole purpose of reducing the number of notations, but this assumption can be relaxed everywhere is this work and all the gates presented can be straightforwardly adapted to cat qubits of different sizes $\alpha$ and $\beta$. Taking $\epsilon_{Z_1 Z_2}$ to be real, the projection of $H$ on the two cat qubit subspaces gives the oscillation rate $\Omega_{Z_1 Z_2} = 2|\alpha|^2\epsilon_{Z_1 Z_2}$ such that the rotation $Z_1 Z_2(\theta)$ is obtained upon the application of the weak Hamiltonian during a time

$$T = \frac{\theta}{4|\alpha|^2\epsilon_{Z_1 Z_2}} = \frac{\theta}{4\bar{n}\epsilon_{Z_1 Z_2}}.$$

The two gates $Z(\theta)$ and $Z_1 Z_2(\theta)$ commute, such that one can be combine them. For instance, noting that a controlled-Z gate can be decomposed as

$$CZ = (-1)^{|11\rangle\langle11|} = e^{-i\frac{\pi}{4}(I_1 - Z_1)(I_2 - Z_2)}$$

and taking $\alpha$ real, the CZ gate is implemented through the Zeno effect by applying the Hamiltonian

$$H = \epsilon_{CZ}\left[-(\boldsymbol{a}_1 + \boldsymbol{a}_1^\dagger + \boldsymbol{a}_2 + \boldsymbol{a}_2^\dagger) + \frac{1}{\sqrt{\bar{n}}}(\boldsymbol{a}_1\boldsymbol{a}_2^\dagger + \boldsymbol{a}_1^\dagger\boldsymbol{a}_2)\right],$$

for a time $T = \pi/(8\sqrt{\bar{n}}\epsilon_{CZ})$.

*ZZZ(θ) gate.* From a theoretical point of view, the above Zeno mechanism can be generalized to construct arbitrary rotations on $n$ qubits. However, the weak Hamiltonian required is of higher order with each added qubit, which makes it increasingly hard to implement, for reasons that we detail in the experimental subsection 4.4. The same trade-off is encountered in the case of the topological gates introduced next.

For instance, the three qubit entangling gate $ZZZ(\theta)$ (which, combined with $CZ$ and $Z$ gates, can be used to implement *e.g* the CCZ gate) can be generated *e.g* using the weak Hamiltonian between the three cat qubits $a_{1,2,3}$

$$H = \epsilon_{Z_1 Z_2 Z_3} a_1 a_2 a_3^\dagger + \text{h.c.},$$

for a time $T = \dfrac{\theta}{8|\alpha|^3 \epsilon_{Z_1 Z_2 Z_3}}$.

**Topological phase gates with adiabatic code deformation.**

*X gate.* As we have argued in subsection 4.1, the only rotation around the $X$-axis that does not convert $Z$ errors into $X$ or $Y$ errors is the rotation of an angle $\pi$, the Pauli $X$ gate. The problem of the bias-preserving implementation can be roughly stated as such. How can we design a process, having a unitary action on the two-dimensional subspace of the cat qubits, that implements a rotation of the coherent state $|\alpha\rangle$ to the coherent state $|-\alpha\rangle$ (and vice-versa) while ensuring that the physical errors of the quantum harmonic oscillator (*e.g.* photon loss) result in at most an exponentially small population remaining on the state $|\alpha\rangle$ after the transfer is done? As we have seen in subsection 4.1, there is no process that can realize this while keeping the state of the system inside the two-dimensional cat qubits manifold during the gate execution. Rather, this is realized using an excursion outside the code space, that can be thought of as a continuous adiabatic deformation of the code space, obtained by varying the complex number $\alpha$ of the two-photon dissipation $\kappa_2 \mathcal{D}[a^2 - \alpha^2]$ in time. When the variations of $\alpha(t)$ are sufficiently slow with respect to $\kappa_2^{-1}$, the driven-dissipative dynamics modelled by the super-operator $\kappa_2 \mathcal{D}[a^2 - \alpha(t)^2]$ stabilizes the two-dimensional manifold spanned by the coherent states $|\alpha(t)\rangle$ and $|-\alpha(t)\rangle$ at all times t, realizing a slow motion of the fixed points of the dynamics in the phase-space.

Remarkably, the quantum information is preserved while the code space is deformed, provided the two states $|\alpha(t)\rangle, |-\alpha(t)\rangle$ remain sufficiently separated in phase-space at all times. This point is crucial in order to implement a *unitary* operation within the cat qubit manifold. The state $|\psi_0\rangle = c_0|\alpha\rangle + c_1|-\alpha\rangle$ at time $t = 0$ evolves under the effect of $\kappa_2 \mathcal{D}[a^2 - \alpha(t)^2]$, with $\alpha(0) = \alpha$, to

$$|\psi(t)\rangle = c_0|\alpha(t)\rangle + c_1|-\alpha(t)\rangle,$$

provided that at all times intermediate times $t' \in [0, t]$, the two following conditions are satisfied

$$|\dot{\alpha}(t')|/|\alpha(t')| \ll \kappa_2,$$
$$|\langle \alpha(t')|-\alpha(t')\rangle|^2 \ll 1.$$

The $X$ gate is realized by choosing a "path" function $\alpha(t)$ such that $|\alpha\rangle$ and $|-\alpha\rangle$ are swapped, *e.g* $\alpha(t) = \alpha e^{i\pi t/T}$, $t \in [0, T]$ where $T \gg \kappa_2^{-1}$ is the gate time. Indeed, the swap $|\alpha\rangle \leftrightarrow |-\alpha\rangle$ corresponds to the map $|\mathcal{C}_\alpha^+\rangle \rightarrow |\mathcal{C}_\alpha^+\rangle$ and $|\mathcal{C}_\alpha^-\rangle \rightarrow -|\mathcal{C}_\alpha^-\rangle$ which is an $X$ operation for the cat qubit.

The process implementing the $X$ gate thus swaps the two states $|\pm\alpha\rangle$ while keeping the quantum information encoded in the superposition of these states intact. With this regard, the gate consists in imparting a topological $\pi$-phase to the coherent states. We call this phase "topological" because the state of the system during the execution of the gate does no longer belong to the cat qubit subspace. It is only at the end of the gate that the state is brought

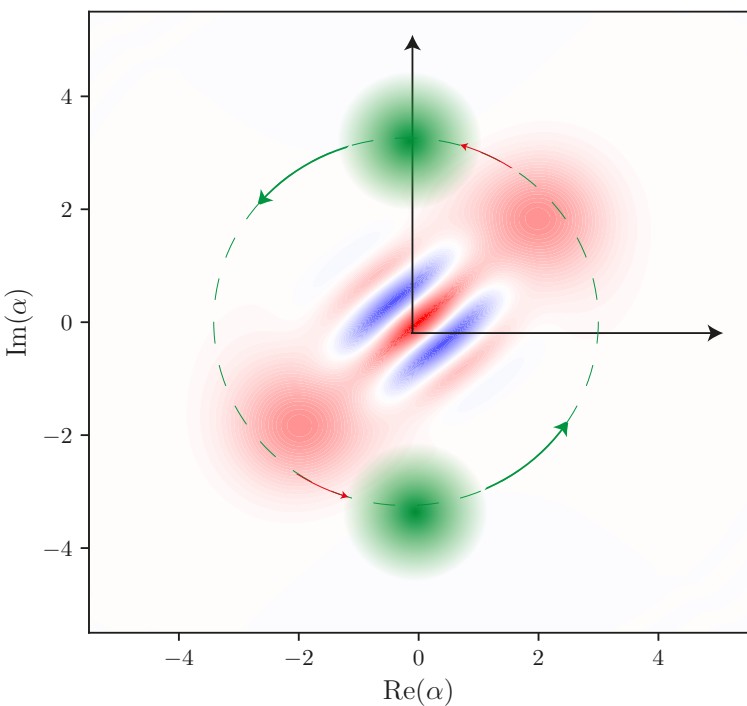

Figure 13: Wigner function of the state of a cat qubit during the execution of an *X* operation. The green dots are the Wigner functions of the instantaneous steady states of the dynamics $\dot{\rho} = \kappa_2 \mathcal{D}[a^2 - \alpha(t)^2]$. These attracting points are slowly rotated from $\pm\alpha$ to $\mp\alpha$ on the dashed circle, as shown by the green arrows. When this rotation is performed slowly, the cat follows the attractive points (red arrows). Reproduced with permission from Ref. [9].

back into this manifold together with an exact $\pi$-phase. This phase is not affected by the imprecision in the rotation angle. Indeed, the phase of the coherent states $|\pm\alpha\rangle$ are locked to the phase of the pump drives. In this sense, each cat qubit is defined with respect to its own pumps. Therefore, even if the rotation angle is not precisely $\pi$, which could happen *e.g.* because of the amplitude and phase fluctuations of the pumping drive, the state has still accumulated a topological $\pi$-phase with respect to its local oscillator. This is to be contrasted with the accumulation of the dynamical phase realized *inside* the code manifold for the Zeno type gates of the previous section.

Note that in addition to this topological $\pi$ phase, there is a geometric phase accumulated due to the particular path taken by $\alpha(t)$. However, this phase is the same for the two states $|\pm\alpha\rangle$ and correspond to a physically meaningless global phase.

In the ideal case of a loss-less harmonic oscillator and in the limit where the gate time $T = +\infty$, the fidelity of this operation with respect to the X operator is 1. This operation is bias-preserving as the errors caused by the finite gate time are only of the phase-flip type, but the bit-flips remain exponentially suppressed in the size of the cat $\bar{n}$. Intuitively, this is possible because the two-photon pumping is never turned off during the gate execution. A schematic representation of this evolution in the phase-space is depicted in Figure 13.

To reduce the phase-flip error rate due to the finite gate time, called the non-adiabatic

errors, the feed-forward Hamiltonian

$$H = -\frac{\pi}{T}a^\dagger a,$$

is turned on while the pumping is being rotated. This Hamiltonian generates the unitary $R(t) = e^{i\frac{\pi}{T}a^\dagger a t}$ which rotates deterministically the qubit state

$$R(t)|\psi_0\rangle = c_0|\alpha(t)\rangle + c_1|-\alpha(t)\rangle,$$

so that it remains at all times in the kernel of the time dependent dissipative channel:

$$\left[a^2 - \alpha(t)^2\right]R(t)|\psi_0\rangle = 0.$$

In presence of this Hamiltonian, there is no need to proceed adiabatically, that is the gate time $T$ can be arbitrarily short.

*CNOT gate.* The idea behind the $X$ gate can be adapted to realize a controlled-$X$ (CNOT) gate between two cat qubits, which consists in applying an $X$ gate to the "target" cat qubit when the "control" qubit is in the computational $|1\rangle \approx |-\alpha\rangle$ state and applying the identity otherwise

$$\text{CNOT} = \tfrac{1}{2}(I_1 + Z_1) \otimes I_2 + \tfrac{1}{2}(I_1 - Z_1) \otimes X_2.$$

In terms of operators acting on the cat qubit, the CNOT gate can be written

$$\text{CNOT} \approx |\alpha\rangle\langle\alpha| \otimes (|\alpha\rangle\langle\alpha| + |-\alpha\rangle\langle-\alpha|) + |-\alpha\rangle\langle-\alpha| \otimes (|\alpha\rangle\langle-\alpha| + |-\alpha\rangle\langle\alpha|).$$

The approximation is exponentially precise in $|\alpha|^2$. This operation is realized by making a rotation of the pumping of the target qubit implementing the $X$ gate that depends on the state of the control qubit, by modifying the dissipation channels of the cat qubits $\mathcal{L}_a = \mathcal{D}[L_a]$ and $\mathcal{L}_b = \mathcal{D}[L_b(t)]$, with:

$$L_a = a^2 - \alpha^2,$$
$$L_b(t) = b^2 - \tfrac{1}{2}\alpha(a + \alpha) + \tfrac{1}{2}\alpha e^{2i\frac{\pi}{T}t}(a - \alpha),$$

where we denote by $a$ (resp. $b$) the mode of the control cat qubit (resp. target cat qubit). The dissipation channel on the control qubit $L_a$ is the two-photon pumping scheme stabilizing the control cat qubit. The second dissipation channel, however, acts on the target cat qubit but also depends on the first mode $a$. It should be understood as follows: when the control qubit $a$ is in the state $|\alpha\rangle$, the operator $L_b(t)$ acts on the target mode as $b^2 - \alpha^2$, stabilizing the idle code space, but when the control qubit is in the state $|-\alpha\rangle$, the pumping becomes $b^2 - (\alpha e^{i\frac{\pi}{T}t})^2$, thus implementing the time-dependent two-photon pumping dissipation used for the $X$ gate. Just like for the $X$ gate, the pumping is always turned on during the gate and the bit-flip errors remain exponentially suppressed throughout the gate, ensuring that the CNOT gate preserves the biased structure of the noise.

In the case of the $X$ gate, the geometric phase corresponded to a physically meaningless global phase, but here this phase is conditioned on the state of the control qubit. As a consequence, the geometric phase induces a deterministic rotation around the $Z$-axis of the control qubit. The rotation angle is given by

$$\vartheta = -i\int_0^T \langle\pm\alpha(t)|\frac{d}{dt}|\pm\alpha(t)\rangle dt = \pi|\alpha|^2.$$

This deterministic geometric phase can be removed by applying the appropriate $Z(\theta)$ operation discussed above. Another option is to ensure the rotation angle $\vartheta$ is a multiple of $2\pi$, either by

setting the number of photons to be an even integer or by choosing a path $\alpha(t)$ such that the result of the integral is a multiple of $2\pi$. Even in this case, the fluctuations along the chosen path will inevitably lead to a certain imprecision in the final value of the geometric phase, leading to additional phase-flip errors.

A major part of the phase-flip errors induced by non-adiabatic effects can be compensated in the same way as for the X gate, by adding a Hamiltonian evolution of the form

$$H = \frac{1}{2}\frac{\pi}{T}\frac{\boldsymbol{a}-\alpha}{2\alpha}\otimes(\boldsymbol{b}^\dagger\boldsymbol{b}-\bar{n})+\text{h.c.},$$

while rotating the pumping. In presence of two-photon pumping, this Hamiltonian is an approximation of the "ideal" Hamiltonian that would perfectly cancel all of the non-adiabatic errors

$$H^* = -\frac{\pi}{T}|-\alpha\rangle\langle-\alpha|\otimes(\boldsymbol{b}^\dagger\boldsymbol{b}-\bar{n}),$$

which triggers a rotation of the target cat qubit in the phase-space conditional to the control cat qubit being in the state $|-\alpha\rangle$. Similar Hamiltonians have been already realized using parametric methods [72], (see subsection 4.4).

*Toffoli gate.* The Toffoli gate is the three-qubit controlled-controlled-X gate (CCX)

$$\begin{aligned}\text{Toffoli} = {}&\tfrac{1}{4}(I_1+Z_1)(I_2+Z_2)I_3 + \tfrac{1}{4}(I_1+Z_1)(I_2-Z_2)I_3\\&+\tfrac{1}{4}(I_1-Z_1)(I_2+Z_2)I_3 + \tfrac{1}{4}(I_1-Z_1)(I_2-Z_2)X_3\,.\end{aligned}$$

This unitary is in the third level of the Clifford hierarchy, thus it does not belong to the Clifford group. In many of the schemes achieving universality, the non-Clifford operations are the most difficult operations to implement. While the Toffoli gate is undeniably the most complicated gate of the physical gate set, its implementation is similar to the two-qubit CNOT gate.

Similarly to the CNOT gate, only the dissipation channel of the target cat qubit needs to be modified, $\mathcal{L}_a = \mathcal{D}[\boldsymbol{L}_a]$, $\mathcal{L}_b = \mathcal{D}[\boldsymbol{L}_b]$ and $\mathcal{L}_c = \mathcal{D}[\boldsymbol{L}_c(t)]$,

$$\boldsymbol{L}_a = \boldsymbol{a}^2-\alpha^2\,,$$
$$\boldsymbol{L}_b = \boldsymbol{b}^2-\alpha^2\,,$$
$$\boldsymbol{L}_c(t) = \boldsymbol{c}^2-\tfrac{1}{4}(\boldsymbol{a}+\alpha)(\boldsymbol{b}+\alpha)+\tfrac{1}{4}(\boldsymbol{a}+\alpha)(\boldsymbol{b}-\alpha)+\tfrac{1}{4}(\boldsymbol{a}-\alpha)(\boldsymbol{b}+\alpha)-\tfrac{1}{4}e^{2i\frac{\pi}{T}t}(\boldsymbol{a}-\alpha)(\boldsymbol{b}-\alpha)]\,.$$

Here, $\mathcal{L}_a$ and $\mathcal{L}_b$ keep stabilizing the two control modes $\boldsymbol{a}$ and $\boldsymbol{b}$ in manifolds spanned by $|\pm\alpha\rangle$, and $\mathcal{L}_c$ rotates the two-photon pumping on the target mode $\boldsymbol{c}$ only when the control cat qubits are in the state $|-\alpha,-\alpha\rangle$.

In theory, assuming the required couplings between any number of modes are available, the mechanism behind the topological X, CNOT and Toffoli gates can be straightforwardly adapted to implement the $n$-qubit entangling gate $C^{n-1}X$ belonging to the $n$-th level of the Clifford hierarchy, where $C^{n-1}$ denotes the controls on the first $n-1$ qubits. Note that in practice, the implementation of the required dissipative channels would involve non-linear processes of higher order which are much more complex to realize and that would typically be weak.

As for the CNOT gate, the deterministic geometric phase associated to the path taken by the target cat qubit can also be eliminated by tailoring the path followed in the phase-space by the cat states during the execution of the gate, or by physically applying $Z(\theta)$ and $ZZ(\theta)$ gates.

Similarly to all the topological gates implemented on the cat qubits involving a continuous evolution of the code space, the fidelity of the dissipative implementation can be improved

by adding a feed-forward Hamiltonian whose role is merely to reduce the phase-flip errors induced by non adiabaticity. Again, the systematic construction of such a Hamiltonian is based on the adaptation of the ideal feed-forward Hamiltonian for the $X$ gate to the particular case where this rotation is realized conditionally on some control cat qubits state. In the specific case of the Toffoli gate, the target cat qubit (mode $c$) undergoes a rotation in the phase-space implementing the $X$ gate only when the joint state of the two control cat qubits is $|-\alpha,-\alpha\rangle$. Thus, in analogy with the $X$ gate, a "perfect" feed-forward Hamiltonian that would exactly cancel the non-adiabatic phase-flip errors is

$$H = -\frac{\pi}{T}|-\alpha\rangle\langle-\alpha| \otimes |-\alpha\rangle\langle-\alpha| \otimes c^\dagger c.$$

Just like for the CNOT gate, the projectors on coherent states are not Hamiltonians that can be implemented; but they can be well approximated by (where the approximation on the cat manifold is exponentially good in $\alpha$)

$$|-\alpha\rangle\langle-\alpha| \approx \frac{\alpha - a}{2\alpha},$$

and the resulting approximate Hamiltonian that we propose to apply while a Toffoli gate is performed to remove most of the non-adiabatic phase-flip errors is thus given by

$$H = -\frac{1}{2}\frac{\pi}{T}\frac{a-\alpha}{2\alpha} \otimes \frac{b-\alpha}{2\alpha} \otimes (c^\dagger c - \bar{n}) + \text{h.c.}$$

*SWAP gate.* The two-qubit SWAP gate, which acts like its name suggests, is trivially compatible with a bias-preserving implementation as it does not convert $Z$-type errors into $X$- or $Y$-type errors. The SWAP gate is often useful to adapt logical circuits to actual constraints on the connectivity graph of the physical qubits. Noting that a SWAP gate can be implemented using three CNOT gates establishes that the SWAP gate can be implemented in a bias-preserving manner, but there is a more direct way to do this. Following the guiding principle of the CNOT gate, the SWAP gate is realized by replacing the regular two-photon dissipation operators $L_a = a^2 - \alpha^2$ and $L_b = b^2 - \alpha^2$ by the following time-dependent operators that combine both modes

$$L_a(t) = a^2 - \frac{1}{2}ab\left(1 - e^{2i\frac{\pi}{T}t}\right) - \frac{1}{2}\alpha^2\left(1 + e^{2i\frac{\pi}{T}t}\right),$$
$$L_b(t) = b^2 - \frac{1}{2}ab\left(1 - e^{-2i\frac{\pi}{T}t}\right) - \frac{1}{2}\alpha^2\left(1 + e^{-2i\frac{\pi}{T}t}\right),$$

for $t \in [0, T]$ where $T$ is the SWAP gate time. The instantaneous joint kernel of these operators is the four dimensional Hilbert space spanned by the set of coherent states

$$\left\{|\alpha,\alpha\rangle, |-\alpha,-\alpha\rangle, |\alpha e^{i\frac{\pi}{T}t}, -\alpha e^{-i\frac{\pi}{T}t}\rangle, |-\alpha e^{i\frac{\pi}{T}t}, \alpha e^{-i\frac{\pi}{T}t}\rangle\right\}.$$

Recalling that $|0\rangle \approx |\alpha\rangle$ and $|1\rangle \approx |-\alpha\rangle$, these two dissipation channels implement the correct mapping corresponding to a SWAP gate:

$$|\alpha,\alpha\rangle \rightarrow |\alpha,\alpha\rangle,$$
$$|-\alpha,-\alpha\rangle \rightarrow |-\alpha,-\alpha\rangle,$$
$$|\alpha,-\alpha\rangle \rightarrow |-\alpha,\alpha\rangle,$$
$$|-\alpha,\alpha\rangle \rightarrow |\alpha,-\alpha\rangle.$$

Similarly to the others gates that are implemented using a rotation of the steady states of the driven-dissipative super-operators in the phase space, the phase-flip errors of a SWAP gate caused by non-adiabaticity are reduced when the Hamiltonian

$$H = -\frac{\pi}{4\alpha^2 T}(a^\dagger a - b^\dagger b)(\alpha^2 - ab) + \text{h.c.}$$

is added during the gate. The operator $(\alpha^2 - \boldsymbol{ab})/2\alpha^2$ acts as identity on the states $|\alpha e^{i\frac{\pi}{T}t}, -\alpha e^{-i\frac{\pi}{T}t}\rangle$ and $|-\alpha e^{i\frac{\pi}{T}t}, \alpha e^{-i\frac{\pi}{T}t}\rangle$ while it vanishes on the states $|\alpha, \alpha\rangle$ and $|-\alpha, -\alpha\rangle$. The above Hamiltonian thus reduces to the required rotating term $\pi(\boldsymbol{a}^\dagger \boldsymbol{a} - \boldsymbol{b}^\dagger \boldsymbol{b})/T$ only when the cat qubits are in a state that is moved around in the phase space, and vanishes otherwise.

## 4.3 Error models

In this subsection, we detail the error models of the various gates introduced above. We give a particular attention to the CNOT gate, as this gate is crucial for the stabilizer measurements. We analyze the error models resulting from two different sources of errors: the single-photon loss of the quantum harmonic oscillator at a rate $\kappa_1$, and the non-adiabaticity of the gates. Actually, apart from state preparation and measurement, and in the absence of any source of decoherence, all the gates have unit fidelity in the limit where the gate time is infinite. We discuss phase-flip and bit-flip errors in two different ways.

The phase-flip errors are exponentially dominant. For these errors, we give explicit analytical formulas. More precisely, the analytical formula for the phase-flip errors induced by photon loss are explicitly calculated. The analytical formula for the phase-flip errors induced by non-adiabaticity are derived using a combination of a systematic adiabatic elimination theory to guess the scaling of the formula together with a numerical fit to determine the constant prefactor. Using this method, we were able to derive the non-adiabatic phase-flip errors for the topological gates (actually, the X gate has no non-adiabatic phase-flip error when the feedforward Hamiltonian is added). A thorough study of all the gates on dissipative cat qubits was recently published in [73]. This paper applies a new method, based on the "shifted Fock basis" adapted to the cat states, and derives analytically the non-adiabatic phase-flip errors for the Zeno gates, thus completing the analysis of phase-flip errors. For the sake of completeness, we give these formulas without including the derivation, which is thoroughly exposed in [73]. Because the phase-flip errors induced by the natural losses of the quantum harmonic oscillator increase with the gate time, while the phase-flip errors induced by non-adiabaticity decrease with the gate time, the combination of these two sources of errors gives rise to an optimal finite gate time that minimizes the phase-flip errors.

We claim that the cat qubit encoding, the two photon stabilization, and the careful bias-preserving implementations of the gates, result in gates for which the bit-flip errors are exponentially suppressed even during the execution of the gates, this point being crucial for a hardware-efficient scaling towards fault-tolerance [9]. Here, we give numerical evidence for this claim by performing numerical process tomography of two gates, the $Z(\theta)$ and the CNOT gate, for increasing cat sizes, for which the exponential suppression of bit-flips is indeed observed.

*Identity and SPAM errors.* The dynamics of an idling cat qubit subject to photon loss is modelled by the master equation

$$\frac{d\rho}{dt} = \kappa_2 \mathcal{D}[\boldsymbol{a}^2 - \alpha^2]\rho + \kappa_1 \mathcal{D}[\boldsymbol{a}]\rho \,, \tag{34}$$

where $\kappa_2$ is the rate of the engineered two-photon dissipation and $\kappa_1$ the rate of single photon loss. The exponential suppression of bit-flips for the idling cat qubits was discussed in subsection 2.3 and Figure 7. Here, we discuss rapidly the model for phase-flip errors. The photon loss operator $\boldsymbol{a}$ induces a phase-flip error on the cat qubit, $\boldsymbol{a}|\mathcal{C}_\alpha^\pm\rangle = \alpha \tanh(|\alpha|^2)^{\pm 1/2}|\mathcal{C}_\alpha^\mp\rangle$. Thus, the phase-flip error probability induced by single photon loss at rate $\kappa_1$ during a time $T$ is given by $p_Z = \bar{n}\kappa_1 T$. This leading order contribution can be calculated explicitly by considering the evolution of the cat state $|\mathcal{C}_\alpha^\pm\rangle$ under the evolution (34). Assuming that $\kappa_1 \ll \kappa_2$ such that the dynamics remains in the cat qubit manifold and looking for a solution of the form

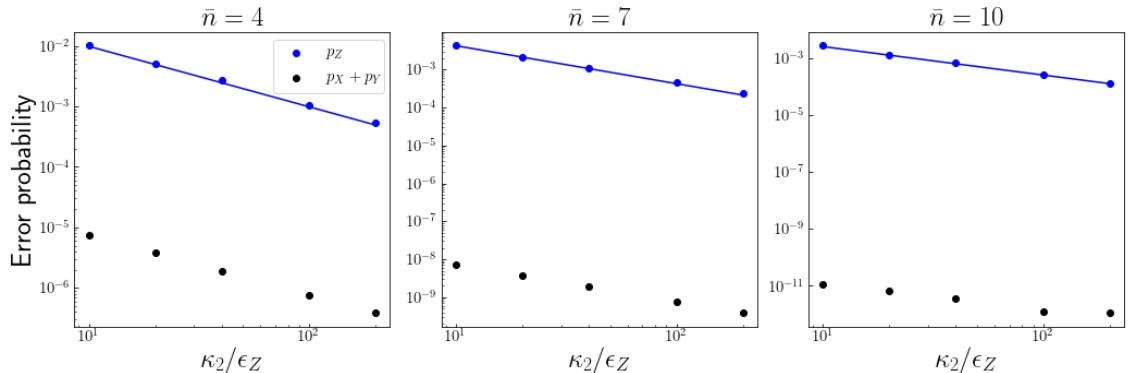

Figure 14: Numerical simulation of the non-adiabatic errors of the $Z = Z(\pi)$ gate implemented by the master equation (35). The non-adiabatic phase-flip error $Z$ is linearly suppressed with the gate time $T$, while the bit-flip type errors $X$ and $Y$ are exponentially suppressed with the mean number of photons $\bar{n}$.

$\rho(t) = (1 - p(t))|\mathcal{C}_\alpha^+\rangle\langle\mathcal{C}_\alpha^+| + p(t)|\mathcal{C}_\alpha^-\rangle\langle\mathcal{C}_\alpha^-|$, the evolution of the population of the cat states is given by (dropping the terms exponentially small in $\alpha$)

$$\dot{p}(t) = \kappa_1\bar{n}(1 - 2p(t)).$$

Thus, starting from the initial cat state $|\mathcal{C}_\alpha^\pm\rangle$ ($p(0) = 0$), the phase-flip error probability is given by $p(t) = \frac{1}{2}(1 - e^{-2\bar{n}\kappa_1 t}) \approx \bar{n}\kappa_1 t$ when $\bar{n}\kappa_1 t \gg 1$. Finally, we expect that the preparation of the cat states $|\mathcal{C}_\alpha^\pm\rangle$ and the measurement of these states can be performed with a similar phase-flip error probability $p_Z = \bar{n}\kappa_1 T$, where $T$ is the typical preparation and measurement time.

*$Z(\theta)$ gate.* The master equation describing the rotation of an angle $\theta$ around the $Z$ axis in time $T = \dfrac{\theta}{4\sqrt{\bar{n}}|\epsilon_Z|}$ is

$$\dot{\rho} = -i[\epsilon_Z \boldsymbol{a} + \epsilon_Z^* \boldsymbol{a}^\dagger, \rho] + \kappa_2\mathcal{D}[\boldsymbol{a}^2 - \alpha^2]\rho. \tag{35}$$

The phase-flip errors induced by photon loss at rate $\kappa_1$, described by adding the term $\kappa_1\mathcal{D}[\boldsymbol{a}]\rho$ to the above master equation, commute with the gate at all times. For this reason, the effect of photon loss can be accounted for separately, and the phase-flip errors induced by photon loss are the same as in the memory case

$$p_Z[\text{photon loss}] = \bar{n}\kappa_1 T = \frac{\kappa_1\sqrt{\bar{n}}\theta}{4|\epsilon_Z|}.$$

Furthermore, in [73], the analytical formula proposed for the non-adiabatic phase-flip errors is

$$p_Z[\text{non-adiabaticity}] = \frac{\theta^2}{16\kappa_2\bar{n}^2 T} = \frac{\theta|\epsilon_Z|}{4\kappa_2\bar{n}^{3/2}}.$$

We perform a numerical simulation of master equation (35) in Figure 14, for a rotation of angle $\theta = \pi$ and in the absence of photon loss (that is, to check the non-adiabatic error model). The dotted points (numerical results) are in good agreement with the analytical formula (blue curve).

Taking into account both the phase-flip errors induced by photon loss and by non-adiabaticity, the total phase-flip error probability for a $Z(\theta)$ gate implemented in time T is given by

$$p_Z = \bar{n}\kappa_1 T + \frac{\theta}{16\kappa_2\bar{n}^2 T},$$

which is minimal for $T^* = \dfrac{\sqrt{\theta}}{4\bar{n}^{3/2}\sqrt{\kappa_1\kappa_2}}$, for which the phase-flip rate is

$$p_Z = \frac{\sqrt{\theta}}{2\sqrt{\bar{n}}}\sqrt{\frac{\kappa_1}{\kappa_2}}.$$

*CZ gate.* The same analysis has been carried through in [73] for the $ZZ(\theta)$ gate, from which the CZ gate can be implemented by combining the single qubit $Z$ rotations. While the errors induced by photon loss result in independent $Z$ errors on the two cat qubits with same probability $\bar{n}\kappa_1 T$ as before, it is shown that the non-adiabatic phase-flip errors result in both independent $Z_1$ and $Z_2$ errors as well as correlated $Z_1 Z_2$ errors. The analytical formula for the overall phase-flip errors are given by

$$p_{Z_1} = p_{Z_2} = \bar{n}\kappa_1 T + \frac{\theta^2}{64\kappa_2\bar{n}^2 T},$$

$$p_{Z_1 Z_2} = \frac{\theta^2}{32\kappa_2\bar{n}^2 T}.$$

Note that the photon loss induced errors increase linearly with $T$ while the non-adiabatic errors decrease linearly with $T$. The optimal gate time minimizing these errors is given by $T^* = \dfrac{\sqrt{\theta}}{4\sqrt{2}\bar{n}^{3/2}\sqrt{\kappa_1\kappa_2}}$.

*X gate.* The master equation implementing the topological $X$ gate in time $T$ is given by

$$\dot{\rho} = i\left[\frac{\pi}{T}\mathbf{a}^\dagger\mathbf{a}, \rho\right] + \kappa_2\mathcal{D}\left[\mathbf{a}^2 - (\alpha e^{i\frac{\pi}{T}t})^2\right],$$

where the (optional) Hamiltonian term is added to compensate the non-adiabatic phase-flip errors while the dissipative term implements the continuous deformation of the cat qubit subspace. Actually, for this gate, the Hamiltonian removes *all* of the non-adiabatic phase-flip errors. Indeed, in the rotating frame of this Hamiltonian, the dynamics reads

$$\dot{\rho} = \kappa_2\mathcal{D}[\mathbf{a}^2 - \alpha^2],$$

which is simply the two-photon stabilization. Thus, for this gate, there are only the phase-flips errors induced by photon loss, which are given by

$$p_Z = \bar{n}\kappa_1 T.$$

*CNOT gate.* We now investigate the error model of the CNOT gate. As we have argued before, this gate is particularly important for error correction, such that a detailed analysis is provided. In particular, we numerically check that the analysis is robust when adding additional sources of errors on the quantum harmonic oscillator, including thermal excitation and dephasing.

In order to understand the effect of the loss of a photon during the execution of the CNOT, let us consider the operation approximately generated by the two dissipation channels $\mathcal{L}_a$ and $\mathcal{L}_b$. In the cat qubits subspaces where the dynamics is confined, these channels implement a unitary operation of the form:

$$\mathbf{U}(t) = |\alpha\rangle\langle\alpha| \otimes \mathbf{I} + |-\alpha\rangle\langle-\alpha| \otimes e^{i\frac{\pi}{T}t\mathbf{b}^\dagger\mathbf{b}},$$

with $\mathbf{U}(0) = \mathbf{I} \otimes \mathbf{I}$ and $\mathbf{U}(T) = \text{CNOT}$.

Consider the effect of a loss of a single photon of the control mode $\boldsymbol{a}$ at an arbitrary time $t \in [0, T]$. The noisy quantum operation $\mathcal{E}_{\boldsymbol{a}}$ performed instead of the CNOT is given by

$$
\begin{aligned}
\mathcal{E}_{\boldsymbol{a}} &= U(T-t)[\boldsymbol{a} \otimes I]U(t) \\
&= \alpha|\alpha\rangle\langle\alpha| \otimes I - \alpha|-\alpha\rangle\langle-\alpha| \otimes e^{i\pi \boldsymbol{b}^\dagger \boldsymbol{b}} \\
&= [\boldsymbol{a} \otimes I]\mathrm{CNOT},
\end{aligned}
$$

which can be written in terms of Pauli operators for the cat qubits as

$$
\mathcal{E}_{\boldsymbol{a}} = Z_1 \mathrm{CNOT}.
$$

In other words, the loss of a photon on the control cat qubit causes a phase-flip on that qubit but does not affect the target cat qubit.

On the other hand, a photon loss occurring on the target cat qubit $\boldsymbol{b}$ at time t propagates as

$$
\begin{aligned}
U(T-t)[I \otimes \boldsymbol{b}]U(t) &= (I \otimes \boldsymbol{b})(|\alpha\rangle\langle\alpha| \otimes I + e^{-i\pi\frac{T-t}{T}}|-\alpha\rangle\langle-\alpha| \otimes e^{i\pi \boldsymbol{b}^\dagger \boldsymbol{b}}) \\
&= (I \otimes \boldsymbol{b})(|\alpha\rangle\langle\alpha| \otimes I + e^{-i\pi\frac{T-t}{T}}|-\alpha\rangle\langle-\alpha| \otimes I)\mathrm{CNOT}.
\end{aligned}
$$

The resulting error

$$
I \otimes \boldsymbol{b}(|\alpha\rangle\langle\alpha| \otimes I + e^{-i\pi\frac{T-t}{T}}|-\alpha\rangle\langle-\alpha| \otimes I),
$$

induced by the propagation of the photon loss can be expressed in terms of the Pauli operators of cat qubits as

$$
\boldsymbol{U}_{\mathrm{err}}(\theta) = \frac{1}{2}(1 + Z_1)Z_2 + \frac{1}{2}e^{i\theta}(1 - Z_1)Z_2,
$$

where $\theta = -i\pi(1 - t/T)$ is a random phase. The time of the jump being uniformly distributed over the interval [0,T], the noisy operation $\mathcal{E}_{\hat{b}}$ can be written

$$
\begin{aligned}
\mathcal{E}_{\boldsymbol{b}}(\rho) &= \bar{n}\kappa_1 T \int_{-\pi}^{0} \frac{d\theta}{\pi} \boldsymbol{U}_{\mathrm{err}}(\theta)\tilde{\rho}\boldsymbol{U}_{\mathrm{err}}(\theta)^\dagger \\
&= \bar{n}\kappa_1 T \left[\frac{1}{2}Z_2\tilde{\rho}Z_2 + \frac{1}{2}Z_1 Z_2\tilde{\rho}Z_1 Z_2 + \frac{i}{\pi}Z_1 Z_2\tilde{\rho}Z_2 - \frac{i}{\pi}Z_2\tilde{\rho}Z_1 Z_2\right],
\end{aligned}
$$

where $\tilde{\rho} = \mathrm{CNOT}\rho\mathrm{CNOT}$ is the image of $\rho$ by a perfect CNOT operation and $\bar{n}\kappa_1 T$ is the average number of photons lost in each mode during the execution of the gate. Note that this analytical formula is an approximation that only accounts for the effect of the loss of a single photon loss. In addition to this dominant phase-flip error corresponding to the loss of a single photon, the cat states are also slightly deformed towards the center of the phase space, causing (exponentially small) bit-flip errors. Importantly, while the bit-flip are still exponentially suppressed with the cat size as $e^{-2\bar{n}}$, the constant prefactor in front of this exponential suppression is significantly larger than for the case where the two-photon dissipation is time independent. Also, the loss of more than a single photon result in a phase-flip error rate slightly different from this one. However, the numerical simulation of the process confirms that this approximation captures well most of the errors that are caused by photon loss.

The factorization of the operation $\mathcal{E}_{\boldsymbol{b}}$ as a perfect CNOT gate followed by some noise operators makes it easier to analyze the effect of the errors. The first two terms indicate that the effect of photon loss on the target cat qubit produces two types of errors of the same strength: phase-flips on the target cat qubit $\frac{1}{2}Z_2\tilde{\rho}Z_2$ as well as a correlated phase-flips on both qubits $\frac{1}{2}Z_1 Z_2\tilde{\rho}Z_1 Z_2$, with some degree of coherence between these two errors.

When losses on both modes are taken into account, the noisy CNOT gate $\mathcal{E}_{a,b}$ is described by the Kraus map:

$$\mathcal{E}_{a,b}(\rho) = \sum_{k=1,2,3} M_k \tilde{\rho} M_k^{\dagger},$$

where, noting $r = \frac{1}{2}\arcsin(2/\pi)$, the Kraus operators are given by

$$M_1 = \sqrt{\bar{n}\kappa_1 T}\, Z_1,$$
$$M_2 = \sqrt{\tfrac{\bar{n}\kappa_1 T}{2}}(\cos r\, I_1 + i\sin r\, Z_1)Z_2,$$
$$M_3 = \sqrt{\tfrac{\bar{n}\kappa_1 T}{2}}(\sin r\, I_1 + i\cos r\, Z_1)Z_2.$$

Let us now consider the phase-flip errors induced by non-adiabaticity. The addition of the feed-forward Hamiltonian

$$H = \frac{1}{2}\frac{\pi}{T}\frac{a-\alpha}{2\alpha}\otimes(b^{\dagger}b-\bar{n}) + \text{h.c.},$$

compensates most of the errors induced by the finite gate time $T$, and it is possible to characterize the remaining errors. Using the systematic adiabatic elimination techniques of [71], one can check that it is only composed of phase-flips on the control cat qubit $Z_1$, with a rate proportional to $(\bar{n}\kappa_2 T)^{-1}$. The exact coefficient of proportionality is estimated by a numerical fit and is found to be around 0.159:

$$p_{Z_1}[\text{non-adiabaticity}] = 0.159(\bar{n}\kappa_2 T)^{-1}.$$

We note that using the shifted Fock basis, the authors of [73] derived an approximate error model and found that

$$p_{Z_1}[\text{non-adiabaticity}] = \frac{\pi^2}{64}(\bar{n}\kappa_2 T)^{-1},$$

in close agreement with our numerical estimation $\dfrac{\pi^2}{64}\approx 0.154$.

The probability of the "environment" induced phase-flip errors, *e.g* by photon loss, increase linearly with the gate time $T$, whereas phase-flip errors caused by non-adiabaticity are reduced when the gate time is increased. This opposite behavior gives rise to a finite optimal gate time $T^*$ for which the gate fidelity is maximal.

More precisely, taking into account phase-flip errors caused by both photon loss and non-adiabaticity, the total phase-flip error probability on the control cat qubit is given by

$$p_{Z_1} = p_{Z_1}[\text{photon loss}] + p_{Z_1}[\text{non-adiabaticity}] = \bar{n}\kappa_1 T + 0.159(\bar{n}\kappa_2 T)^{-1}.$$

The gate fidelity $\mathcal{F}$ of the implemented CNOT operation, defined in equation (36), is given by

$$\mathcal{F} = \sqrt{1-(p_{Z_1}+p_{Z_2}+p_{Z_1 Z_2})} = \sqrt{1-2\bar{n}\kappa_1 T - 0.159(\bar{n}\kappa_2 T)^{-1}}.$$

The highest value of the fidelity that can be achieved is set by the ratio $\kappa_1/\kappa_2$

$$\mathcal{F} = \sqrt{1-1.13\sqrt{\frac{\kappa_1}{\kappa_2}}},$$

achieved for the optimal gate time

$$T^* = 0.282\left[\bar{n}\sqrt{\frac{\kappa_1}{\kappa_2}}\right]^{-1}\kappa_2^{-1}.$$

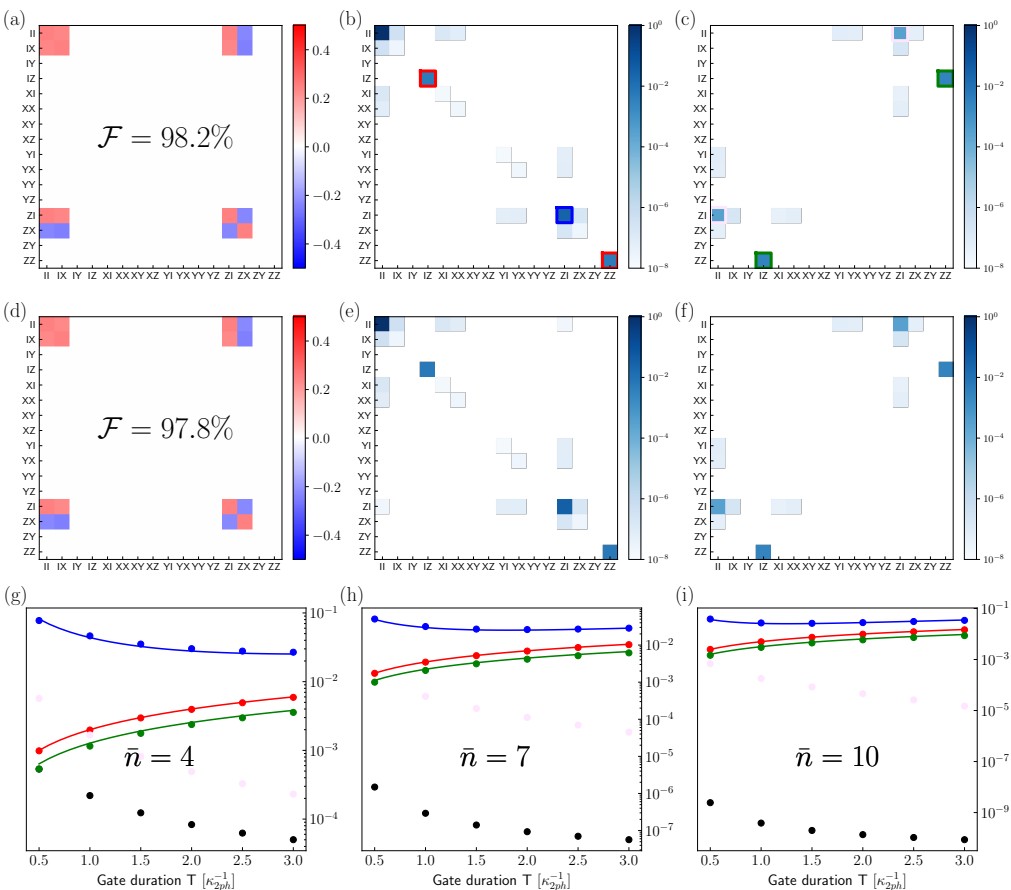

Figure 15: Process tomography of the CNOT gate in presence of noise. The CNOT process is numerically simulated for $\bar{n} = 7$ photons cat qubits using two different error models. First, we consider photon loss on both modes $\kappa_1 \mathcal{D}[\boldsymbol{a}] + \kappa_1 \mathcal{D}[\boldsymbol{b}]$ (a-c). Then, we consider a more elaborate error model including photon loss $\kappa_1(1 + n_{th})\mathcal{D}[\boldsymbol{a}] + \kappa_1(1 + n_{th})\mathcal{D}[\boldsymbol{b}]$, thermal excitations $\kappa_1 n_{th} \mathcal{D}[\boldsymbol{a}^\dagger] + \kappa_1 n_{th} \mathcal{D}[\boldsymbol{b}^\dagger]$ ($n_{th} = 10\%$) and dephasing on both modes $\kappa_\phi \mathcal{D}[\boldsymbol{a}^\dagger \boldsymbol{a}] + \kappa_\phi \mathcal{D}[\boldsymbol{b}^\dagger \boldsymbol{b}]$ (d-f). In both cases, we set $\kappa_1/\kappa_2 = 10^{-3}$ and the gate time is chosen optimal $T^* = 0.282[\bar{n}\sqrt{\kappa_1 \kappa_2}]^{-1} \approx 1.27\kappa_2^{-1}$ (see main text). We plot the real part of the process matrix $\chi$ (a,d), and the real (b,e) and imaginary (c,f) part of the error matrix $\chi^{\mathrm{err}}$. In the lower row (g,h,i), we check the validity of the analytical error model for photon loss for various gate times and cat sizes. The dots illustrate the simulation results where the full master equation in presence of loss is considered, the plain lines correspond to the analytical formula provided in the main text. The blue dots correspond to the diagonal process matrix element corresponding to $Z_1$ errors, the red dots correspond to the coinciding diagonal matrix elements corresponding to $Z_2$ and $Z_1 Z_2$ errors. The green dots correspond to the off-diagonal elements corresponding to the coherence between $Z_2$ and $Z_1 Z_2$ errors. The pale magenta dots correspond to the off-diagonal elements corresponding to coherence between $Z_1$ and $I$, this coherence is due to high-order non-adiabatic effects (not included in our model). The black dots correspond to all of the remaining errors, including bit-flip type ones. It is clear that these errors are exponentially suppressed with the mean number of photons $\bar{n}$. Reproduced with permission from Ref. [9].

For the ratio $\frac{\kappa_1}{\kappa_2} = 10^{-3}$ considered in Figure 15, this theoretical formula predicts a gate fidelity of $\mathcal{F} = 98.2\%$, in agreement with the numerical simulation.

The validity of this error model is checked numerically in Figure 15 (a,b,c). The full master equation of the system is simulated in presence of photon loss. The process matrix $\chi$ plotted in (a) completely characterizes the quantum operation $\mathcal{E}$ performed via the relation [5]

$$\mathcal{E}(\rho) = \sum_{mn} \chi_{mn} P_m \rho P_n^\dagger,$$

where $\{P_j\}$ is the set of two-qubit Pauli operators. The gate fidelity $\mathcal{F}$ is defined as [5]

$$\mathcal{F}(U, \mathcal{E}) = \min_{|\psi\rangle} \mathcal{F}(U|\psi\rangle, \mathcal{E}(|\psi\rangle\langle\psi|)), \tag{36}$$

where $U = $ CNOT is the perfect CNOT operation the minimum is taken over the set of all possible two-qubit states $|\psi\rangle$. The unitary of the perfect CNOT is factored out in order to obtain the process error matrix $\chi^{\text{err}}$ (real part in (b), imaginary part in (c)), which characterizes the noise alone:

$$\mathcal{E}(\rho) = \sum_{mn} \chi_{mn}^{\text{err}} P_m \tilde{\rho} P_n^\dagger,$$

with $\tilde{\rho} = \text{CNOT}\rho\text{CNOT}$ the image of $\rho$ by a perfect CNOT. In other words, we decompose the noisy CNOT into a perfect CNOT followed by some noise process, characterized by the process error matrix $\chi^{\text{err}}$. As can be seen in the real part of $\chi^{\text{err}}$ (Figure 15-b), photon loss and non-adiabaticity only cause phase-flip errors $Z_1$, $Z_2$ or $Z_1 Z_2$.

We further investigate our theoretical model for errors caused by photon loss by plotting in Figure 15(g,h,i) the values of the coefficients of the error matrix $\chi^{\text{err}}$ (marked by colored squares) as a function of gate duration. The blue dots correspond to phase-flip errors on the control cat qubit $Z_1$ induced by a combination of non-adiabatic errors and the photon loss. The plain blue line corresponds to the analytical formula

$$p_{Z_1} = \bar{n}\kappa_1 T + 0.159(\bar{n}\kappa_2 T)^{-1}.$$

The red dots represent the phase-flip errors on target qubit $Z_2$ and the correlated phase-flip errors $Z_1 Z_2$. These values coincide and are given by

$$P_{Z_2} = P_{Z_1 Z_2} = \tfrac{1}{2}\bar{n}\kappa_1 T,$$

as is represented by the plain line in red. The off-diagonal term corresponding to the coherence between $Z_2$ and $Z_1 Z_2$ errors (green dots) also fit very well our expectation. The pale purple dots correspond to the off-diagonal term representing the coherence between $I$ and $Z_1$ errors. In order to capture such a coherence, one needs to push the non-adiabatic perturbation techniques [71] up to third order, which we have not done yet. Most importantly, the remaining errors (namely the ones that contain an X or Y Pauli operator) represented by the black lines are exponentially suppressed by the cat size, that confirms the bias-preserving aspect of the gate.

As discussed in subsection 2.3, in presence of the two-photon pumping scheme, any physical noise process with a local effect in the phase space of the harmonic oscillator causes bit-flips that are exponentially suppressed in the size of the cat qubits, thus preserving the biased structure of the noise. We now provide a numerical evidence of this fact for a more elaborate set of physical noise processes for the superconducting cavity: photon loss $a$, thermal excitation $a^\dagger$ with a non-zero temperature, and photon dephasing $a^\dagger a$.

In Figure 15, we characterize the performed operation by plotting the process matrix $\chi$ (d), and the real part (e) and imaginary part (f) of the error matrix. In this simulation, $\kappa_1/\kappa_2 = 10^{-3}$, the photon loss is given by $\kappa_1(1 + n_{\text{th}})\mathcal{D}[a]$ and thermal excitations by

$\kappa_1 n_{\text{th}} \mathcal{D}[a^\dagger]$ with $n_{\text{th}} = 10\%$, and the dephasing on the cavity is given by $\kappa_\phi \mathcal{D}[a^\dagger a]$ with $\kappa_\phi = \kappa_1$.

Note that the resulting error matrix and gate fidelity are barely affected by the added thermal excitations and photon dephasing. The addition of thermal noise and dephasing slightly decrease the fidelity of the operation, from 98.2% to 97.8%, but as expected, this decrease is caused by an increased rate of phase-flip errors, while all bit-flip errors remain exponentially suppressed.

Very interestingly, the phase-flip error probability induced by the cavity dephasing was computed explicitly in the recent work [73]. The fact that cavity dephasing at rate $\kappa_\phi$, described by the dissipation super-operator $\kappa_\phi \mathcal{D}[a^\dagger a]$, can lead to phase-flip errors might be surprising. Indeed, the dephasing operator $a^\dagger a$ commutes with the photon number parity operator $(-1)^{a^\dagger a}$, such that one may naively think that it cannot induce transitions between the two cat states $|\mathcal{C}_\alpha^\pm\rangle$ of well-defined photon number parity. While this is in general true for an idling cat qubit, the photon number parity of the cat states *during* the execution of the CNOT gate does not remain well-defined, thus exposing the cat state to some dephasing-induced phase-flips. In [73], it was shown that cavity dephasing $\kappa_\phi \mathcal{D}[a^\dagger a]$ results in a phase-flip error only on the control cat qubit, with probability

$$p_{Z_1} = \frac{1}{2}\kappa_\phi \bar{n} T \,.$$

Indeed, this results from the combination of conditional dissipation $L_b(t)$ and a noise process leading to leakage out of the code space. Indeed, all leakage at a rate $\kappa_l$ of the target cat qubit would lead to phase-flips of the control cat qubit at a rate $\kappa_l/2$.

*Toffoli gate.* The effect induced by photon loss during the execution of the Toffoli gate can be derived in the same way as for the CNOT. A photon loss occurring on one of the two control modes $a$, $b$ does not propagate to the other modes and results in a dephasing error $Z_1$ and $Z_2$, respectively. When the target mode $c$ loses a photon, it gives rise to a correlated error between the three modes. More precisely, the noisy Toffoli operation $\mathcal{E}_{a,b,c}$ can be decomposed into a perfect Toffoli operation, again denoted by

$$\tilde{\rho} = \text{Toffoli}\, \rho\, \text{Toffoli}\,,$$

followed by a noise process modelled by the Kraus map

$$\mathcal{E}_{a,b,c}(\rho) = \sum_{k=1,2,3,4} M_k \tilde{\rho} M_k^\dagger \,,$$

$$\begin{aligned}
M_1 &= \sqrt{\bar{n}\kappa_1 T} Z_1 \,, \\
M_2 &= \sqrt{\bar{n}\kappa_1 T} Z_2 \,, \\
M_3 &= \sqrt{\bar{n}\kappa_1 T}(\cos r(I_1 I_2 - \mathcal{Z}_{12}) - i\sin r\, \mathcal{Z}_{12}) Z_3 \,, \\
M_4 &= \sqrt{\bar{n}\kappa_1 T}(\sin r(I_1 I_2 - \mathcal{Z}_{12}) - i\cos r\, \mathcal{Z}_{12}) Z_3 \,,
\end{aligned}$$

where $\mathcal{Z}_{12} = \frac{1}{4}(I_1 I_2 - Z_1 - Z_2 - Z_1 Z_2)$ acts on the two control cat qubits.

Because of the analogies in the way the CNOT and the Toffoli gates are implemented, it is useful to think of the Toffoli gate as a CNOT where the control state $|-\alpha\rangle$ is replaced by $|-\alpha, -\alpha\rangle$. In particular, the methods of [71] that we used to characterize the effect of non-adiabaticity predict similar results for the Toffoli gate. We anticipate that the effect of the finite gate time is to dephase the "trigger" state $|-\alpha, -\alpha\rangle$ with respect to the other three possible states of the pair of control cat qubits. In terms of Pauli operator, this only results in phase-flip

errors $Z_1$, $Z_2$ and $Z_1 Z_2$ on the two control cat qubits with equal probability $p \approx 0.085(\bar{n}\kappa_2 T)^{-1}$ but it does not cause any error on the target cat qubit, or bit-flip type errors. We note that the thorough analysis of [73] include the non-adiabatic phase-flip errors for the Toffoli gate and are in accordance with these predictions.

## 4.4  Towards experimental realization of bias-preserving gates

We now discuss how all the different operations proposed above could be realized within the framework of circuit QED. As it will become clear throughout this section, all the recipes for realizing these operations belong to the well studied class of parametric methods with circuit QED. The multi-wave mixing property of Josephson junctions in a transmon or in a more elaborate circuit such as the ATS [39] can be combined with the application of microwave drives (also called pumps) at well chosen frequencies to realize the stabilization of the cat qubits, and to process the information encoded in them. Throughout this subsection, one can roughly judge of whether the implementation that we propose is reasonable for near-term experiments. Indeed, while low orders parametric processes (at most quadratic or cubic) are now more and more common in the framework of circuit QED, it remains a formidable challenge to engineer higher order non-linearities with sufficient strength. In all of the implementations proposed below, we restrict ourselves to (rather) low order non-linearities with these facts in mind, such that our whole scheme shall seem reasonable to implement in a near future. Specifically, the most difficult operation to implement experimentally (according to the non-linearity order metric) should be the feed-forward Hamiltonian required for the reduction of the phase-flip error rate of the Toffoli gate.

$Z(\theta)$ *gate.* The use of quantum Zeno dynamics to perform bias-preserving rotations around the $Z$-axis of the cat qubit was demonstrated experimentally in [50]. In addition to the (time-independent) two-photon dissipation realized as detailed in Section 3, the continuous rotation around the $Z$ axis is triggered by turning on a weak resonant drive at the frequency of the cat qubit mode $\omega_a$.

$ZZ(\theta)$ *gate.* The Hamiltonian required for the $Z_1 Z_2(\theta)$ gate is the following "beam-splitter" Hamiltonian

$$H_{BS} = \epsilon_{Z_1 Z_2}(\boldsymbol{a}\boldsymbol{b}^\dagger + \boldsymbol{b}\boldsymbol{a}^\dagger),$$

where $\boldsymbol{a}$ and $\boldsymbol{b}$ are each hosting a cat qubit.

This Hamiltonian was recently realized experimentally [74], using the four-wave mixing capability of the Josephson junction in presence of two pump tones. More precisely, denoting $\xi_1$, $\xi_2$ and $\omega_1, \omega_2$ the normalized amplitudes and frequencies of the two pumps, respectively, and by $\boldsymbol{c}$ the anharmonic mode of the transmon used in the bridge configuration to mediate the coupling between $\boldsymbol{a}$ and $\boldsymbol{b}$, the Hamiltonian of the system in a displaced rotating frame is given by

$$H = -E_J \cos[\varphi_a(\boldsymbol{a}e^{-i\omega_a t} + \boldsymbol{a}^\dagger e^{i\omega_a t}) + \varphi_b(\boldsymbol{b}e^{-i\omega_b t} + \boldsymbol{b}^\dagger e^{i\omega_a t})$$
$$+ \varphi_c(\boldsymbol{c}e^{-i\omega_c t} + \boldsymbol{c}^\dagger e^{i\omega_c t} + \xi_1 e^{-i\omega_1 t} + \xi_1^* e^{i\omega_1 t} + \xi_2 e^{-i\omega_2 t} + \xi_2^* e^{i\omega_2 t})].$$

Expanding the cosine to second order and keeping the non-rotating term when the frequency matching condition $\omega_1 - \omega_2 = \omega_a - \omega_b$ is verified produces the required beam-splitter interaction

$$H_{\text{int}}/\hbar = g(e^{i\varphi}\boldsymbol{a}\boldsymbol{b}^\dagger + e^{-i\varphi}\boldsymbol{a}^\dagger \boldsymbol{b}),$$

where $\varphi$ is determined by the relative phase of the two drives and the coupling coefficient is

$$g = E_J \varphi_a \varphi_b \varphi_c^2 |\xi_1||\xi_2| = \sqrt{\chi_{ac}\chi_{bc}}|\xi_1||\xi_2|,$$

where $\chi_{ac}$ (resp. $\chi_{bc}$) is effective cross-Kerr strength between $\boldsymbol{a}$ and $\boldsymbol{c}$ (resp. $\boldsymbol{b}$ and $\boldsymbol{c}$).

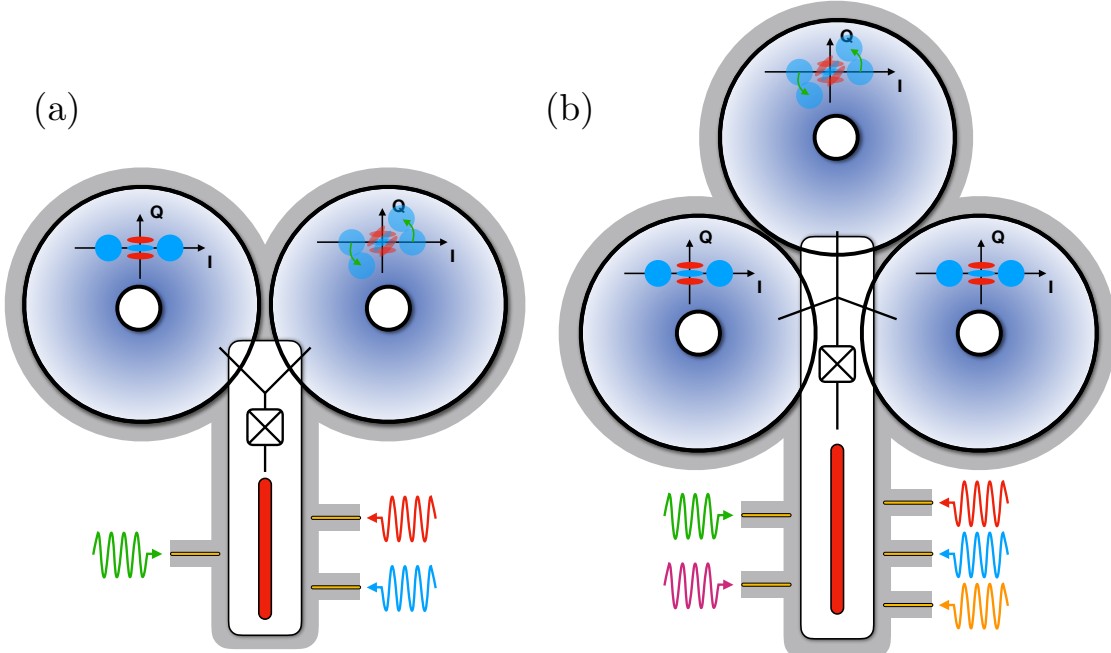

Figure 16: Proposal for an experimental implementation of bias-preserving CNOT and Toffoli gates for cat qubits using a transmon as the non-linear element. A similar sketch can be drawn to use a fluxed pumped ATS instead as the source of non-linearity. (a) Setup for implementing a bias-preserving CNOT gate. The cat qubits are encoded in high-Q cylindrical post-cavities (in blue, resonance frequencies $\omega_a$ and $\omega_b$). The two cavities are coupled via a Y-shape transmon as in [75] to a low-Q stripline resonator (in red, resonance frequency $\omega_d$) playing the role of the buffer mode. The system is driven with three micro-wave pumps at frequencies $\omega_1 = 2\omega_b - \omega_d$, $\omega_2 = (\omega_a - \omega_d)/2$, $\omega_3 = \omega_d$. (b) Similar setup for implementing a bias-preserving Toffoli gate with three cat qubits encoded in high-Q post-cavities (frequencies $\omega_a, \omega_b, \omega_c$) all coupled to a single stripline resonator (frequency $\omega_d$). The system is driven with five micro-wave pumps at frequencies $\omega_1 = 2\omega_c - \omega_d$, $\omega_2 = \omega_a + \omega_b - \omega_d$, $\omega_3 = (\omega_a - \omega_d)/2$, $\omega_4 = (\omega_b - \omega_d)/2$, and $\omega_5 = \omega_d$. Reproduced with permission from Ref. [9].

*X gate.* The realization of the *X* gate requires to modify the two-photon pumping scheme continuously in time to implement the effective dissipation operator

$$\kappa_2 \mathcal{D}\left[\boldsymbol{a}^2 - \exp(2i\pi t/T)\alpha^2\right],$$

In section 3.1, we have seen that the complex number $\alpha$ that parametrizes the two-dimensional cat qubit subspace is given by $\alpha = \sqrt{-\frac{\epsilon_d}{g_2}}$. Thus, the phase of this complex number can be tuned by changing the phase of the resonant drive applied on the buffer $\epsilon_d$ between 0 and $2\pi$ in a time $T$. Hence, the realization of the dissipative part of the *X* gate is actually a straightforward modification of the two-photon pumping scheme already realized experimentally.

In order to remove the phase-flip errors induced by the non-adiabaticity of this variation, one can additionally implement a Hamiltonian of the form $-\Delta \boldsymbol{a}^\dagger \boldsymbol{a}$ with $\Delta = \pi/T$. This can be done by taking the pump at frequency $2\omega_a - \omega_b - 2\Delta$ instead of $2\omega_a - \omega_b$ and furthermore detuning the drive $\epsilon_d$ from resonance by value $\Delta$.

*CNOT gate.*

The implementation of a CNOT gate between two cat qubits encoded in storage modes $\boldsymbol{a}$

and $b$ requires the implementation of the two dissipative channels

$$\kappa_2 \mathcal{D}[a^2 - \alpha^2],$$
$$\kappa_2 \mathcal{D}\left[b^2 - \alpha^2 - \frac{\alpha}{2}(1 - e^{\frac{2i\pi t}{T}})(a - \alpha)\right].$$

The implementation of the second one can be realized by coupling the two storage modes $a$ and $b$ to a buffer mode $d$, using a Y-shape transmon similar to [75] or a fluxed pump ATS (see Figure 16). Driving the buffer mode at three different frequencies

$$\omega_1 = 2\omega_b - \omega_d,$$
$$\omega_2 = \tfrac{1}{2}(\omega_a - \omega_d),$$
$$\omega_3 = \omega_d,$$

one can engineer an interaction Hamiltonian of the form

$$H_{\text{CNOT}} = (g_{bd}b^2 d^\dagger + g_{bd}^* b^{2\dagger}d) + (g_{ad}ad^\dagger + g_{ad}^* a^\dagger d) + (\epsilon_d d^\dagger + \epsilon_d^* d).$$

Note that following the experiment [74], the "beam-splitter" conversion triggered by the pump at frequency $\omega_2$ could also be realized by two pumps at frequencies $\omega_2, \omega_2'$ verifying the matching condition $\omega_2 - \omega_2' = \omega_a - \omega_d$.

In this interaction Hamiltonian, the first term $g_{bd}b^2 d^\dagger + g_{bd}^* b^{2\dagger}d$ models the exchange of two storage photons at frequency $\omega_b$ with one buffer photon at frequency $\omega_d$ via a pump photon at frequency $\omega_1$. The second term $g_{ad}ad^\dagger + g_{ad}^* a^\dagger d$ models the exchange of one storage photon at frequency $\omega_a$ with one buffer photon at frequency $\omega_d$ via two pump photons at frequency $\omega_2$. The amplitudes and phases of $g_{bd}$ and $g_{ad}$ are modulated by the amplitude and phase of the corresponding pumps. Finally, the last term $\epsilon_d d^\dagger + \epsilon_d^* d$ models the resonant interaction of the drive at frequency $\omega_d$ with the buffer mode. Similarly to the driven two-photon dissipation, one can adiabatically eliminate the highly dissipative buffer mode to achieve an effective dissipation operator

$$\kappa_2 \mathcal{D}[b^2 + c_a a + c]$$

where the dissipation rate $\kappa_2$ is roughly given by $4|g_{bd}|^2/\kappa_d$, $\kappa_d$ being the loss rate of the buffer mode, the complex constant $c_a$ is given by $g_{ad}/g_{bd}$ and the complex constant $c$ by $\epsilon_d/g_{bd}$. Similarly to the $X$-operation, it is clear that by varying the amplitudes and phases of the pump at frequency $\omega_2$ and the resonant drive at frequency $\omega_d$, one can engineer a dissipation operator with time-varying constants $c_a$ and $c$ given by

$$c_a(t) = -\frac{\alpha}{2}\left(1 - e^{\frac{2i\pi t}{T}}\right),$$
$$c(t) = -\frac{\alpha^2}{2}\left(1 + e^{\frac{2i\pi t}{T}}\right).$$

This corresponds to the dissipator required for the bias-preserving CNOT operation. Importantly, the time-dependent function $c_a$ takes the value 0 at times $t = 0$ and $t = T$. For this reason, before and after the gate, the two cat qubits involved in the CNOT are defined by their own local oscillators. The fluctuations of the pumps during the execution of the gate merely result in a slight modification of the geometric paths taken. This can only lead to small fluctuations of the geometric phase and therefore an effective phase-flip type error. The phase-flip probability induced by the non-adiabaticity of the evolution can be reduced by adding the effective Hamiltonian

$$H = \frac{1}{2}\frac{\pi}{T}\frac{a - \alpha}{2\alpha} \otimes (b^\dagger b - |\alpha|^2) + \text{h.c.}$$

Such a Hamiltonian has also been recently implemented using a detuned parametric pumping method [72].

*Toffoli gate.* In order to realize a bias-preserving Toffoli gate between three cat qubits encoded in storage modes $\boldsymbol{a}$, $\boldsymbol{b}$ and $\boldsymbol{c}$, further to two-photon driven dissipation on the two control cat qubits, a time-dependent dissipator given by

$$\kappa_2 \mathcal{D}\left[\boldsymbol{c}^2 - \alpha^2 + \frac{1}{4}\left(1 - e^{\frac{2i\pi t}{T}}\right)(\boldsymbol{ab} - \alpha(\boldsymbol{a} + \boldsymbol{b}) + \alpha^2)\right]$$

is required. Similarly to the CNOT gate, a way to achieve this is to couple the three modes to a highly dissipative buffer mode as shown in Figure 16. Driving the buffer mode at five different frequencies $\omega_1 = 2\omega_c - \omega_d$, $\omega_2 = \omega_a + \omega_b - \omega_d$, $\omega_3 = (\omega_a - \omega_d)/2$, $\omega_4 = (\omega_b - \omega_d)/2$, and $\omega_5 = \omega_d$, one can engineer an effective interaction Hamiltonian of the form

$$\begin{aligned}
\boldsymbol{H}_{\text{Toffoli}} = (g_{cd}\boldsymbol{c}^2\boldsymbol{d}^\dagger + g_{cd}^*\boldsymbol{c}^\dagger\boldsymbol{d}) + (g_{abd}\boldsymbol{abd}^\dagger + g_{abd}^*\boldsymbol{a}^\dagger\boldsymbol{b}^\dagger\boldsymbol{d}) + (g_{ad}\boldsymbol{ad}^\dagger + g_{ad}^*\boldsymbol{a}^\dagger\boldsymbol{d}) \\
+ (g_{bd}\boldsymbol{bd}^\dagger + g_{bd}^*\boldsymbol{b}^\dagger\boldsymbol{d}) + (\epsilon_d\boldsymbol{d}^\dagger + \epsilon_d^*\boldsymbol{d}).
\end{aligned}$$

Here again, all these effective terms are achieved in a parametric manner and using the 4-wave mixing property of the Josephson junction. The amplitude and phase of each interaction term can be modulated by the amplitude and phase of the associated pump. After the adiabatic elimination of the buffer mode, we achieve a dissipation operator

$$\kappa_2 \mathcal{D}[\boldsymbol{c}^2 + c_{ab}\boldsymbol{ab} + c_a\boldsymbol{a} + c_b\boldsymbol{b} + c],$$

where $\kappa_2$ is given by $4g_{cd}^2/\kappa_d$, and the complex constants $c_{ab} = g_{abd}/g_{cd}$, $c_a = g_{ad}/g_{cd}$, $c_b = g_{bd}/g_{cd}$, $c = \epsilon_d/g_{cd}$. By varying the amplitudes and phases of the pumps in time, we obtain time-varying constants

$$\begin{aligned}
c_{ab}(t) &= \frac{1}{4}\left(1 - e^{\frac{2i\pi t}{T}}\right), \\
c(t) &= -\frac{\alpha^2}{4}\left(3 + e^{\frac{2i\pi t}{T}}\right), \\
c_a(t) = c_b(t) &= -\frac{\alpha}{4}\left(1 - e^{\frac{2i\pi t}{T}}\right).
\end{aligned}$$

This implements a bias-preserving Toffoli gate between the cat qubits encoded in the three modes $\boldsymbol{a}, \boldsymbol{b}$ and $\boldsymbol{c}$. Here again, it should be noted that the functions $c_{ab}$, $c_a$, $c_b$ vanish at the beginning and at the end of the gate execution, so that each cat qubit gets back to being defined by its own local oscillators. Similarly to the CNOT gate, the pump fluctuations during the gate only result in a slight increase in the rate of phase-flip type errors, but do not lead to unsuppressed bit-flip type ones. In order to reduce the phase-flip probability induced by the non-adiabaticity, we use an additional Hamitonian

$$\boldsymbol{H} = -\frac{1}{2}\frac{\pi}{T}\frac{\boldsymbol{a} - \alpha}{2\alpha} \otimes \frac{\boldsymbol{b} - \alpha}{2\alpha} \otimes (\boldsymbol{c}^\dagger\boldsymbol{c} - |\alpha|^2) + \text{h.c.}$$

This Hamiltonian is the most complicated parametric Hamiltonian to realize in this scheme. While, in theory, it could be realized as previously by using a high-order multi-wave mixing with appropriate pumps, this approach would produce the required term with a very small amplitude. Instead, a better strategy to engineer this Hamiltonian could be to use a cascade of low-order multi-wave mixing such as the ones realized [76, 77]. However, due to the complexity of engineering this higher order Hamiltonian, perhaps the first generation of Toffoli gates could be implemented without this improvement.

## 5 Conclusion

In these notes, after a general introduction to quantum error correction, autonomous error correction and bosonic codes as hardware shortcuts, we have focused on dissipation based cat qubits. We have demonstrated that such qubits can be seen as qubits where one component of noise (bit-flips) is robustly and exponentially suppressed by increasing the average number of photons in the Schrödinger cat state that encodes the information. We have overviewed the experimental approaches to realize and stabilize such a qubit using superconducting circuits. We have also provided a thorough discussion of logical gates that can be performed on such qubits while preserving the exponential bit-flip suppression.

Note that, more recently another type of cat-qubit confinement based on Hamiltonian Kerr effect has been considered and similar studies on bit-flip suppression, bias-preserving logical gates and their experimental realization have been performed. This topic has not been discussed in these notes. The interested reader can find an extensive overview of this topic in the references [66, 69, 78–81]. In particular, the two recent references [80, 81] provide a comparison between the two types of Hamiltonian and dissipative confinement.

Finally, we do not discuss here how such biased noise qubits can lead to a significant overhead reduction in realizing a fault-tolerant quantum processor. This topic has been thoroughly discussed in the references [9, 63, 64, 73, 82–86]. In particular, the three references [9, 73, 86] focus on the case of cat qubits and propose a fault-tolerant approach based on a repetition code or a thin surface code of cat qubits. These references demonstrate how the set of elementary bias-preserving operations introduced in Section 4 are sufficient to reach a universal set of fault-tolerant gates at the level of the repetition code.

## Acknowledgements

These lecture notes summarize the work that has been done throughout the past 10 years in collaboration with many researchers, students and postdocs at Yale University (Applied Physics Department and Yale Quantum Institute) and at Quantic team (a joint team between Inria, Mines Paristech, ENS, Sorbonne Université and CNRS based in Paris). Mazyar Mirrahimi is grateful to his colleagues at Yale, Michel Devoret, Steven Girvin, Robert Schoelkopf, Liang Jiang, Leonid Glazman and their group members for having welcomed him and his students from 2011 to 2019, the period through which most of these ideas have been developed. The authors also thank Zaki Leghtas, Pierre Rouchon, Alain Sarlette, Philippe Campagne-Ibarcq and all other members of Quantic team in Paris who have been developing and pushing these ideas both towards their experimental realization and towards a general mathematical theory of quantum systems.

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
