# Peer review of "Quantum computation with cat qubits"

_SciPost Physics Lecture Notes, doi:SciPost Phys. Lect. Notes 72 (2023)_

## Round 2 · Referee Report · Anonymous · 2022-12-16

Strengths

Very pedagogical.
Some calculations are well detailed and insightful.

Weaknesses

The level of detail (e.g. in calculations) is fluctuating a lot in the text. Sometimes, notations are not introduced. This might be ok for an already experienced reader, but not for a beginner.

Report

The journal's criteria are met.

Requested changes

p. 3
- The term dynamics is not obviously applying to formula (1). A clearer approach would be to write the time dependence of the density matrix in the definition of the quantum map.
- One should add on top of the condition that the single bit errors is small, that errors should not be correlated
p. 4
equation (2)
perhaps a clearer statement would be to include: if and only if there exists an Hermitian matrix c_mu,nu such that forall mu, nu, etc..
p. 5
- in the derivation, writing sqrt(d_mu d_nu) would be more logical at the second step of the calculation
- One should precise that the |g_mu> are an orthonormal basis of E
p. 8
- does the “decay time” correspond to the inverse of the total decay time or only the coupling loss ?
- the rate in the Geerlings experiment should be read kappa_c^0 instead of kappa_c
p. 10
(which is the one pursued in these notes) : this parenthesis is redundant, as this is restated later.
p. 11
- shouldn’t the state after 1 photon loss be |psi(n-1 mod 4)_alpha> instead of |psi(n+1 mod 4)_alpha>?
- a clearer phrasing for “therefore the measurement of the photon number parity […] occurrence of a single photon loss” could be : “therefore measuring a change in the photon number parity indicates the occurrence of a single photon loss”
p. 21
fig. 6 : There are no x or y labels in the plot (even if obvious)
p. 41
it is not clear to me what is epsilon in the bound for the Lyapounov exponent
p. 42
the Clifford hierarchy has not been defined in the text. A reference would be welcome here.
p. 44
the meaning of "Fortunately, no gate is lost at this step" is obscure, given that the previous paragraph restricted some gates, such as the Hadamard gate.
p. 45
The caption is redundant with the text
p. 46
the finite gate time of the gates -> the finite time of the gates*
p. 61
fig 15 : the pale magenta dots ("pale purple" in the text) are not visible. The font size of labels is very small, it could be wise to increase the figure size.
p. 64
- typo : dynamicsto - >dynamics to
- Expanding the cosine : it should be mentionned that the expansion is done at the second order
p. 66
this hamiltonian has already been introduced previously under the name "feed forward", so the ref 70 could also be placed before (p59)

  • validity: top
  • significance: top
  • originality: top
  • clarity: high
  • formatting: good
  • grammar: excellent

Author:  Mazyar Mirrahimi  on 2023-01-05  [id 3210]

(in reply to Report 1 on 2022-12-16)

We would like to thank the referee for the careful reading of our manuscript and for their comments/suggestions that we find very relevant for its improvement. We have included nearly all of the proposed modifications. Here are the responses to the few questions by the referee:

"p.5 - In the derivation, writing sqrt(d_mu d_nu) would be more logical at the second step of the calculation"

Response: d_\mu is more accurate as it comes from the multiplication on the left side by E_\mu and the right side by E_\mu^\dag.

"p.8- - Does the “decay time” correspond to the inverse of the total decay time or only the coupling loss ?"

Response: We imagine that the referee asks about the decay rate of the cavity $\kappa_c$. It represents the inverse of the total cavity decay time.

"p.8- - the rate in the Geerlings experiment should be read kappa_c^0 instead of kappa_c”

Response: We do not understand this comment. Here $\kappa_c$ represents the cavity decay rate.

"p.41 -it is not clear to me what is epsilon in the bound for the Lyapounov exponent"

Response: It still corresponds to |g_2|/\kappa_b as defined earlier.

"p. 66 this hamiltonian has already been introduced previously under the name "feed forward", so the ref 70 could also be placed before (p59)"

Response: We prefer to keep the reference here as this is in this section that we talk about experimental implementations.

Anonymous on 2023-03-02  [id 3420]

(in reply to Mazyar Mirrahimi on 2023-01-05 [id 3210])

I am happy with the answers and the modifications.

---

## Editorial Decision

published